# Wasserstein Coreset via Sinkhorn Loss

**Haoyun Yin**                                                                          *yin164@purdue.edu*
*Department of Statistics*
*Purdue University*

**Yixuan Qiu**                                                                        *qiuyixuan@sufe.edu.cn*
*School of Statistics and Data Science*
*Shanghai University of Finance and Economics*

**Xiao Wang**                                                                          *wangxiao@purdue.edu*
*Department of Statistics*
*Purdue University*

**Reviewed on OpenReview:** *https://openreview.net/forum?id=DrMCDS88IL*

## Abstract

Coreset selection, a technique for compressing large datasets while preserving performance, is crucial for modern machine learning. This paper presents a novel method for generating high-quality Wasserstein coresets using the Sinkhorn loss, a powerful tool with computational advantages. However, existing approaches suffer from numerical instability in Sinkhorn's algorithm. We address this by proposing stable algorithms for the computation and differentiation of the Sinkhorn optimization problem, including an analytical formula for the derivative of the Sinkhorn loss and a rigorous stability analysis of our method. Extensive experiments demonstrate that our approach significantly outperforms existing methods in terms of sample selection quality, computational efficiency, and achieving a smaller Wasserstein distance.

## 1 Introduction

In the last decade, big data stored in databases have grown massively and become difficult to capture, form, store, manage, share, analyze, and visualize using typical database software tools (Sagiroglu & Sinanc, 2013). To address this challenge, the coreset technique, also known as representative sampling, has been developed to select a handful of representative samples that summarize the original dataset. Formalizing the notion of *representative* requires care, as a representative sample for a clustering algorithm may differ from that for a classification algorithm (Claici et al., 2018). Consequently, coreset construction methods usually depend on specific tasks, such as clustering (Feldman, 2020), classification (Coleman et al., 2019), logistic regression (Wang et al., 2018), mixture models (Lucic et al., 2018), low-rank approximation (Cohen et al., 2016), matrix sketching(Drineas et al., 2012), and Bayesian inference (Campbell & Broderick, 2018).

In this context, it is essential to establish a clear framework for discussing coreset problems. Let $B$ denote the number of samples in a coreset, and define the set $\Delta_B := \{\nu_B \mid \nu_B = \frac{1}{B} \sum_{i=1}^{B} \delta_{X_i}, X_i \in \mathcal{X}\}$, which consists of all probability measures supported on $B$ points, each having mass $\frac{1}{B}$. Given a measure $\xi$ on the metric space $(\mathcal{X}, d)$ and a particular objective function $f : \mathcal{X} \to \mathbb{R}$, we say that $\nu_B \in \Delta_B$ is a $\delta$-coreset of $\xi$ for the specific task defined by $f$ if

$$|\mathbb{E}_\xi[f(X)] - \mathbb{E}_{\nu_B}[f(X)]| < \delta, \quad \text{with } B \ll N.$$

To extend this notion beyond a single task, we can consider a family $\mathcal{F}$ of objective functions $f : \mathcal{X} \to \mathbb{R}$. We define the integral probability metric (IPM) associated with $\mathcal{F}$ as

$$d_{\mathcal{F}}(\xi, \nu) = \sup_{f \in \mathcal{F}} |\mathbb{E}_{\xi}[f(X)] - \mathbb{E}_{\nu}[f(X)]|. \tag{1}$$

A measure $\nu_B \in \Delta_B$ is then called a task-agnostic $\delta$-coreset of $\xi$ if $d_{\mathcal{F}}(\xi, \nu_B) < \delta$, thereby controlling the approximation error uniformly over all functions in $\mathcal{F}$. This task-agnostic formulation emphasizes the coreset's ability to perform well for a wide class of objective functions, rather than being tailored to a single specific task.

In particular, consider the following two classes of functions:

$$\mathcal{F}_1 = \{f : |f(x) - f(y)| \leq d(x, y), \ \forall x, y \in \mathcal{X}\}$$

and

$$\mathcal{F}_2 = \{f : \|f\|_{H^1(\mu)} \leq 1\},$$

where $H^1$ is the Sobolev space $\{f \in L^2 : \partial x_i f \in L^2\}$ endowned with the norm

$$\|f\|_{H^1(\mu)} = \left( \int_{\mathcal{X}} |f(x)|^2 \, d\mu(x) \ + \ \int_{\mathcal{X}} \|\nabla f(x)\|^2 \, d\mu(x) \right)^{\frac{1}{2}}.$$

For these function classes, with $W_1$ and $W_2$ being the 1-Wasserstein distance and 2-Wasserstein distance, respectively, one can show that:

(a) $d_{\mathcal{F}_1}(\mu, \mu_B) = W_1(\mu, \mu_B)$ (Santambrogio, 2015),

(b) $d_{\mathcal{F}_2}(\mu, \mu_B) \leq \sqrt{C} W_2(\mu, \mu_B)$, where $C$ is a constant (Peyré et al., 2019).

These results indicate that the Wasserstein distance provides a suitable upper bound for the relevant integral probability metrics. Consequently, using a Wasserstein-based objective in coreset construction ensures that the resulting coreset is task-agnostic. By minimizing the Wasserstein distance, we inherently minimize an upper bound on the IPM for a broad range of objective functions $f \in \mathcal{F}_1$ or $f \in \mathcal{F}_2$, and approximates the target measure with respect to a large family of tasks.

These insights into task-agnostic coresets rely on integral probability metrics and our specific focus on Wasserstein-based bounds. However, task-agnostic coreset constructions have also been explored through other frameworks. A few examples of task-agnostic coresets are the mse-rep-points (Fang & Wang, 1993), energy-rep-points (Mak & Joseph, 2018), and mmd-rep-points (Dwivedi & Mackey, 2021).

The *mse-rep-points*, or mean squared error representative points, aim to minimize the expected distance from a random point drawn from the distribution to its closest representative point. These points, also known as principal points, have been effectively used in various applications, including quantizer design and optimal stratified sampling. In practice, they are often generated by performing k-means clustering on a large batch sample from the distribution, and then using the converged cluster centers as representative points. However, a primary drawback of mse-rep-points is that they do not necessarily converge to the underlying distribution, which limits their effectiveness in some scenarios (MacQueen et al., 1967; Graf & Luschgy, 2007).

The second class of coresets, *energy-rep-points* aim to minimize a measure of statistical potential. This class includes methods such as minimum-energy designs (Joseph et al., 2015) and minimum Riesz energy points (Borodachov et al., 2014). Although these point sets converge in distribution, their convergence rate is slow, and the construction of such point sets can become computationally expensive as the dimensionality of the data increases. Mak & Joseph (2018) improved these methods and try to generate support points by minimizing the energy distance. However, energy distance, regarded as a kernel discrepancy for nonuniform distributions with the specific kernel choice of the negative Euclidean norm, may not be well-suited for high-dimensional manifold data. Such data is commonly encountered in contemporary machine learning tasks, where the properties of the negative Euclidean norm can pose challenges.

A third class of task-agnostic coresets, the *mmd-rep-points*, aims to minimize the maximum mean discrepancy (MMD) between the empirical distribution and the distribution of the representative points. The MMD is a kernel-based metric that quantifies discrepancies between distributions, and mmd-rep-points are constructed by selecting points that reduce this measure. However, the effectiveness of mmd-rep-points is highly sensitive to the choice of kernel, which can significantly influence the resulting approximation. Moreover, the method proposed by Dwivedi & Mackey (2021) requires $O(n \min(n, p))$ memory storage and does not support mini-batch operations, making it challenging to apply to large-scale datasets frequently encountered in contemporary machine learning settings.

In contrast, the Wasserstein distance, the core of optimal transport (OT, Villani (2009)), is recognized as the natural geometry for probability measures for its efficacy, and has gained widespread use across multiple machine learning domains, including generative modeling (Arjovsky et al., 2017) and domain adaptation (Courty et al., 2017). Let $(\mathcal{X}, d)$ be a metric space that is Polish. For $p \in [1, \infty)$, the $p$-Wasserstein distance is defined as $W_p(\mu, \nu) = \left( \inf_{\gamma \in \Pi(\mu, \nu)} \int_{\mathcal{X} \times \mathcal{X}} d(x, y)^p \gamma(dx, dy) \right)^{1/p}$, where $\Pi(\mu, \nu)$ is the set of all joint distributions with marginals $\mu$ and $\nu$. According to Theorem 1.1 in Kloeckner (2012), we know that if $\mathcal{X} = \mathbb{R}^d$, and let $\nu_B^*$ denote the minimizer of $W_p(\xi, \nu_B)$ in the space of all measures supported on at most $B$ points in $\mathcal{X}$, *i.e.*,

$$\nu_B^* = \operatorname*{argmin}_{\nu_B \in \left\{ \nu : \nu = \sum_{i=1}^{B} \omega_i \delta_{X_i} \right\}} W_p(\xi, \nu_B),$$

where $\{\omega_i\}_{i=1}^B$ is any set of weights, $X_i \in \mathcal{X}$ and $\delta_X$ indicates the Dirac measure at $X$, then it holds that $W_p(\xi, \nu_B^*) = \Theta(n^{-1/d})$.

Despite the various appealing theoretical properties, one major barrier for the wide applications of the Wasserstein distance is the difficulty in computation, especially when the size of the dataset is large. For two discrete distributions, OT solves a linear programming problem of $nm$ variables, where $n$ and $m$ are the number of Diracs that define the two distributions. Assuming $n = m$, standard linear programming solvers for OT have a complexity of $\mathcal{O}(n^3 \log n)$ (Pele & Werman, 2009), which quickly becomes formidable as $n$ increases, except for some special cases (Peyré et al., 2019).

To resolve this issue, many approximate solutions to the Wasserstein distance have been proposed, among which the Sinkhorn loss has gained massive popularity (Cuturi, 2013). The Sinkhorn loss can be viewed as an entropic-regularized Wasserstein distance, which adds a smooth penalty term to the original objective function of OT. The Sinkhorn loss is attractive as its optimization problem can be efficiently solved, at least in exact arithmetics, via Sinkhorn's algorithm (Sinkhorn, 1964; Sinkhorn & Knopp, 1967), which merely involves matrix-vector multiplications and some minor operations. Therefore, it is especially suited to modern computing hardware, such as GPUs. Recent theoretical advancements indicate that Sinkhorn's algorithm achieves an $\varepsilon$-approximation to the unregularized OT problem with a computational complexity of $\mathcal{O}(n^2 \log(n) \varepsilon^{-2})$ (Dvurechensky et al., 2018). Meanwhile, the accelerated Sinkhorn algorithm improves this complexity to $\mathcal{O}(n^{7/3} \log(n)^{1/3} \varepsilon^{-4/3})$ (Lin et al., 2022).

However, one critical pain point of the Sinkhorn loss, though typically ignored in theoretical studies, is that Sinkhorn's algorithm is numerically unstable (Peyré et al., 2019). We show in numerical experiments that even for very simple settings, Sinkhorn's algorithm can quickly lose precision. Various stabilized versions of Sinkhorn's algorithm, though showing better stability, still suffer from slow convergence in these cases. Moreover, since modern deep generative models mostly rely on the gradient-based learning framework, it is crucial to use the Sinkhorn loss with differentiation support. One simple and natural method for differentiating the Sinkhorn loss is to unroll Sinkhorn's algorithm, adding every iteration to the auto-differentiation computing graph (Genevay et al., 2018; Cuturi et al., 2019). However, this approach is typically costly when the number of iterations are large. The slow convergence rate resulted from numerical instability of Sinkhorn's algorithm may further exacerbate this issue.

In this paper, we solve the coreset problem using the Sinkhorn loss. We rigorously analyze both the computation and differentiation of the Sinkhorn optimization problem, designing corresponding algorithms that are provably efficient and stable. Utilizing the analytic differentiation of Sinkhorn loss, we have designed a gradient-based algorithm for the strategic selection of the coreset, which is task-agnostic and suitable for

high-dimensional manifold data. As a result, it can be applied to a wide range of machine learning tasks. We have demonstrated the superior performance of our method in extensive simulation studies and practical applications with image data. Our major contributions are:

(i) We derive an analytic expression for the derivative of the Sinkhorn loss, which can be efficiently computed during the back-propagation phase in machine learning.

(ii) We have rigorously analyzed the advocated algorithms for Sinkhorn loss computation and differentiation (SLCD), and show that they have desirable efficiency and stability properties.

(iii) We designed a task-agnostic coreset selection method called Wasserstein coreset via Sinkhorn loss (WCSL) and highlighted its superior performance in low-budget scenarios, in extensive simulation studies and practical applications with imaging data.

This paper is organized as follows. In Section 2, we introduce the notation and provide the necessary background on the Sinkhorn loss. The construction of a Wasserstein coreset can be viewed as a two-layer optimization problem: the *outer* layer selects the coreset points, by minimizing the Sinkhorn loss. The *inner* layer computes the Sinkhorn loss, which requires optimization of the transport plan. In Section 3, we present the *outer* layer, our WCSL algorithm, and explain why computing the derivative of the Sinkhorn loss is essential. Section 4 addresses numerical challenges in the Sinkhorn algorithm and introduces the SLCD algorithm to efficiently compute and differentiate the Sinkhorn loss, resulting in the *inner* layer. Section 5 provides theoretical analysis on the convergence and stability of the SLCD algorithm, the consistency of the Wasserstein coreset, and the convergence properties of the WCSL algorithm. Finally, Section 6 presents numerical experiments that demonstrate the performance of WCSL in comparison to several alternative methods.

## 2 Background

Throughout this article we focus on discrete OT problems. We denote the $(n-1)$-dimensional probability simplex by $\Delta^n = \{w \in \mathbb{R}^n_+ : w^{\mathrm{T}}\mathbf{1}_n = 1\}$, and let $\mu = \sum_{i=1}^n a_i \delta_{x_i}$ and $\nu = \sum_{j=1}^m b_j \delta_{y_j}$ be two discrete probability measures supported on data points $\{x_i\}_{i=1}^n$ and $\{y_j\}_{j=1}^m$, respectively, where $a = (a_1, \ldots, a_n)^{\mathrm{T}} \in \Delta^n$ and $b = (b_1, \ldots, b_m)^{\mathrm{T}} \in \Delta^m$. Define $\Pi(a,b) = \{T \in \mathbb{R}^{n \times m}_+ : T\mathbf{1}_m = a, T^{\mathrm{T}}\mathbf{1}_n = b\}$, and let $M(\{x_i\}_{i=1}^n, \{y_j\}_{j=1}^m) \in \mathbb{R}^{n \times m}$ be a cost matrix with entries $M_{ij} = d(x_i, y_j)$ for any selected distance $d(\cdot, \cdot)$. Without loss of generality, we assume that $n \geq m$, as their roles can be exchanged. Then we can characterize the OT problem by the following optimization problem,

$$W(\mu, \nu) = \min_{P \in \Pi(a,b)} \langle P, M \rangle, \tag{2}$$

where $\langle A, B \rangle = \mathrm{tr}(A^{\mathrm{T}}B)$.

For the Sinkhorn loss, we use the following notation. For $x, y \in \mathbb{R}$, $x \wedge y$ means $\min\{x, y\}$. For a vector $v = (v_1, \ldots, v_k)^{\mathrm{T}}$, let $v^{-1} = (v_1^{-1}, \ldots, v_k^{-1})^{\mathrm{T}}$, $\tilde{v} = (v_1, \ldots, v_{k-1})^{\mathrm{T}}$, and use $\mathbf{diag}(v)$ to denote the diagonal matrix formed by $v$. Let $u = (u_1, \ldots, u_l)^{\mathrm{T}}$ be another vector, and use $u \oplus v$ to denote the $l \times k$ matrix with entries $(u_i + v_j)$. For a matrix $A = (a_{ij}) = (A_1, \ldots, A_k)$ with column vectors $A_1, \ldots, A_k$, let $\tilde{A} = (A_1, \ldots, A_{k-1})$, and $\mathrm{e}_\lambda[A]$ be the matrix with entries $e^{\lambda a_{ij}}$. The symbol $\odot$ denotes the elementwise multiplication operator between matrices or vectors. $\|\cdot\|$ and $\|\cdot\|_F$ stand for the Euclidean norm for vectors and Frobenius norm for matrices, respectively.

An optimal solution to (2), denoted as $P^*$, is typically called an optimal transport plan, and can be viewed as a joint distribution whose marginals coincide with $\mu$ and $\nu$. The optimal value $W(\mu, \nu) = \langle P^*, M \rangle$ is then called the Wasserstein distance between $\mu$ and $\nu$ if the cost matrix $M$ satisfies some suitable conditions (Proposition 2.2 of Peyré et al., 2019).

Solving the optimization problem (2) can be difficult even for moderate $n$ and $m$. One approach to regularizing the optimization problem is to add an entropic penalty term to the objective function, leading

to the entropic-regularized OT problem (Cuturi, 2013):

$$\tilde{S}_\lambda(\mu, \nu) = \min_{T \in \Pi(a,b)} \langle T, M \rangle - \lambda^{-1} h(T), \tag{3}$$

where $h(T) = \sum_{i=1}^n \sum_{j=1}^m T_{ij}(1 - \log T_{ij})$ is the entropy term. The new objective function is $\lambda^{-1}$-strongly convex on $\Pi(a,b)$, so (3) has a unique global solution, denoted as $T_\lambda^*$, *i.e.*,

$$T_\lambda^* = \underset{T \in \Pi(a,b)}{\operatorname{argmin}} \langle T, M \rangle - \lambda^{-1} h(T).$$

In this article, $T_\lambda^*$ is referred to as the Sinkhorn transport plan. To simplify the notation, we omit the subscript $\lambda$ in $T_\lambda^*$ hereafter when no confusion is caused. The entropic-regularized Wasserstein distance, also known as the Sinkhorn distance or Sinkhorn loss in the literature (Cuturi, 2013), is then defined as

$$S_\lambda(\mu, \nu) = \langle T^*, M \rangle. \tag{4}$$

It is worth noting that in the literature, $S_\lambda$ and $\tilde{S}_\lambda$ are sometimes referred to as the *sharp* and *regularized* Sinkhorn loss, respectively. In this article we focus on the sharp version $S_\lambda$, and simply call it the Sinkhorn loss for brevity, as $S_\lambda$ achieves a faster rate at approximating the Wasserstein distance than $\tilde{S}_\lambda$, suggested by the following proposition.

**Proposition 2.1** (Luise et al., 2018)**.** *There exist constants $C_1, C_2 > 0$ such that for any $\lambda > 0$, $|S_\lambda(\mu, \nu) - W(\mu, \nu)| \le C_1 e^{-\lambda}$ and $|\tilde{S}_\lambda(\mu, \nu) - W(\mu, \nu)| \le C_2/\lambda$. The constants $C_1$ and $C_2$ are independent of $\lambda$, and depend on $\mu$ and $\nu$.*

## 3 Wasserstein Coreset via Sinkhorn Loss (WCSL)

In this section, we will use the sinkhorn loss to generate the Wasserstein coreset. We call this method the Wasserstein coreset via Sinkhorn loss. The objective of WCSL is to find the empirical distribution of the Wasserstein coreset:

$$\nu_B^* = \underset{\nu_B \in \Delta_B}{\operatorname{argmin}} W(\xi, \nu_B). \tag{5}$$

As we have mentioned earlier, the standard solver of Wasserstein distance quickly becomes formidable as $n$ increases. To address this issue, we propose to use the Sinkhorn loss to approximate the Wasserstein distance in the coreset problem. The objective of the empirical distribution of the Wasserstein coreset via Sinkhorn loss is revised as:

$$\nu_B^* = \underset{\nu_B \in \Delta_B}{\operatorname{argmin}} S_\lambda(\xi, \nu_B). \tag{6}$$

The algorithm for WCSL (Algorithm 1) is an iterative procedure that seeks to minimize the Sinkhorn loss between the full measure and the measure uniformly supported on the coreset.

Given a measure $\xi$, the basic idea of the algorithm is to randomly sample an initial set $\mathcal{D}^{(0)}$ consisting of $B$ i.i.d. sample points from $\xi$, and then update the points we get with gradient descent based on the gradient of the Sinkhorn loss between the measure uniformly supported on our selected points and the measure $\xi$, until convergence. Then we output the optimized points $\mathcal{D}^*$ as the coreset, given by

$$\mathcal{D}^* = \underset{\{\mathbf{x}_i\}_{i=1}^B}{\operatorname{argmin}} S_\lambda \left( \xi, \frac{1}{B} \sum_{i=1}^B \delta_{\mathbf{x}_i} \right). \tag{7}$$

With a slight abuse of notation, we denote $S_\lambda\big(\{\mathbf{x}_i\}_{i=1}^{B_1}, \{\mathbf{y}_j\}_{j=1}^{B_2}\big)$ as an abbreviation for $S_\lambda\Big(\frac{1}{B_1} \sum_{i=1}^{B_1} \delta_{\mathbf{x}_i}, \frac{1}{B_2} \sum_{j=1}^{B_2} \delta_{\mathbf{y}_j}\Big)$, where $\{\mathbf{x}_i\}_{i=1}^{B_1}$ and $\{\mathbf{y}_j\}_{j=1}^{B_2}$ are two sets of points.

In detail of WCSL, we use one mini-batch of i.i.d sample $\mathcal{Y}^{(l,t)}$ from $\xi$ to update our coreset in each iteration. The preparation step within each iteration is to compute the matrix $M(\mathcal{D}^{(l,t)}, \mathcal{Y}^{(l,t)})$, which

---

**Algorithm 1** Algorithm for Wasserstein coreset via Sinkhorn loss (WCSL)

---

1: Sample $\mathcal{D}^{(0)} = \{\mathbf{x}_i^{(0)}\}_{i=1}^B \overset{\text{i.i.d.}}{\sim} \xi$. Set $l = 0$.
2: **repeat**
3:     Initialize $\mathcal{D}^{(l,0)} \leftarrow \mathcal{D}^{(l)}$
4:     **for** $t = 0, \ldots, T - 1$ **do**
5:         Resample $\mathcal{Y}^{(l,t)} = \{\mathbf{x}_m^{(l,t)}\}_{m=1}^b \overset{\text{i.i.d.}}{\sim} \xi$
6:         Compute $M(\mathcal{D}^{(l,t)}, \mathcal{Y}^{(l,t)})$ the distance matrix based on the selected distance.
7:         Update $\mathcal{D}^{(l,t+1)} \leftarrow \mathcal{D}^{(l,t)} + \theta \cdot \nabla_{\mathcal{D}^{(l,t)}} M \nabla_M S_\lambda(\mathcal{D}^{(l,t)}, \mathcal{Y}^{(l,t)})$ (Or Adam update)
8:     **end for**
9:     Update $\mathcal{D}^{(l+1)} \leftarrow \mathcal{D}^{(l,T)}$
10:     Set $l \leftarrow l + 1$.
11: **until** $S_\lambda(\mathcal{D}^{(l)}, \mathcal{D}^{(l-1)}) < \delta$
12: **return** $\mathcal{D}^{(l)}$

---

holds the pairwise distances, typically the squared $L_2$ norm, between the elements of $\mathcal{D}^{(l,t)}$ and $\mathcal{Y}^{(l,t)}$. The squared $L_2$ norm is chosen to ensure compatibility with the definition of the 2-Wasserstein distance ($W_2$). This distance matrix $M$ is then utilized to update $\mathcal{D}^{(l,t)}$ using the chain rule, as $\nabla_{\mathcal{D}^{(l,t)}} S_\lambda(\mathcal{D}^{(l,t)}, \mathcal{Y}^{(l,t)}) = \nabla_M S_\lambda(\mathcal{D}^{(l,t)}, \mathcal{Y}^{(l,t)}) \nabla_{\mathcal{D}^{(l,t)}} M$. In general, we can get $\nabla_{\mathcal{D}^{(l,t)}} M$ by automatic differentiation. However, the computation of $\nabla_M S_\lambda(\mathcal{D}^{(l,t)}, \mathcal{Y}^{(l,t)})$ poses a significant challenge, and this will be discussed in detail in Section 4, where the Algorithm 2 for the computation and differentiation of the Sinkhorn loss is presented.

## 4 Computation and Differentiation of the Sinkhorn Loss

To use the Sinkhorn loss in deep neural networks or other machine learning tasks, it is crucial to obtain the derivative of $S_\lambda(\mu, \nu)$ with respect to the distance matrix. Differentiating the Sinkhorn loss typically involves two stages: the solution stage and the differentiation stage. In the solution stage, the Sinkhorn loss or the transport plan is computed using some optimization algorithms, and in the differentiation stage, the derivative is computed using either an analytic expression or an automatic differentiation technique. Before we analyze both stages in detail, we first present the issues of Sinkhorn's algorithm.

### 4.1 Issues of Sinkhorn's Algorithm

In the existing literature, one commonly used method for the computation of the Sinkhorn loss is Sinkhorn's algorithm (Sinkhorn, 1964; Sinkhorn & Knopp, 1967). Unlike the original linear programming problem (2), the solution to the Sinkhorn problem has a special structure. Cuturi (2013) shows that the optimal solution $T^*$ can be expressed as

$$T^* = \mathbf{diag}(u^*) M_e \mathbf{diag}(v^*) \tag{8}$$

for some vectors $u^*$ and $v^*$, where $M_e = \left(e^{-\lambda M_{ij}}\right)$. Sinkhorn's algorithm starts from an initial vector $v^{(0)} \in \mathbb{R}_+^m$, and generates iterates $u^{(k)} \in \mathbb{R}_+^n$ and $v^{(k)} \in \mathbb{R}_+^m$ as follows:

$$u^{(k+1)} \leftarrow a \odot [M_e v^{(k)}]^{-1}, \quad v^{(k+1)} \leftarrow b \odot [M_e^{\mathrm{T}} u^{(k+1)}]^{-1}. \tag{9}$$

It can be proved that $u^{(k)} \to u^*$ and $v^{(k)} \to v^*$, and then the Sinkhorn transport plan $T^*$ can be recovered by (8).

Sinkhorn's algorithm is very efficient, as it only involves matrix-vector multiplication and other inexpensive operations. However, one major issue of Sinkhorn's algorithm is that the entries of $M_e = \left(e^{-\lambda M_{ij}}\right)$ may easily underflow when $\lambda$ is large, making some elements of the vectors $M_e v^{(k)}$ and $M_e^{\mathrm{T}} u^{(k+1)}$ close to zero. As a result, some components of $u^{(k+1)}$ and $v^{(k+1)}$ would overflow. Therefore, Sinkhorn's algorithm in its original form is unstable, and in practice the iterations (9) are typically carried out in the logarithmic scale, which we call the Sinkhorn-log algorithm for simplicity. In addition, there are some other works that also attempt to improve the numerical stability of Sinkhorn's algorithm (Schmitzer, 2019; Cuturi et al., 2022).

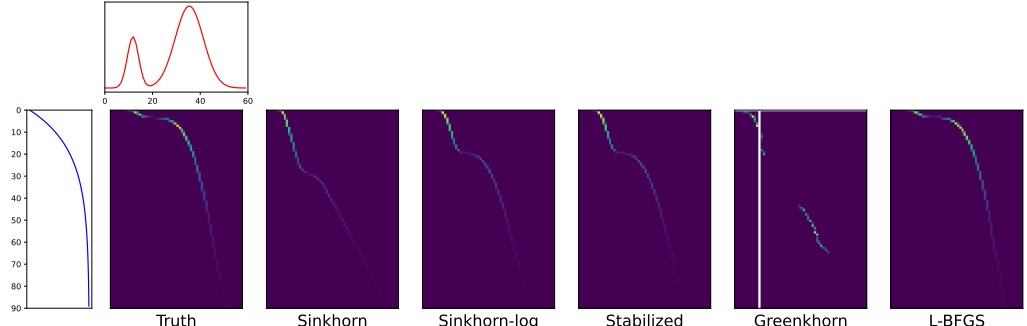

Figure 1: Visualization of Sinkhorn plans computed by different algorithms.

*Remark: Due to numerical instability and slow convergence, the transport plan obtained through Sinkhorn's algorithm and various widely used algorithms, deviates significantly from the ground truth. In contrast, the transport plan computed using L-BFGS demonstrates a near-perfect visual alignment with the true solution. Detailed experimental settings are provided in Appendix B.1.*

---

**Algorithm 2** Sinkhorn Loss Computation and Differentiation (SLCD)

---

1: **Solution Stage:**
2:       Use L-BFGS to solve $\beta^* = \text{argmin}_\beta f(\beta)$, where $f(\beta)$ is defined in (12) and its gradient $\nabla_{\tilde{\beta}} f$ in (13).
3:       Compute $\alpha^*$ using (11) and compute $T^*$ using (14).
4:       Compute the Sinkhorn loss using (4).
5: **Differentiation Stage:**
6:       Compute the analytical derivative using (15) from Theorem 4.1.

---

Despite the advancements of Sinkhorn's algorithm, one critical issue observed in practice is that Sinkhorn-type algorithms may be slow to converge, especially for small regularization parameters. This would severely slow down the computation, and may even give misleading results when the user sets a moderate limit on the total number of iterations.

In Figure 1, we show a motivating example to highlight these issues. Consider two probability measures for the Sinkhorn problem, and the true $T^*$ matrix is visualized in Figure 1, along with the solutions given by various widely-used algorithms, including Sinkhorn's algorithm, Sinkhorn-log, the stabilized scaling algorithm (Stabilized, Algorithm 2 of Schmitzer, 2019), and the Greenkhorn algorithm (Altschuler et al., 2017; Lin et al., 2022). In Figure 1, it is clear that the plans given by Sinkhorn's algorithm and Greenkhorn are farthest to the true value, and Greenkhorn generates NaN values reflected by the white stripes in the plot. In contrast, the stable algorithms Sinkhorn-log and Stabilized greatly improve them. Sinkhorn's algorithm and Sinkhorn-log are equivalent in exact arithmetics, so their numerical differences highlight the need for numerically stable algorithms. However, Sinkhorn-log and Stabilized still have visible inconsistencies with the truth even after 1000 iterations.

### 4.2   The Advocated Alternative for Sinkhorn Loss Computation

To solve the numerical instability and slow convergence issue of Sinkhorn's algorithm, we advocate an alternative scheme to solve the optimal plan $T^*$, as shown in the solution stage of Algorithm 2. Then we show both theoretically and empirically that this method enjoys great efficiency and stability.

Consider the dual problem of (3), which has the following form (Proposition 4.4 of Peyré et al., 2019):

$$\max_{\alpha,\beta} \mathcal{L}(\alpha,\beta) := \max_{\alpha,\beta} \alpha^{\mathrm{T}} a + \beta^{\mathrm{T}} b - \lambda^{-1} \sum_{i=1}^{n} \sum_{j=1}^{m} e^{-\lambda(M_{ij}-\alpha_i-\beta_j)}, \quad \alpha \in \mathbb{R}^n, \beta \in \mathbb{R}^m. \tag{10}$$

Let $\alpha^* = (\alpha_1^*, \ldots, \alpha_n^*)^{\mathrm{T}}$ and $\beta^* = (\beta_1^*, \ldots, \beta_m^*)^{\mathrm{T}}$ be one optimal solution to (10), and then the Sinkhorn transport plan $T^*$ can be recovered as $T^* = \mathrm{e}_\lambda[\alpha^* \oplus \beta^* - M]$. Remarkably, (10) is equivalent to an *unconstrained* convex optimization problem, so a simple gradient ascent method suffices to find its optimal solution. But in practice, quasi-Newton methods such as the limited memory BFGS method (L-BFGS, Liu & Nocedal, 1989) can significantly accelerate the convergence.

It is worth noting that we can reduce the number of variables to be optimized in (10) based on the following two findings. First, as pointed out by Cuturi et al. (2019), the variables $(\alpha, \beta)$ have one redundant degree of freedom: if $(\alpha^*, \beta^*)$ is one solution to (10), then so is $(\alpha^* + c\mathbf{1}_n, \beta^* - c\mathbf{1}_m)$ for any $c$. Therefore, we globally set $\beta_m = 0$ without loss of generality. Second, let $\alpha^*(\beta) = \arg\max_\alpha \mathcal{L}(\alpha, \beta)$ for a given $\beta = (\tilde{\beta}^{\mathrm{T}}, \beta_m)^{\mathrm{T}} = (\tilde{\beta}^{\mathrm{T}}, 0)^{\mathrm{T}}$, and define $f(\beta) = -\mathcal{L}(\alpha^*(\beta), \beta)$. Then we only need to minimize $f(\beta)$ with $(m-1)$ free variables to get $\beta^*$, and $\alpha^*$ can be recovered as $\alpha^* = \alpha^*(\beta^*)$. As a result, we have the following lemma, with proof given in Appendix C.2:

**Lemma 1.** *The dual problem (10) is equivalent to* $\min_\beta f(\beta)$, *where* $\alpha^*(\beta)$, $f(\beta)$, *and* $\nabla_{\tilde{\beta}} f$ *have the following simple closed-form expressions:*

$$\alpha^*(\beta)_i = \lambda^{-1} \log a_i - \lambda^{-1} \log \left[ \sum_{j=1}^m e^{\lambda(\beta_j - M_{ij})} \right], \tag{11}$$

$$f(\beta) = -\alpha^*(\beta)^{\mathrm{T}} a - \beta^{\mathrm{T}} b + \lambda^{-1}, \tag{12}$$

$$\nabla_{\tilde{\beta}} f = \tilde{T}(\beta)^{\mathrm{T}} \mathbf{1}_n - \tilde{b}. \tag{13}$$

*Then for a specific* $\beta$, *we can recover the transport plan* $T(\beta)$ *by:*

$$T(\beta) = \mathrm{e}_\lambda[\alpha^*(\beta) \oplus \beta - M]. \tag{14}$$

With $f(\beta)$ and $\nabla_{\tilde{\beta}} f$, the L-BFGS algorithm can be readily used to minimize $f(\beta)$ and obtain $\beta^*$. Each gradient evaluation requires $\mathcal{O}(mn)$ exponentiation operations, which is comparable to Sinkhorn-log. Although exponentiation is more expensive than matrix-vector multiplication as in Sinkhorn's algorithm, this extra cost can be greatly remedied by modern hardware such as GPUs. On the other hand, we would show in Section 5 that L-BFGS has a strong guarantee on numerical stability, which is critical for many scientific computing problems.

Using L-BFGS to solve (10) is a known practice (Cuturi & Peyré, 2018; Flamary et al., 2021), but little is known about its stability in solving the regularized OT problem. For the motivating example in Section 4.1, we demonstrate the advantage of L-BFGS by showing its transport plan in the rightmost plot of Figure 1. We limit its maximum number of gradient evaluations to 1000, and hence comparable to other methods. Clearly, the L-BFGS solution is visually identical to the ground truth.

To study the difference between L-BFGS and Sinkhorn's algorithm in more depth, we compute the objective function value of the dual problem (10) at each iteration for both Sinkhorn-log and L-BFGS. The results are visualized in Figure 2, with three different values of $\lambda^{-1}$, $\lambda^{-1} = 0.1, 0.01, 0.001$. Figure 2 gives a clear clue to the issue of Sinkhorn-log: it has a surprisingly slow convergence speed compared to L-BFGS when $\lambda^{-1}$ is small. Theoretically, Sinkhorn algorithms will eventually converge with sufficient iterations, but in practice, a moderate limit on computational budget may prevent them from generating accurate results. To this end, L-BFGS appears to be a better candidate when one needs a small $\lambda^{-1}$ for better approximation to the OT problem. In Appendix B.2 We have designed more experiments to study the stability and accuracy of the Sinkhorn loss computation across different algorithms.

### 4.3 The Analytic Differentiation of Sinkhorn Loss

For the differentiation stage, one commonly-used method is *unrolled* Sinkhorn's algorithm, which is based on the fact that Sinkhorn's computation algorithm is differentiable with respect to $a$, $b$, and $M$. Therefore, one can use automatic differentiation software to compute the corresponding derivatives in the differentiation stage. This method is used in Genevay et al. (2018) for learning generative models with the Sinkhorn loss,

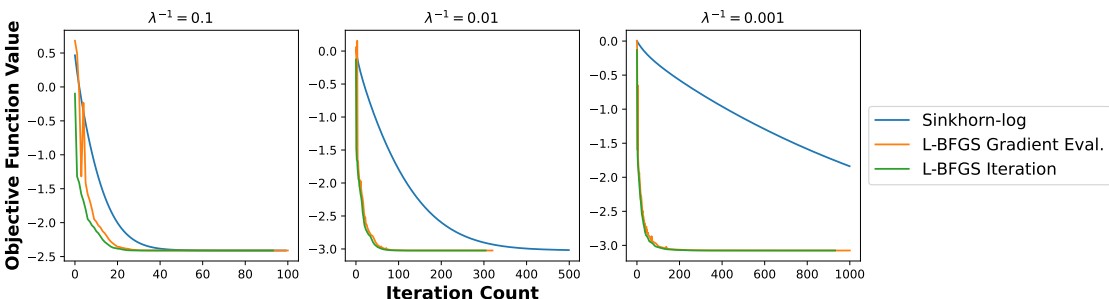

Figure 2: Comparing the convergence speed of Sinkhorn-log and L-BFGS.

*Remark: Sinkhorn-log converges slowly compared to L-BFGS, especially when the regularization parameter $\lambda^{-1}$ of the Sinkhorn loss is small. Since each L-BFGS iteration may involve more than one gradient evaluation, for L-BFGS we plot the values against both the outer iteration and gradient evaluation counts.*

but in practice it may be extremely slow if the solution stage takes a large number of iterations. To avoid the excessive cost of unrolled algorithms, various implicit differentiation methods have been developed for the Sinkhorn loss (Feydy et al., 2019; Campbell et al., 2020; Xie et al., 2020; Eisenberger et al., 2022), but they still do not provide the most straightforward way to compute the gradient.

To this end, we advocate the analytical differentiation of the Sinkhorn loss, which uses the optimal dual variables $(\alpha^*, \beta^*)$ from the solution stage to directly compute the derivative, as presented in the differentiation stage in Algorithm 2. The analytic form for $\nabla_{a,b} S_\lambda(\mu, \nu)$ has been studied in Luise et al. (2018), and to our best knowledge, few result has been presented for $\nabla_M S_\lambda(\mu, \nu)$. Our first main theorem, given in Theorem 4.1, fills this gap and derives the analytic form for $\nabla_M S_\lambda(\mu, \nu)$.

**Theorem 4.1.** *For a fixed $\lambda > 0$,*

$$\nabla_M S_\lambda(\mu, \nu) = T^* + \lambda(s_u \oplus s_v - M) \odot T^*, \tag{15}$$

*where $T^* = e_\lambda[\alpha^* \oplus \beta^* - M]$, $s_u = a^{-1} \odot (\mu_r - \tilde{T}^* \tilde{s}_v)$, $s_v = (\tilde{s}_v^\mathrm{T}, 0)^\mathrm{T}$, $\mu_r = (M \odot T^*)\mathbf{1}_m$, $\tilde{\mu}_c = (\tilde{M} \odot \tilde{T}^*)^\mathrm{T} \mathbf{1}_n$, $\tilde{s}_v = D^{-1}\left[\tilde{\mu}_c - \tilde{T}^{*\mathrm{T}}(a^{-1} \odot \mu_r)\right]$, and $D = \mathbf{diag}(\tilde{b}) - \tilde{T}^{*\mathrm{T}}\mathbf{diag}(a^{-1})\tilde{T}^*$. In addition, $D$ is positive definite, and hence invertible, for any $\lambda > 0$, $a \in \Delta^n$, $b \in \Delta^m$, and $M$.*

Assuming $n \geq m$, the main computational cost is in forming $D$, which requires $\mathcal{O}(m^2 n)$ operations for matrix-matrix multiplication, and in computing $\tilde{s}_v$, which costs $\mathcal{O}(m^3)$ operations for solving a positive definite linear system. Theorem 4.1 provides a simple and efficient way to compute the gradient of the Sinkhorn loss with respect to the cost matrix $M$.

# 5  Stability Analysis and Convergence

With the computation and differentiation algorithms advocated for the Sinkhorn loss in Algorithm 2, we are able to find the Wasserstain coreset using Algorithm 1 efficiently. In this section, we focus on the theoretical properties of these algorithms, and show that they enjoy provable efficiency and stability.

## 5.1  Stability Analysis of Algorithm 2

As a first step, we consider the objective function (12), and show that $f(\beta)$ has a well-behaved minimizer, which does not underflow or overflow.

**Theorem 5.1.** *Let $f^*$ denote the minimum value of $f(\beta)$ and $\beta^*$ an optimal solution, and let $\alpha^* = \alpha^*(\beta^*)$. Then $f^* > -\infty$, $\beta^*$ is unique, $\|\alpha^*\| < \infty$, and $\|\beta^*\| < \infty$. In particular, let $I = \arg\max_i T^*_{im}$, $a_{\max} = \max_i a_i$, and $c = \log(n/b_m)$. Then $\underline{L}_{\alpha_i} \leq L_{\alpha_i} \leq \alpha^*_i \leq U_{\alpha_i}$ and $L_{\beta_j} \leq \beta^*_j \leq U_{\beta_j} \leq \overline{U}_{\beta_j}$ for all $i = 1, \ldots, n$*

*and $j = 1, \ldots, m$, where*

$$U_{\alpha_i} = M_{im} + \lambda^{-1} \cdot \log(a_i \wedge b_m), \qquad U_{\beta_j} = M_{Ij} - M_{Im} + \lambda^{-1}\left[\log(a_I \wedge b_j) + c\right],$$

$$L_{\alpha_i} = \lambda^{-1} \cdot \log(a_i/m) - \max_j (U_{\beta_j} - M_{ij}), \qquad L_{\beta_j} = \lambda^{-1} \cdot \log(b_j/n) - \max_i (U_{\alpha_i} - M_{ij}),$$

$$\overline{U}_{\beta_j} = \max_i\{M_{ij} - M_{im}\} + \lambda^{-1}\left[\log(a_{\max} \wedge b_j) + c\right], \qquad \underline{L}_{\alpha_i} = \lambda^{-1} \cdot \log(a_i/m) - \max_j (\overline{U}_{\beta_j} - M_{ij}).$$

Theorem 5.1 shows that the optimal dual variables $\gamma^*$ are bounded, and more importantly, the bounds are roughly at the scale of the cost matrix entries $M_{ij}$ and the log-weights $\log(a_i)$ and $\log(b_j)$. Therefore, at least the target of the optimization problem is well-behaved. Moreover, one useful application of Theorem 5.1 is to impose a box constraint on the variables, adding further stability to the optimization algorithm.

After verifying the properties of the optimal solution, a more interesting and critical problem is to seek a stable algorithm to approach the optimal solution. Indeed, in Theorem 5.2 we prove that the solution stage of Algorithm 2 is one such method.

**Theorem 5.2.** *Let $\{\tilde{\beta}^{(k)}\}$ be a sequence of iterates generated by the L-BFGS algorithm starting from a fixed initial value $\tilde{\beta}^{(0)}$, and define $\beta^{(k)} = (\tilde{\beta}^{(k)\mathrm{T}}, 0)^{\mathrm{T}}$, $T^{(k)} = \mathrm{e}_\lambda[\alpha^*(\beta^{(k)}) \oplus \beta^{(k)} - M]$, and $f^{(k)} = f(\beta^{(k)})$. Then there exist constants $0 \leq r < 1$ and $C_1, C_2 > 0$, such that for each $k > 0$, let $\varepsilon^{(k)} := r^k(f^{(0)} - f^*)$:*

*(a)* $f^{(k)} - f^* \leq \varepsilon^{(k)}$.          *(Linear convergence for the objective value)*

*(b)* $\|\beta^{(k)} - \beta^*\|^2 \leq C_1 \varepsilon^{(k)}$.          *(Linear convergence for the iterates)*

*(c)* $T^{(k)}\mathbf{1}_m = a$, $\|\nabla_{\tilde{\beta}} f(\beta^{(k)})\|^2 = \|\tilde{T}^{(k)\mathrm{T}}\mathbf{1}_n - \tilde{b}\|^2 \leq C_2 \varepsilon^{(k)}$.      *(Exponential decay of the gradient)*

*(d)* $T_{ij}^{(k)} < \min\{a_i, b_j + \sqrt{C_2 \varepsilon^{(k)}}\}$, $1 \leq j \leq m-1$.      *($T^{(k)}$ does not overflow)*

*(e)* $\max_j T_{ij}^{(k)} > a_i/m$, $\max_i T_{ij}^{(k)} > (b_j - \sqrt{C_2 \varepsilon^{(k)}})/n$, $1 \leq j \leq m-1$.    *($T^{(k)}$ does not underflow)*

*The explicit expressions for the constants $C_1, C_2, r$ are given in Appendix A.*

Theorem 5.2 reveals some important information. First, both the objective function value and the iterates have linear convergence speed, so the solution stage using L-BFGS takes $\mathcal{O}(\log(1/\varepsilon))$ iterations to obtain an $\varepsilon$-optimal solution. Second, the marginal error for $\mu$, measured by $\|T^{(k)}\mathbf{1}_m - a\|$, is exactly zero due to the partial optimization on $\alpha$. The other marginal error $\|\tilde{T}^{(k)\mathrm{T}}\mathbf{1}_n - \tilde{b}\|$, which is equal to the gradient norm, is also bounded at *any* iteration, and decays exponentially fast to zero. This validates the numerical stability of the L-BFGS algorithm. Third, the estimated transport plan at any iteration $k$, $T^{(k)}$, is also bounded and stable. This result can be compared with the formulation in (8): it is not hard to find that $u^*$, $v^*$, and $M_e$, when computed individually, can all be unstable due to the exponentiation operations, especially when $\lambda^{-1}$ is small. In contrast, $T^*$ and $T^{(k)}$, thanks to the results in Theorem 5.2, do not suffer from this issue.

We emphasize that Theorem 5.2 provides novel results that are not direct consequences of the L-BFGS convergence theorem given in Liu & Nocedal (1989). First, classical theorems only guarantee the convergence of objective function values and iterates as in (a) and (b), whereas we provide richer information such as the marginal errors and transport plans specific to OT problems. More importantly, our results are all nonasymptotic with computable constants. To achieve this, we carefully analyze the eigenvalue structure of the dual Hessian matrix, which is of interest by itself.

Likewise, we show that the derivative of $\nabla_M S_\lambda(\mu, \nu)$ as in Theorem 4.1 can also be computed in a numerically stable way. Let $\widehat{\nabla_M S}$ be the $k$-step approximation to $\nabla_M S := \nabla_M S_\lambda(\mu, \nu)$ using the L-BFGS algorithm, *i.e.*, replacing every $T^*$ in $\nabla_M S$ by $T^{(k)}$. Then we show that the error on gradient also decays exponentially fast.

**Theorem 5.3.** *Using the symbols defined in Theorems 4.1 and 5.2, let $\sigma = 1/\sigma_{\min}(D)$, where $\sigma_{\min}(D)$ is the smallest eigenvalue of $D$. Assume that for some $k_0$,*

$$\varepsilon^{(k_0)} < C_1^{-1}\left[\frac{\min\{1, (6\sigma\|D\|_F)^{-1}\}}{4\lambda}\right]^2,$$

*and then for every $k \geq k_0$, $\|\widehat{\nabla_M S} - \nabla_M S\|_F \leq C_S \sqrt{\varepsilon^{(k)}} = C_S \sqrt{f^{(0)} - f^*} \cdot r^{k/2}$, where the explicit expression for $C_S$ is given in Appendix A. $k_0$ always exists as $\varepsilon^{(k)}$ decays to zero exponentially fast, which is ensured by Theorem 5.2(a).*

## 5.2 Consistency of Wasserstein Coreset

We examine the distributional convergence of the empirical measure induced by the Wasserstein coreset to the target distribution $\mu$.

**Theorem 5.4.** *Let $\boldsymbol{X} \sim \mu$, $\boldsymbol{X}_B \sim \mu_B$, where $\mu_B$ is the empirical distribution of Wasserstein coresets, which is defined in (5). Then as $B \to \infty$, $\boldsymbol{X}_B \xrightarrow{d} \boldsymbol{X}$.*

To summarize, the theorem shows that the Wasserstein coreset is indeed representative of the target distribution, when the number of points $B$ grows large. Consequently, this result implies the *consistency* of the Wasserstein coreset, a critical property for any coreset algorithm.

**Corollary 5.5.** *Let $\boldsymbol{X} \sim \mu$, $\boldsymbol{X}_B \sim \mu_B$, with $\mu_B$ defined in (5). (a) If $g : \mathcal{X} \to \mathbb{R}$ is continuous, then $g(\boldsymbol{X}_B) \xrightarrow{d} g(\boldsymbol{X})$. (b) If $g$ is continuous and bounded, then $\mathbb{E}[g(\boldsymbol{X}_B)] \to \mathbb{E}[g(\boldsymbol{X})]$.*

Part (a) of Corollary 5.5 follows from the continuous mapping theorem and Theorem 5.4, and part (b) holds due to the Portmanteau theorem. The corollary establishes the consistency of the Wasserstein coreset for integration, thus validating its use in a range of applications. Specifically, part (a) indicates that the Wasserstein coreset can be effectively employed for uncertainty propagation in stochastic simulations. Part (b) further demonstrates that any continuous and bounded function $g$ can be accurately estimated using the Wasserstein coreset, as the sample average of $g(\mathbf{X}_B)$ converges to the expectation of $g(\mathbf{X})$.

## 5.3 WCSL Algorithm Convergence Analysis

Without assuming convexity, as convexity may not hold in practical scenarios, our aim is to demonstrate the convergence of WCSL to a stationary point and establish a robust convergence criterion. First, we demonstrate that $S_\lambda(\xi, \mathcal{D})$ converges to a stationary point in Theorem 5.6. We only provide the convergence guarantee when using the Adam update, as Adam is a better performed optimization algorithm in practice. We then provide a convergence criterion for the WCSL algorithm in Proposition 5.7.

**Theorem 5.6.** *Using the Adam update, after $N$ iterations for some $N \in \mathbb{N}^*$ such that $N > \frac{\beta_1}{1-\beta_1}$, where $\beta_1$ is the first moment decay rate in Adam, the Sinkhorn loss $S_\lambda(\xi, \mathcal{D}^{(N)})$ converges to a stationary point. Specifically, there exists a constant $C > 0$ such that*

$$\mathbb{E}\left[\left\|\nabla_{\mathcal{D}^{(N)}} S_\lambda(\xi, \mathcal{D}^{(N)})\right\|^2\right] \leq C \, \frac{\log N}{\sqrt{N}}.$$

Theorem 5.6 establishes the theoretical convergence property of WCSL after a finite number of iterations. Now we need a practical method for monitoring the convergence of the WCSL algorithm and ensuring that the algorithm is progressing towards a stationary point of the Sinkhorn loss.

As mentioned in the WCSL algorithm (Algorithm 1), the stopping criterion is $S_\lambda(\mathcal{D}^{(l)}, \mathcal{D}^{(l-1)}) < \delta$. We now provide justifications for this stopping criterion to be a proper convergence criterion.

**Proposition 5.7.** *Let $\xi$ be the target distribution, and $\{\mathcal{D}^{(l)}\}_{l \geq 0}$ be a sequence of representative samples updated by Algorithm 1. If $S_\lambda(\mathcal{D}^{(l)}, \mathcal{D}^{(l-1)}) < \delta$, then $|S_\lambda(\xi, \mathcal{D}^{(l)}) - S_\lambda(\xi, \mathcal{D}^{(l-1)})| < \delta$.*

The convergence of the WCSL algorithm can be characterized by $|S_\lambda(\xi, \mathcal{D}^{(l)}) - S_\lambda(\xi, \mathcal{D}^{(l-1)})| < \delta$, for a chosen $\delta > 0$. This criterion implies that the difference in the objective function, specifically the Sinkhorn loss between successive iterations, is sufficiently small, indicating that the algorithm is approaching a stationary point. Given Proposition 5.7, we know that $S_\lambda(\mathcal{D}^{(l)}, \mathcal{D}^{(l-1)})$ serves as an upper bound for $|S_\lambda(\xi, \mathcal{D}^{(l)}) - S_\lambda(\xi, \mathcal{D}^{(l-1)})|$. Thus, we can ascertain that the algorithm is progressing towards a stationary point of $S_\lambda(\xi, \mathcal{D})$ if $S_\lambda(\mathcal{D}^{(l)}, \mathcal{D}^{(l-1)})$, which can be computed efficiently, is small. This provides a practical method for monitoring the convergence of the WCSL algorithm and ensuring that the iterations yield diminishing improvements, thus verifying the convergence of the algorithm.

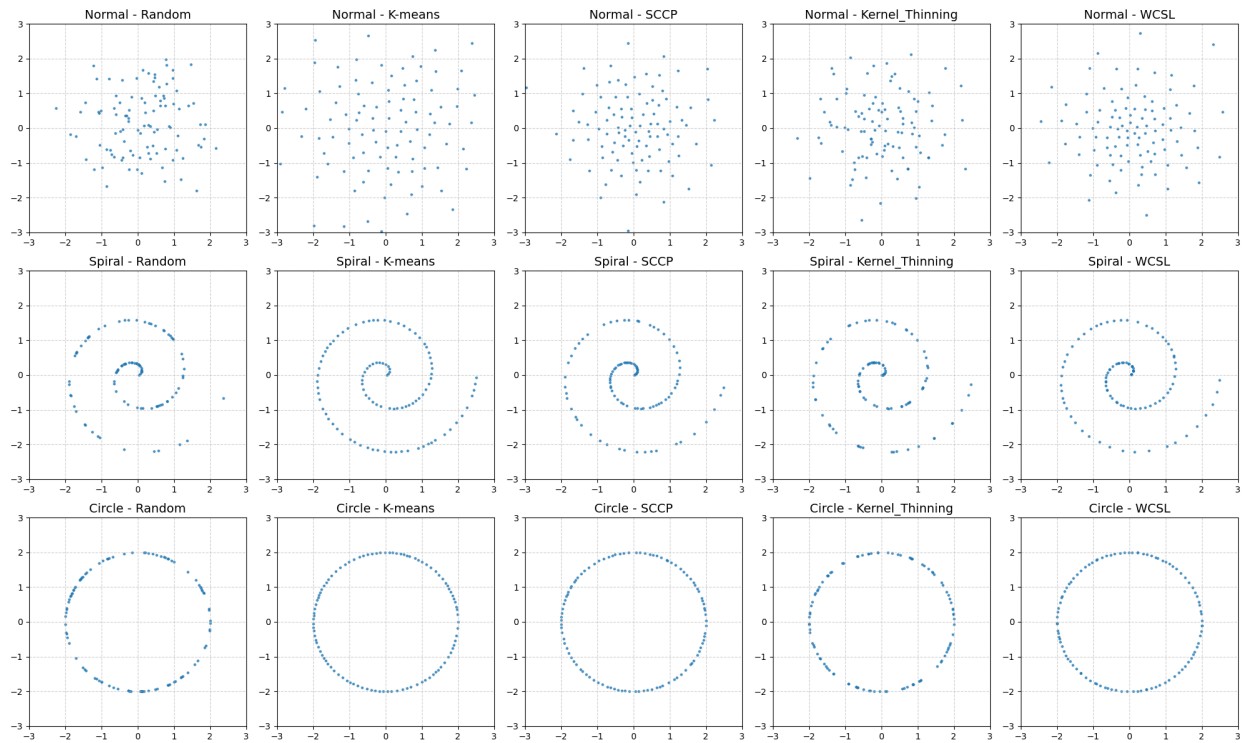

Figure 3: Comparative visualization of samples selected from 2-dimensional datasets.

# 6 Numerical Experiments

## 6.1 Simulations

Several simulations are presented to demonstrate the effectiveness and efficiency of the WCSL method. We evaluate the sample selection quality of the WCSL method visually and in terms of distance metrics. We also performed a runtime analysis for the WCSL method. The results provide insights into the practical applicability of the WCSL method in various data settings.

To evaluate the performance of the WCSL method, we compared it against several alternative sampling techniques: random sampling, the centroids obtained via k-means clustering, the SCCP method (Mak & Joseph, 2018), and kernel thinning (Dwivedi & Mackey, 2021). Random sampling, which selects samples uniformly at random from the dataset, serves as a straightforward benchmark. The k-means centroids approach involves partitioning the dataset into clusters and then using the resulting cluster centroids as representative points. The SCCP method compresses a continuous probability distribution into a finite set of support points that minimize the energy distance. This approach guarantees that the support points converge in distribution to the original distribution and often yields improved integration error rates for a broad class of functions compared to standard Monte Carlo methods. Finally, the kernel thinning method selects representative points by minimizing the maximum mean discrepancy (MMD), thereby providing another kernel-based strategy for dataset compression.

To visually show the sample selection quality difference between random samples and coreset techiniques, we present a comparative visualization of subsamples selected from synthetic 2-dimensional datasets. The experiment settings include datasets of $n = 1000$ points each, reduced to coresets of $B = 100$ points. The three configurations tested are a normal distribution, a 2-dimensional spiral, and a circular distribution. Figure 3 illustrates the results of coreset techniques. All the coreset techniques demonstrate superior performance in capturing the original data distributions compared to random sampling. Random sampling is not consistent and often fails to accurately represent the underlying distribution, highlighting the necessity

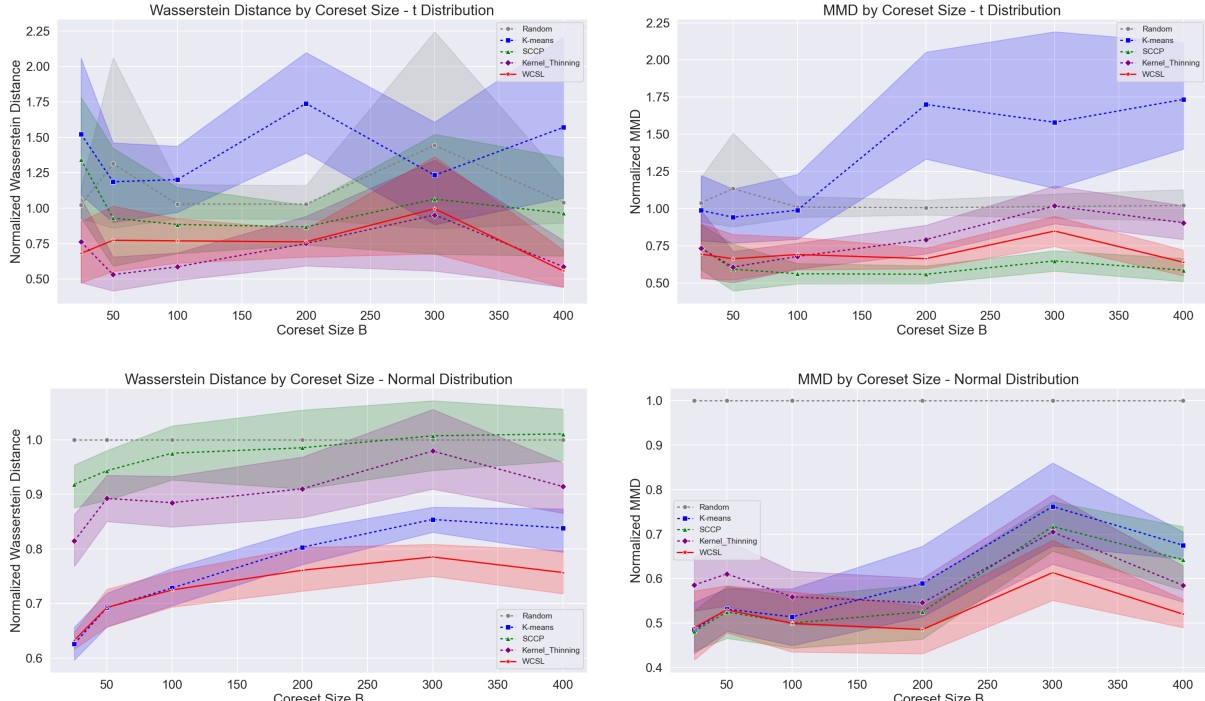

Figure 4: Comparison of different sample selection techniques across various datasets. Lower is better.

of using coreset techniques. Among the methods tested, the WCSL method performed well, effectively preserving the essential data characteristics in the reduced sample sets. This visual assessment underscores the potential of the WCSL method in maintaining the integrity of the original data structure in various data reduction scenarios.

However, visual comparisons alone are insufficient to rigorously determine which coreset method performs better. Therefore, we compute both the Wasserstein distance and MMD between the full dataset and the coreset obtained using various techniques. We use a 100-dimensional $t$-distribution and a 100-dimentional normal distribution with $n = 10000$ points as our datasets, and reduce to coresets of various sizes, ranging from $B = 25$ to $B = 400$. In our analysis, we seek to ensure meaningful comparisons across different sample selection techniques. Consequently, we normalize the computed Wasserstein distance and MMD values by the distance metrics between the randomly sampled subsets and the full dataset. This normalization procedure aligns the distance scales, allowing direct comparisons to be made between the various coreset construction methods.

The results in Figure 4 demonstrate that the coresets generated by WCSL consistently achieve low normalized MMD and Wasserstein distance values. This indicates that WCSL more effectively preserves the original data distribution and structure. In contrast, random sampling leads to persistently high distance metrics, underscoring its limited ability to capture the underlying distribution. While the k-means centroids approach sometimes outperforms random sampling, it displays instability and can yield results even worse than random sampling in certain instances. The SCCP and kernel thinning methods perform moderately well, but their effectiveness varies across different metrics and scenarios. Overall, these findings suggest that WCSL provides a more robust and reliable representation of the original data, ensuring that its essential statistical properties are well-maintained.

Before evaluating the computational efficiency of the WCSL method empirically, we first present its time complexity analysis. The WCSL algorithm involves two layers of optimization: the inner SLCD problem and the outer coreset construction. To analyze the SLCD problem, we decompose it into a solution stage

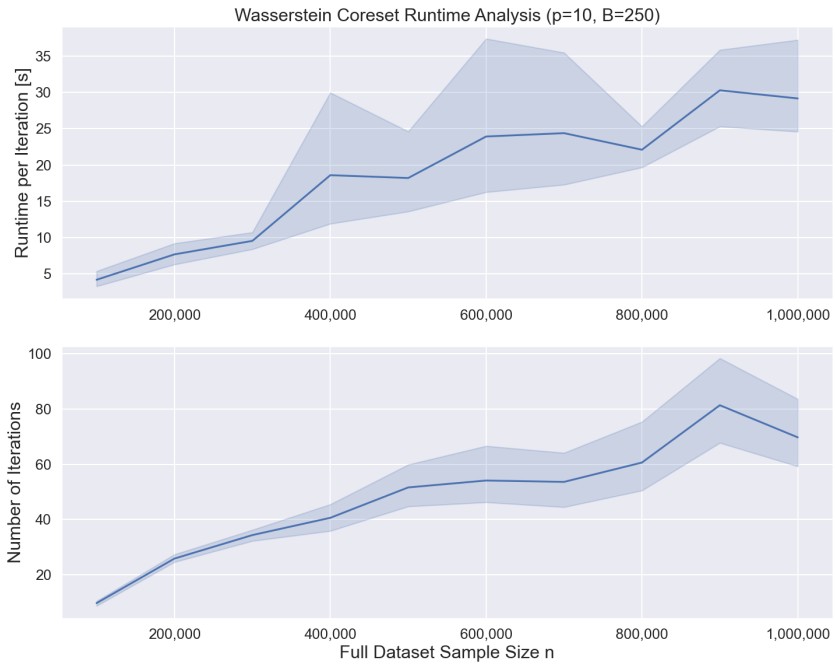

Figure 5: Runtime analysis of the WCSL method with respect to full dataset size $n$.

and a differentiation stage. In each SLCD step, the solution stage involves computing the distance matrix, which requires $O(m^2p)$ time, where $m$ represents the batch size and $p$ denotes the data dimension. Each iteration of the L-BFGS optimization procedure has a complexity of $O(m^2)$, and the algorithm typically requires $O(\log(1/\epsilon))$ iterations to converge, where $\epsilon$ denotes the convergence tolerance. In the differentiation stage, computing the derivative incurs an additional computational cost of $O(m^3)$. As a result, the overall time complexity of each SLCD step is $O(m^2(p+\log(1/\epsilon)+m))$. Since $m \ll n$, where $n$ is the full dataset size, we treat $m$ as negligible. Consequently, because each iteration processes the entire dataset in mini-batches, the time complexity of each WCSL iteration simplifies to $O(np)$.

Multiple iterations are typically needed for WCSL to converge, and the number of these iterations depends on the chosen stopping criterion. Based on our empirical observations, for a fixed stopping criterion, we assume that $O(n)$ iterations are required. Under this assumption, the overall time complexity is $O(n^2p)$, which remains tractable for large-scale data scenarios.

To further demonstrate the computational efficiency of the method, we measure the execution times for a range of dataset sizes $n$ while fixing the data dimension $p = 10$ and coreset size $B = 250$. We vary $n$ from 100,000 to 1,000,000, and record the per-iteration runtime. Figure 5 shows that the runtime per iteration increases proportionally with $n$, reflecting the anticipated linear scaling in computational demand. Moreover, the number of required iterations also grows roughly linearly with $n$, underscoring the suitability of the WCSL method for large-scale data applications. Together, these results confirm that WCSL provides an efficient and practical solution for large-scale problems.

We also provide the running time of the WCSL with respect to coreset size $B$ and data dimension $p$ in Figure 6. The running time is measured with the coreset size varied from $B = 100$ to $B = 1000$ given sample size $n = 10000$ and data dimension $p = 25$. The running time is also measured with the data dimension varying from $p = 10$ to $p = 100$ given the sample size $n = 10000$ and the coreset size $B = 250$. The running time per iteration increases linearly with respect to the coreset size B and stays almost flat with respect to the data dimension $p$, while the number of iterations required for convergence decreases as the coreset size B increases and increase as the data dimension p increases. These findings highlight the trade-offs between

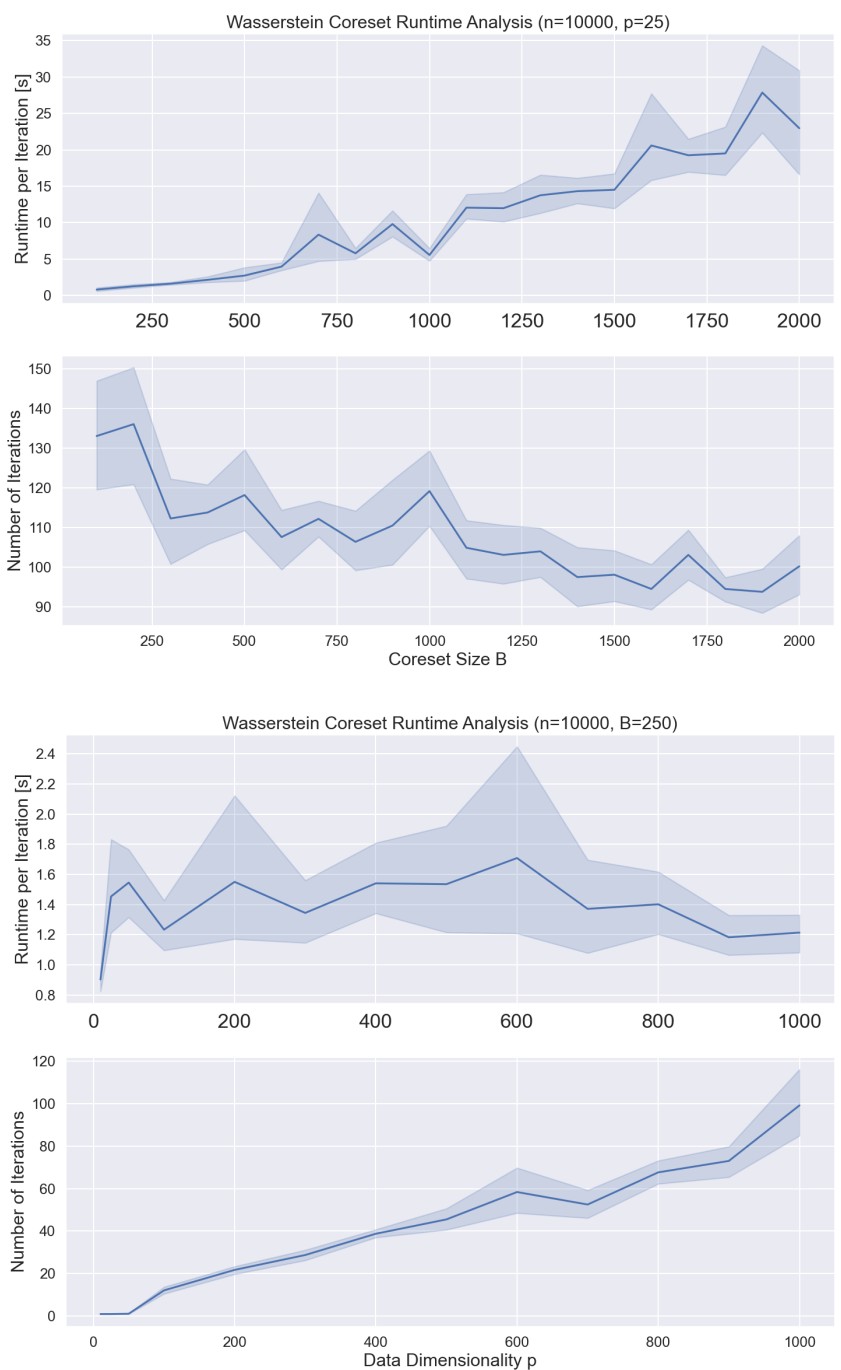

Figure 6: Runtime analysis of WCSL with respect to coreset size $B$ and data dimension $p$.

coreset size and computational efficiency, providing valuable insights for optimizing the WCSL method for large-scale data applications.

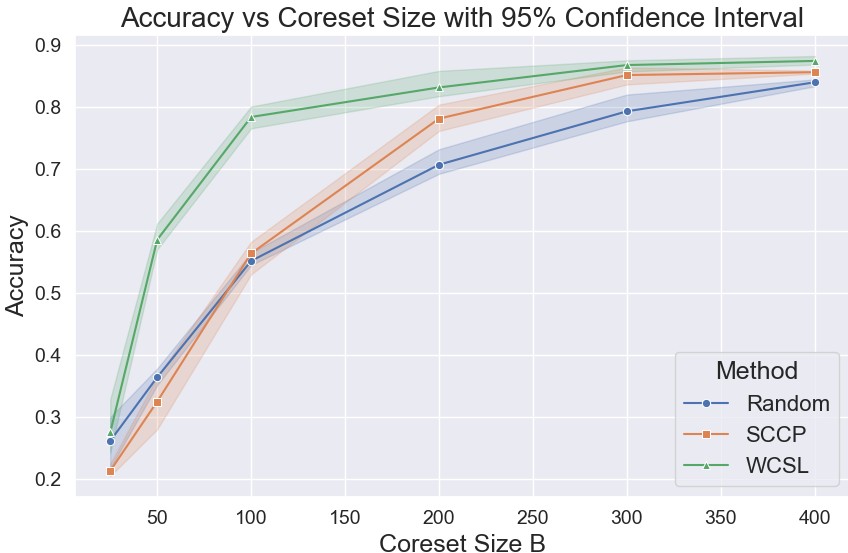

Figure 7: Classification accuracy across various coreset sizes, showcasing the performance of prediction models trained on subsets selected via different methods.

## 6.2  Active Learning

One of the potential applications of the Wasserstein coreset is in the field of active learning. In active learning, it involves the data points $X \in \mathcal{X}$ and their corresponding labels $Y \in \mathcal{Y} = \{1, \ldots, C\}$, with $C$ being the number of label categories. A labeling function $\Omega : \mathcal{X} \to \mathcal{Y}$ determines the accurate label for any given sample in active learning scenarios. Our primary goal is to minimize the expected risk in a given data distribution with a parameterized loss function $\ell(X, Y; w)$. This objective is formally expressed as $\min_w \mathbb{E}[\ell(X, Y; w)]$. Ideally, we have access to a labeled dataset of $N$ labeled pairs $\{(X_i, Y_i)\}_{i=1}^N$ to reduce the expected risk.

However, in practice, we have an unlabeled dataset of $N$ observations $\{X_i\}_{i=1}^N$ and a budget to get $\Omega(X_i)$, the labels of $X_i$, for $B$ times, where $B \ll N$. Building on the work of Sener & Savarese (2018), we recognize that the expected risk is bounded by the summation of three factors: the generalization error from training with fully labeled dataset instead of the true population measure, the empirical risk from training on a selected subset of $B$ points, and the discrepancy in loss between training on the entire dataset versus the selected subset, the subset loss. The mathematical representation of this constraint is:

$$\mathbb{E}[\ell(X, Y; w)] \leq \underbrace{\left| \mathbb{E}[\ell(X, Y; w)] - \frac{1}{N} \sum_{i=1}^N \ell(X_i, \Omega(X_i); w) \right|}_{\text{generalization error}} + \underbrace{\frac{1}{B} \sum_{j=1}^B \ell(X_j, \Omega(X_j); w)}_{\text{empirical risk}}$$

$$+ \underbrace{\frac{1}{N} \sum_{i=1}^N \ell(X_i, \Omega(X_i); w) - \frac{1}{B} \sum_{j=1}^B \ell(X_j, \Omega(X_j); w)}_{\text{subset loss}}.$$

The generalization error remains unaffected by the chosen subset, and when $B$ is small, the empirical risk is typically negligibly small. This suggests that active learning strategies should prioritize minimizing the loss associated with the chosen subset.

## MNIST and FashionMNIST Coresets

Figure 8: Comparative visualization of coreset samples on MNIST and FashionMNIST datasets.

Notice that if $\ell \in \mathcal{F}$, then the subset loss would be bounded by the coreset loss:

$$\frac{1}{N}\sum_{i=1}^{N}\ell(\mathbf{x}_i, \Omega(\mathbf{x}i); \mathbf{w}) - \frac{1}{B}\sum_{j=1}^{B}\ell(\mathbf{x}_j, \Omega(\mathbf{x}j); \mathbf{w}) \leq d_{\mathcal{F}}(\xi, \nu_B)$$

This motivates us to use coreset technique in enhancing the efficiency of choosing subsets in active learning. We consider a family of 500-dimensional $t$-distributions, categorized into 25 classes. Each class is defined by a one-unit shift to the right in a randomly selected dimension of the $t$-distribution. The full dataset comprises 20,000 points. To establish a baseline, we train a logistic regression model on the entire dataset and use this model to relabel the dataset, treating these labels as the ground truth for subsequent classification tasks. Subsequently, we generate coresets using random selection, SCCP, and WCSL method. Logistic regression models are then trained on these coresets of sizes from 25 to 400, with their classification accuracies evaluated thereafter.

Given the complexity of the dataset with 25 classes, achieving high classification accuracy with small sample sizes is challenging. Especially for coreset sizes of 25 or 50, it is likely that some classes will not be represented in the coresets unless an effective coreset selection technique is used. This can lead to high classification error rates. Despite this challenge, Figure 7 shows that models trained on Wasserstein coreset-selected samples demonstrated superior performance compared to those trained on coresets generated by random sampling or SCCP, particularly with smaller coreset sizes. This improvement in classification accuracy underscores the ability of the Wasserstein coreset approach to capture the underlying distribution of the full dataset effectively. Consequently, models trained using Wasserstein-selected coresets exhibit enhanced classification performance, validating the strategic advantage of this method in active learning contexts.

### 6.3 MNIST & FashionMNIST

To further demonstrate the practical applicability of the WCSL method, we applied it to image datasets, the MNIST and FashionMNIST datasets. Representative samples generated by the WCSL method are depicted in Figure 8, demonstrating the ability of the method to retain characteristic data features in the reduced sample set.

We also evaluate the performance of the WCSL method in terms of distance metrics, comparing it with other subsampling techniques. The kernel thinning method proves to be very slow on image datasets; thus, we do not include its results. The results, presented in Figure 9, indicate that the WCSL method consistently

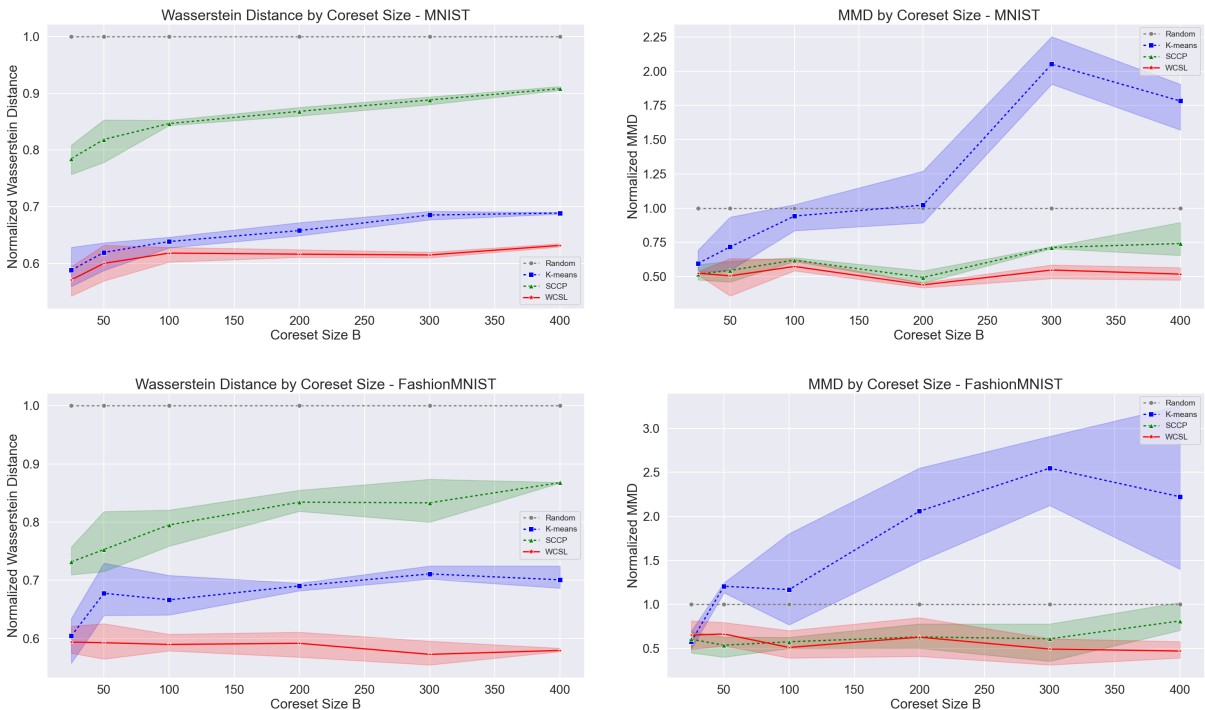

Figure 9: Comparative visualization of Wasserstein distance and MMD by coreset techniques and size. Lower is better.

exhibits lower Wasserstein distances and reduced MMD values compared to random subsampling. This affirms the method's superior capability in approximating the original data distribution, ensuring that the reduced samples are highly representative of the full dataset.

# 7 Discussion

In conclusion, this paper has introduced a novel and robust Wasserstein coreset construction method that utilizes the entropic-regularized framework of the Sinkhorn loss to effectively approximate large datasets while preserving their intrinsic geometric structure. Through rigorous analysis and extensive experimentation, we have demonstrated the stability and efficiency of our computation and differentiation algorithms in settings that have traditionally posed numerical challenges. Our findings reveal that Wasserstein coresets yield superior performance in model accuracy, particularly under budget constraints where label scarcity is an issue in active learning scenarios. This work not only advances the methodology of data distillation, but also offers tangible benefits for active learning and other machine learning tasks that require fast, accurate, and cost-effective data approximation techniques. The success of the WCSL method in various practical scenarios, including imaging data, confirms its potential as a valuable tool in big data analytics, affirming its role in enhancing the scalability and accessibility of machine learning applications.

In light of these advances, it is pertinent to reflect on two specific aspects of the coreset methodology within the broader context of data representation and active learning strategies. First, while the coreset is designed to mirror the underlying data distribution, it does not inherently ensure that the samples are drawn from the original dataset. This distinction becomes critical in applications where authenticity is paramount, such as dataset splitting for training and validation in machine learning models, financial audits where transactional integrity is necessary, or medical studies that require exact patient data without generalizations. Ensuring the representativeness of coresets drawn from the original dataset might involve additional constraints in the selection algorithm or post-hoc verification procedures to confirm the origin of the data points. Secondly, the

active learning framework benefits from the Wasserstein coreset's efficiency, particularly in the initial stages of model training. However, as learning progresses, there might be a need for a more explorative approach to sample selection. This could involve adaptively adjusting the coreset as new labels are acquired or integrating uncertainty measurements to guide the selection process. Exploration strategies such as entropy maximization or variance reduction could be particularly beneficial in iterative learning settings, ensuring that each new query contributes maximal information to the learning model. In both cases, the challenge lies in striking a balance between representativeness and practical constraints. Future work may explore hybrid strategies that leverage the Wasserstein coreset's strengths while addressing its limitations in specific application scenarios. The goal is to refine the technique to become a versatile tool that is able to handle a wide array of tasks in the burgeoning field of machine learning.

## Acknowledgements

Xiao Wang's research was supported in part by the NSF Grant SES-2316428. Yixuan Qiu's work was supported in part by National Natural Science Foundation of China (12101389), Shanghai Pujiang Program (21PJC056), MOE Project of Key Research Institute of Humanities and Social Sciences (22JJD110001), and Shanghai Research Center for Data Science and Decision Technology.

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

## Appendix

## A  Explicit expressions for constants

We first define a few user constants for the L-BFGS algorithm. Let $m_0$ be the maximum number of correction vectors used to construct the BFGS matrix $B_k$, and $c_1 \in (0, 1/2)$, $c_2 \in (c_1, 1)$ are two constants related to the Wolfe condition: we assume the L-BFGS algorithm uses some line search algorithm to select the step sizes $\alpha_k$ that satisfy:

$$f(x_k + \alpha_k d_k) \leq f(x_k) + c_1 \alpha_k g_k^{\mathrm{T}} d_k,$$
$$g(x_k + \alpha_k d_k)^{\mathrm{T}} d_k \geq c_2 g_k^{\mathrm{T}} d_k,$$

where $f(\cdot)$ and $g(\cdot)$ stand for the objective function and gradient function, respectively, $x_k$ is the $k$-th iterate, $g_k = g(x_k)$, $d_k = -B_k^{-1} g_k$ is the search direction, and $B_k$ is the BFGS matrix that approximates the Hessian matrix. $m_0$, $c_1$, and $c_2$ are selected by the user.

For Theorem 5.2, let $\tilde{\beta}^{(0)}$ be the initial value, and let $\mu = M^{\mathrm{T}} a$ and $u_i = \max_j M_{ij}$, $i = 1, \ldots, n$. Then we define the following constants:

$$U_c = b_m^{-1} \left[ \left( \max_{1 \leq j \leq m-1} \mu_j \right) + \lambda^{-1} \sum_{i=1}^{n} a_i \log a_i - \lambda^{-1} + f(\beta^{(0)}) \right]_+$$

$$A_i = \lambda^{-1} \log a_i - U_c - \lambda^{-1} \log \left( e^{-\lambda(M_{im} + U_c)} + \sum_{j=1}^{m-1} e^{-\lambda M_{ij}} \right), \quad i = 1, \ldots, n$$

$$M_1 = \lambda \cdot \frac{n - m + 2}{2(n - m + 1)} \cdot \min_{1 \leq i \leq n} e^{\lambda(A_i - M_{im})}, \qquad M_2 = \lambda \left[ 1 - \sum_{i=1}^{n} e^{\lambda(A_i - M_{im})} \right]$$

$$M_3 = M_2 - \log M_1 - 1, \qquad M_4 = m - 1 + m_0 M_2 - m_0 \left[ \log M_1 - \log(1 + m_0 M_2) \right]$$

$$C_1 = 2/M_1, \qquad C_2 = 2 M_1^{-1} M_2^2$$

$$r = 1 - c_1(1 - c_2) M_1 / M_2 e^{-(M_3 + M_4)}.$$

For Theorem 5.3,

$$C_S = 4\lambda \sqrt{C_1} \left[ \|\nabla_M S\|_F + 2\lambda \|T^*\|_F (C_v + C_u) \right]$$
$$C_v = 2\sigma(\|\mu_c\| + 3\|T^{*\mathrm{T}}(a^{-1} \odot \mu_r)\| + 3\|D\|_F \|s_v\|)$$
$$C_u = \|\mu_r\| + 2 C_v \|\mathbf{diag}(a^{-1}) T^*\|_F + \|a^{-1} \odot (T^* s_v)\|.$$

## B  Additional experiment details

### B.1  Settings of the motivating example

Consider a small problem of $n = 90$ and $m = 60$. Let $x_i = 5(i - 1)/(n - 1)$, $i = 1, \ldots, n$ be equally-spaced points on $[0, 5]$, and similarly define $y_j = 5(j - 1)/(m - 1)$, $j = 1, \ldots, m$. The cost matrix is set to $M_{ij} = (x_i - y_j)^2$, and the weights $a$ and $b$ are specified as follows. Let $f_1$ be the density function of an exponential distribution with mean 1, and $f_2$ be the density function of a mixture of two normal distributions, $0.2 \cdot N(1, 0.04) + 0.8 \cdot N(3, 0.25)$. And then we set $\tilde{a}_i = f_1(x_i)$, $\tilde{b}_j = f_2(y_j)$, $a_i = \tilde{a}_i / \sum_{k=1}^{n} \tilde{a}_k$, and $b_j = \tilde{b}_j / \sum_{k=1}^{m} \tilde{b}_k$.

For example showed in figure 1 We fix the regularization parameter $\lambda^{-1}$ to be 0.001. This value is selected such that the resulting Sinkhorn plan $T_\lambda^*$ is visually close to the OT plan $P^*$. The maximum number of iterations is 10000 for Greenkhorn and 1000 for other algorithms. Other d

In Figure 10, we show the Wasserstein and Sinkhorn transport plans under different values of $\lambda^{-1}$. It can be seen that when $\lambda^{-1} \leq 0.001$, $T_\lambda^*$ is visually indistinguishable from $P^*$.

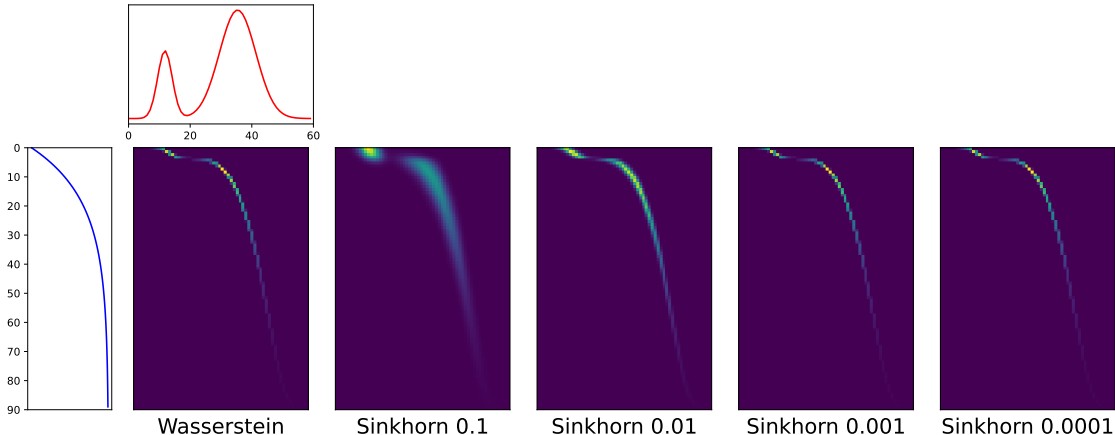

Figure 10: Visualization of Sinkhorn plans with different $\lambda^{-1}$ values.

We compute the true $T_\lambda^*$ using the $\varepsilon$-scaling algorithm (Algorithm 3 of Schmitzer (2019)). This algorithm is typically accurate, but it requires solving a sequence of Sinkhorn problems with increasing $\lambda$'s, where the solution corresponding to the previous $\lambda$ is used as a warm start for the next one. Therefore, its computational cost is typically large, and it does not compare fairly with other methods. Due to this reason, in this article we mainly use the $\varepsilon$-scaling algorithm to compute high-precision reference values, and do not include it for method comparison.

## B.2 Sinkhorn Loss Computation stability and accuracy

To further compare the numerical stability and accuracy of different algorithms for computing the Sinkhorn loss, we consider the following experiment. First, we simulate data $X_1, \dots, X_n \sim f_1(x)$ and $Y_1, \dots, Y_m \sim f_2(y)$ from some specific distributions $f_1(x)$ and $f_2(y)$, $x, y \in \mathbb{R}^p$, and construct the cost matrix as $M = (M_{ij})$, where $M_{ij} = \|X_i - Y_j\|^2$. The weights are fixed as $a = n^{-1}\mathbf{1}_n$ and $b = m^{-1}\mathbf{1}_m$. For each of the five methods compared in the motivating example in Section 4.1, let $T = (T_{ij})$ be the computed Sinkhorn transport plan and $T^*$ be the true value (computed using the $\varepsilon$-scaling algorithm), and then we compute two types of errors: the error on the transport plan,

$$\mathrm{Err}_{plan}(T) = \sqrt{\sum_{i,j}(T_{ij} - T_{ij}^*)^2},$$

and the error on the Sinkhorn loss value,

$$\mathrm{Err}_{loss}(T) = |\langle T, M \rangle - \langle T^*, M \rangle| = |\langle T - T^*, M \rangle|.$$

For each configuration of the experiment, we simulate the data 100 times, and visualize the distribution of the errors using boxplots.

In our experiment, we fix $n = 150$, $m = 200$, and consider varying dimensions $p = 1, 10, 50$. The Sinkhorn regularization parameters compared are $\lambda^{-1} = 0.01, 0.001$, and for each $\lambda^{-1}$ we set a specific maximum number of iterations for all algorithms. Two data generation models are considered:

(a) Both $f_1(x)$ and $f_2(y)$ are multivariate normal distributions $N(0, I_p)$;

(b) Both $f_1(x)$ and $f_2(y)$ have independent components. Each marginal distribution of $f_1$ is an exponential distribution with mean 1, and each marginal distribution of $f_2$ is a mixture of two normal distributions, $0.2 \cdot N(1, 0.04) + 0.8 \cdot N(3, 0.25)$.

The final results are demonstrated in Figure 11, where all the errors are shown in the log-scale.

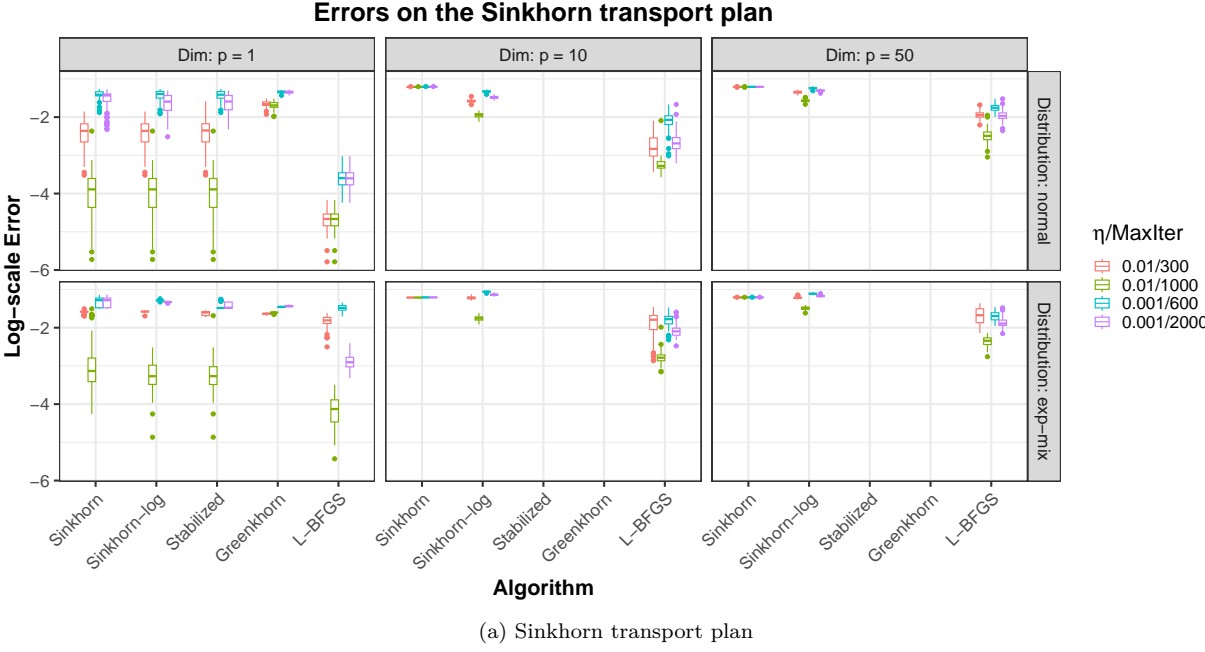

(a) Sinkhorn transport plan

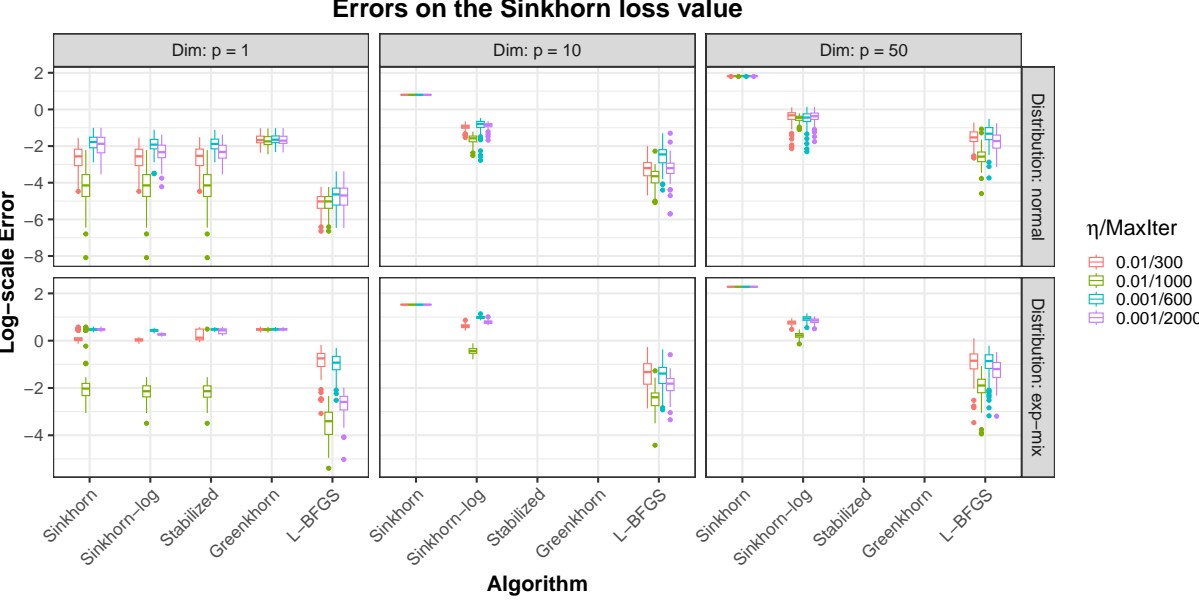

(b) Sinkhorn loss value

Figure 11: Comparing different algorithms on the errors of Sinkhorn transport plan and loss value. The missing boxplots indicate that the corresponding results are NaNs.

In Figure 11, many boxplots for the Stabilized and Greenkhorn algorithms are missing, since they produce all NaN values in the 100 simulations due to numerical overflows. For Sinkhorn, even if it generates no NaNs values explicitly for $\lambda^{-1} = 0.001$, it does not give any meaningful results either. This implies that numerical stability is a critical issue in computing the Sinkhorn loss.

For Sinkhorn-log, it gives reasonably small errors when the regularization is large ($\lambda^{-1} = 0.01$) with sufficient number of iterations, but its accuracy quickly deteriorates when $\lambda^{-1}$ decreases to 0.001. Moreover, we can find that Sinkhorn-log is sensitive to the limits on number of iterations. For example, when $p = 1$ and

$\lambda^{-1} = 0.01$, the loss value error can be as small as $10^{-6}$ given 1000 maximum number of iterations, but if we restrict the limit to 200, the error can be as large as $10^{-2}$ or even $10^0$, depending on the data distribution.

In contrast, the difference on maximum number of iterations has a minor effect on the L-BFGS algorithm, indicating that it converges fast, and additional iterations are not needed. These findings demonstrate the advantage of the advocated L-BFGS method in both numerical stability and accuracy.

### B.3 Running time of Solution Stage and Differentiation Stage

In this section we compare the running time of three algorithms on differentiating the Sinkhorn loss.

(a) The proposed algorithm "Analytic": L-BFGS in the solution stage, and analytic differentiation in the differentiation stage.

(b) "Implicit" implemented in the OTT-JAX library (Cuturi et al., 2022): Sinkhorn-log in the solution stage, and implicit differentiation in the differentiation stage.

(c) "Unroll" implemented in the OTT-JAX library: Sinkhorn-log in the solution stage, and unrolled automatic differentiation in the differentiation stage.

We use the second data generation model in Section B.2 to simulate data points, and use the three algorithms above to compute the Sinkhorn loss and its derivative with respect to the cost matrix. For each configuration, we randomly generate the data 100 times, and compute their mean solving time and mean total time. The stopping rule implemented in the OTT-JAX library is $\|T\mathbf{1}_m - a\| + \|T^{\mathrm{T}}\mathbf{1}_n - b\| < \varepsilon_{ott}$, and one of the terms would be exactly zero in the last iteration. The stopping rule for L-BFGS is $\|T^{\mathrm{T}}\mathbf{1}_n - b\|_\infty < \varepsilon_{lbfgs}$. To account for such a difference, we set $\varepsilon_{lbfgs} = 10^{-6}$, and let $\varepsilon_{ott} = \sqrt{\max\{n, m\}} \cdot \varepsilon_{lbfgs}$. In fact, this setting favors the competing method, as its stopping criterion is strictly weaker than the proposed one. To test whether the algorithms actually converge under the given criteria, we also report the number of converging cases within the 100 repetitions.

Results for different data dimensions and regularization parameters are given in Table 1 and Table 2, where the former uses 1000 maximum number of iterations, and the latter uses 10000.

## C  Proofs of theorems

### C.1  Technical Lemmas

In this section we state a few technical lemmas that are used to prove our main theorems. Lemma 2 to Lemma 4 below are standard conclusions in vector calculus, Lemma 5 and Lemma 6 are derived from the eigenvalue theory, and Lemma 7 and Lemma 8 are related to the Sinkhorn problem.

We first introduce the following notations. For any $x \in \mathbb{R}$, let $[x]_+ = \max\{x, 0\}$. For a matrix $A = (a_{ij}) \in \mathbb{R}^{n \times m}$, the vectorization operator, $\mathbf{vec}(A)$, creates a vector by stacking the column vectors of $A$ together, i.e.,

$$\mathbf{vec}(A) = (a_{11}, \dots, a_{n1}, a_{12}, \dots, a_{n2}, \dots, a_{1m}, \dots, a_{nm})^{\mathrm{T}}.$$

For two matrices $A = (a_{ij}) \in \mathbb{R}^{n \times m}$ and $B \in \mathbb{R}^{p \times q}$, the Kronecker product of $A$ and $B$ is defined as

$$A \otimes B = \begin{pmatrix} a_{11}B & \cdots & a_{1m}B \\ \vdots & \ddots & \vdots \\ a_{n1}B & \cdots & a_{nm}B \end{pmatrix}.$$

For a differentiable vector-valued function $f : \mathbb{R}^n \to \mathbb{R}^m$, the partial derivative of $f$ with respect to $x$ is defined as

$$\frac{\partial f(x)}{\partial x^{\mathrm{T}}} = \begin{pmatrix} \frac{\partial f_1(x)}{\partial x_1} & \cdots & \frac{\partial f_1(x)}{\partial x_n} \\ \vdots & \ddots & \vdots \\ \frac{\partial f_m(x)}{\partial x_1} & \cdots & \frac{\partial f_m(x)}{\partial x_n} \end{pmatrix}.$$

Table 1: Running time of three algorithms for differentiating the Sinkhorn loss, with maximum 1000 iterations.

| | | Mean solving time (ms) | Std. of solving time | Mean total time (ms) | Std. of total time | Converged |
|---|---|---|---|---|---|---|
| $m = n = 64$ $p = 8$ $\lambda^{-1} = 0.1$ | Analytic | 19.07 | 2.11 | **19.33** | 2.13 | 100 |
| | Implicit | 27.77 | 0.69 | 39.66 | 3.41 | 0 |
| | Unroll | 27.53 | 0.13 | 101.30 | 0.36 | 0 |
| $m = n = 64$ $p = 8$ $\lambda^{-1} = 0.01$ | Analytic | 23.21 | 3.45 | **23.48** | 3.46 | 100 |
| | Implicit | 27.33 | 0.05 | 50.52 | 5.19 | 0 |
| | Unroll | 27.39 | 0.12 | 100.66 | 0.39 | 0 |
| $m = n = 128$ $p = 16$ $\lambda^{-1} = 0.1$ | Analytic | 23.79 | 2.96 | **24.19** | 2.96 | 100 |
| | Implicit | 29.81 | 0.61 | 58.21 | 4.91 | 0 |
| | Unroll | 29.75 | 0.06 | 107.18 | 0.19 | 0 |
| $m = n = 128$ $p = 16$ $\lambda^{-1} = 0.01$ | Analytic | 32.24 | 4.99 | **32.68** | 5.00 | 100 |
| | Implicit | 29.73 | 0.05 | 85.56 | 7.48 | 0 |
| | Unroll | 29.65 | 0.08 | 106.66 | 0.46 | 0 |
| $m = n = 256$ $p = 32$ $\lambda^{-1} = 0.1$ | Analytic | 28.60 | 3.82 | **29.40** | 3.85 | 100 |
| | Implicit | 33.69 | 0.32 | 82.53 | 4.21 | 0 |
| | Unroll | 33.60 | 0.23 | 123.25 | 0.07 | 0 |
| $m = n = 256$ $p = 32$ $\lambda^{-1} = 0.01$ | Analytic | 43.94 | 6.08 | **44.86** | 6.12 | 100 |
| | Implicit | 33.75 | 0.20 | 137.46 | 11.66 | 0 |
| | Unroll | 33.78 | 0.10 | 122.99 | 0.27 | 0 |
| $m = n = 512$ $p = 64$ $\lambda^{-1} = 0.1$ | Analytic | 41.35 | 5.64 | **43.05** | 5.53 | 100 |
| | Implicit | 46.06 | 0.05 | 130.96 | 11.97 | 0 |
| | Unroll | 46.02 | 0.04 | 167.64 | 0.22 | 0 |
| $m = n = 512$ $p = 64$ $\lambda^{-1} = 0.01$ | Analytic | 67.76 | 9.65 | **69.42** | 9.65 | 100 |
| | Implicit | 46.08 | 0.05 | 230.34 | 11.72 | 0 |
| | Unroll | 46.09 | 0.05 | 167.68 | 0.20 | 0 |

We use $I_n$ to denote the $n \times n$ identity matrix, and $\sigma_{\max}(\cdot)$ and $\sigma_{\min}(\cdot)$ stand for the largest and smallest eigenvalues of some symmetric matrix, respectively.

**Lemma 2.** *Given two matrices $A \in \mathbb{R}^{m \times n}$ and $B \in \mathbb{R}^{n \times r}$,*

$$\mathbf{vec}(AB) = (I_r \otimes A)\mathbf{vec}(B) = (B^{\mathrm{T}} \otimes I_m)\mathbf{vec}(A).$$

**Lemma 3.** *Let $f : \mathbb{R}^m \to \mathbb{R}^n$ and $g : \mathbb{R}^m \to \mathbb{R}^n$ be two vector-valued differentiable functions of $x \in \mathbb{R}^m$. Then*

$$\frac{\partial}{\partial x^{\mathrm{T}}} f(x)^{\mathrm{T}} g(x) = g(x)^{\mathrm{T}} \frac{\partial f(x)}{\partial x^{\mathrm{T}}} + f(x)^{\mathrm{T}} \frac{\partial g(x)}{\partial x^{\mathrm{T}}}.$$

**Lemma 4.** *Let $f : \mathbb{R}^m \to \mathbb{R}^l$ and $g : \mathbb{R}^m \to \mathbb{R}^r$ be two vector-valued differentiable functions of $x \in \mathbb{R}^m$. Then*

$$\frac{\partial}{\partial x^{\mathrm{T}}} \mathbf{vec}(f(x)g(x)^{\mathrm{T}}) = (g(x) \otimes I_l)\frac{\partial f(x)}{\partial x^{\mathrm{T}}} + (I_r \otimes f(x))\frac{\partial g(x)}{\partial x^{\mathrm{T}}}.$$

**Lemma 5.** *Let $A$ and $B$ be two $n \times n$ positive definite matrices, and let $\alpha_1 \geq \cdots \geq \alpha_n > 0$ and $\beta_1 \geq \cdots \geq \beta_n > 0$ be the ordered eigenvalues of $A$ and $B$, respectively. Then for any $x \in \mathbb{R}^n$,*

$$x^{\mathrm{T}} A^{1/2} B A^{1/2} x \leq \alpha_1 \beta_1 \|x\|^2.$$

Table 2: Running time of three algorithms for differentiating the Sinkhorn loss, with maximum 10000 iterations.

| | | Mean solving time (ms) | Std. of solving time | Mean total time (ms) | Std. of total time | Converged |
|---|---|---|---|---|---|---|
| $m = n = 64$ $p = 8$ $\lambda^{-1} = 0.1$ | Analytic | 19.01 | 2.10 | **19.27** | 2.10 | 100 |
| | Implicit | 267.95 | 24.57 | 277.86 | 24.58 | 12 |
| | Unroll | 269.49 | 24.76 | 986.46 | 90.67 | 12 |
| $m = n = 64$ $p = 8$ $\lambda^{-1} = 0.01$ | Analytic | 23.28 | 3.46 | **23.54** | 3.48 | 100 |
| | Implicit | 275.42 | 0.34 | 284.84 | 4.74 | 0 |
| | Unroll | 275.54 | 0.49 | 1011.70 | 0.75 | 0 |
| $m = n = 128$ $p = 16$ $\lambda^{-1} = 0.1$ | Analytic | 23.76 | 2.92 | **24.16** | 2.90 | 100 |
| | Implicit | 297.36 | 0.34 | 320.29 | 4.32 | 0 |
| | Unroll | 297.93 | 0.63 | 1074.47 | 1.33 | 0 |
| $m = n = 128$ $p = 16$ $\lambda^{-1} = 0.01$ | Analytic | 32.21 | 5.00 | **32.68** | 5.01 | 100 |
| | Implicit | 297.79 | 0.30 | 324.95 | 9.99 | 0 |
| | Unroll | 297.90 | 0.32 | 1073.24 | 0.84 | 0 |
| $m = n = 256$ $p = 32$ $\lambda^{-1} = 0.1$ | Analytic | 28.72 | 3.84 | **29.57** | 3.83 | 100 |
| | Implicit | 342.23 | 1.53 | 386.80 | 9.22 | 0 |
| | Unroll | 343.18 | 0.70 | 1256.10 | 2.94 | 0 |
| $m = n = 256$ $p = 32$ $\lambda^{-1} = 0.01$ | Analytic | 44.45 | 6.14 | **45.32** | 6.18 | 100 |
| | Implicit | 343.57 | 2.01 | 411.12 | 18.92 | 0 |
| | Unroll | 341.00 | 2.02 | 1253.67 | 3.33 | 0 |
| $m = n = 512$ $p = 64$ $\lambda^{-1} = 0.1$ | Analytic | 41.50 | 5.60 | **43.27** | 5.53 | 100 |
| | Implicit | 469.04 | 0.77 | 552.64 | 15.31 | 0 |
| | Unroll | 465.03 | 1.82 | 1705.51 | 6.36 | 0 |
| $m = n = 512$ $p = 64$ $\lambda^{-1} = 0.01$ | Analytic | 67.97 | 9.69 | **69.68** | 9.67 | 100 |
| | Implicit | 464.59 | 1.04 | 624.28 | 26.48 | 0 |
| | Unroll | 464.16 | 0.55 | 1707.04 | 3.31 | 0 |

*Proof.* Let $U_{1 \times n} = x^{\mathrm{T}} A^{1/2}$, and then $u := UU^{\mathrm{T}} = x^{\mathrm{T}} Ax \leq \alpha_1 \|x\|^2$, and $UBU^{\mathrm{T}} = x^{\mathrm{T}} A^{1/2} B A^{1/2} x$. By Theorem A.4 in page 788 of Marshall et al. (2011), we immediately get

$$UBU^{\mathrm{T}} = \mathrm{tr}(UBU^{\mathrm{T}}) \leq \beta_1 u \leq \alpha_1 \beta_1 \|x\|^2.$$

$\square$

**Lemma 6.** *Let $A$ and $B$ be two symmetric matrices of the same size. Then*

$$\sigma_{\min}(A) + \sigma_{\min}(B) \leq \sigma_{\min}(A + B) \leq \sigma_{\max}(A + B) \leq \sigma_{\max}(A) + \sigma_{\max}(B).$$

*Proof.* Using the well-known identity $\sigma_{\max}(A) = \max_{\|x\|=1} x^{\mathrm{T}} Ax$, we have

$$\sigma_{\max}(A + B) = \max_{\|x\|=1} x^{\mathrm{T}}(A + B)x \leq \max_{\|x\|=1} x^{\mathrm{T}} Ax + \max_{\|x\|=1} x^{\mathrm{T}} Bx = \sigma_{\max}(A) + \sigma_{\max}(B).$$

Applying the inequality above to $-(A + B)$, we get the result on the opposite direction. $\square$

**Lemma 7.** *Let $f(\beta)$ be defined as in (12), and let $\mu = M^{\mathrm{T}}a$ and $u_i = \max_j M_{ij}$, $i = 1, \ldots, n$. If $f(\beta) \le c$, then we have $\max_j \beta_j \le U_c$ and $\min_j \beta_j \ge L_c$, where*

$$U_c = b_m^{-1} \left[ \left( \max_{1 \le j \le m-1} \mu_j \right) + \lambda^{-1} \sum_{i=1}^n a_i \log a_i - \lambda^{-1} + c \right]_+ \tag{16}$$

$$L_c = - \left( \min_{1 \le j \le m-1} b_j \right)^{-1} \left[ \sum_{i=1}^n a_i u_i + \lambda^{-1} \sum_{i=1}^n a_i \log a_i - \lambda^{-1} + c \right]_+. \tag{17}$$

*Proof.* By definition,

$$f(\beta) = \lambda^{-1} \sum_{i=1}^n a_i \log \left[ \sum_{j=1}^m e^{\lambda(\beta_j - M_{ij})} \right] - \lambda^{-1} \sum_{i=1}^n a_i \log a_i - \beta^{\mathrm{T}} b + \lambda^{-1}.$$

If $f(\beta) \le c$, then

$$c_0 := c + \lambda^{-1} \sum_{i=1}^n a_i \log a_i - \lambda^{-1} \ge \lambda^{-1} \sum_{i=1}^n a_i \log \left[ \sum_{j=1}^m e^{\lambda(\beta_j - M_{ij})} \right] - \beta^{\mathrm{T}} b.$$

By definition we have $\beta_m = 0$, and let $J = \arg\max_{1 \le j \le m-1} \beta_j$. Then

$$c_0 \ge \lambda^{-1} \sum_{i=1}^n a_i \log \left[ e^{\lambda(\beta_J - M_{iJ})} \right] - \beta^{\mathrm{T}} b = \sum_{i=1}^n a_i (\beta_J - M_{iJ}) - \sum_{j=1}^{m-1} \beta_j b_j$$

$$\ge \beta_J - \sum_{i=1}^n a_i M_{iJ} - \beta_J \sum_{j=1}^{m-1} b_j = b_m \beta_J - \mu_J \ge b_m \beta_J - \max_{1 \le j \le m-1} \mu_j,$$

which verifies (16) by noting that $\max_j \beta_j = [\beta_J]_+$.

Next, let $K = \arg\min_{1 \le j \le m-1} \beta_j$. We can assume that $\beta_K < 0$, since otherwise the trivial bound $\min_j \beta_j = \beta_m = 0$ is already met. Consider the sets $S_+ = \{j : \beta_j > 0\}$ and $S_- = \{j : \beta_j < 0\}$. Then clearly,

$$\beta^{\mathrm{T}} b = \beta_K b_K + \sum_{\substack{j \ne K \\ j \in S_+}} \beta_j b_j + \sum_{\substack{j \ne K \\ j \in S_-}} \beta_j b_j \le \beta_K b_K + [\beta_J]_+ \cdot \sum_{\substack{j \ne K \\ j \in S_+}} b_j + 0 \le \beta_K b_K + [\beta_J]_+ (1 - b_m).$$

Also note that $\log(\sum_{j=1}^m e^{x_j}) \ge \max_j x_j$ for any $x_1, \ldots, x_n \in \mathbb{R}$, so

$$\lambda^{-1} \sum_{i=1}^n a_i \log \left[ \sum_{j=1}^m e^{\lambda(\beta_j - M_{ij})} \right] \ge \lambda^{-1} \sum_{i=1}^n a_i \log \left[ \sum_{j=1}^m e^{\lambda(\beta_j - u_i)} \right] = \lambda^{-1} \sum_{i=1}^n a_i \left[ \log \left( \sum_{j=1}^m e^{\lambda \beta_j} \right) - \lambda u_i \right]$$

$$= \lambda^{-1} \log \left( \sum_{j=1}^m e^{\lambda \beta_j} \right) - \sum_{i=1}^n a_i u_i \ge \lambda^{-1} \cdot \max_j (\lambda \beta_j) - \sum_{i=1}^n a_i u_i$$

$$= \max_j \beta_j - \sum_{i=1}^n a_i u_i = [\beta_J]_+ - \sum_{i=1}^n a_i u_i.$$

As a result,

$$c_0 \ge \lambda^{-1} \sum_{i=1}^n a_i \log \left[ \sum_{j=1}^m e^{\lambda(\beta_j - M_{ij})} \right] - \beta^{\mathrm{T}} b \ge [\beta_J]_+ - \sum_{i=1}^n a_i u_i - \beta_K b_K - [\beta_J]_+ (1 - b_m)$$

$$= b_m [\beta_J]_+ - \sum_{i=1}^n a_i u_i - \beta_K b_K \ge - \sum_{i=1}^n a_i u_i - \beta_K b_K,$$

and then

$$\beta_K \geq -b_K^{-1} \left[ \sum_{i=1}^n a_i u_i + c_0 \right]_+ \geq - \left( \min_{1 \leq j \leq m-1} b_j \right)^{-1} \left[ \sum_{i=1}^n a_i u_i + c_0 \right]_+ ,$$

which verifies (17). □

**Lemma 8.** *Let $T$ be an $n \times m$ matrix with strictly positive entries, and suppose that $n \geq m$. Define $\mu = T\mathbf{1}_m$, $\nu = T^{\mathrm{T}}\mathbf{1}_n$, and*

$$H = \begin{pmatrix} \mathbf{diag}(\mu) & \tilde{T} \\ \tilde{T}^{\mathrm{T}} & \mathbf{diag}(\tilde{\nu}) \end{pmatrix}, \qquad D = \mathbf{diag}(\tilde{\nu}) - \tilde{T}^{\mathrm{T}}\mathbf{diag}(\mu)^{-1}\tilde{T}.$$

*Then*

$$\sigma_{\max}(D) \leq \max_{1 \leq j \leq m-1} \nu_j,$$

$$\sigma_{\min}(D) \geq \sigma_{\min}(H) \geq \frac{n-m+2}{2(n-m+1)} \cdot \min_{1 \leq i \leq n} T_{im},$$

$$\sigma_{\min}(D) \geq \min_{1 \leq i \leq m-1} \sum_{j=1}^{m-1} D_{ij} = \min_{1 \leq j \leq m-1} \sum_{i=1}^{n} \mu_i^{-1} T_{ij} T_{im}.$$

*Proof.* Consider the matrix $S = H - sJ$, where $J$ is an $(n+m-1) \times (n+m-1)$ matrix filled of ones, and $s$ is a positive scalar. Let

$$R_k = \sum_{j \neq k} |S_{kj}|, \quad k = 1, \ldots, n+m-1.$$

Suppose $s \leq \min_{1 \leq i \leq n, 1 \leq j \leq m-1} T_{ij}$, and then for $k = 1, \ldots, n$, it is easy to find that

$$R_k = (n-1)s + \sum_{j=1}^{m-1} (T_{kj} - s) = (n-1)s + \mu_k - T_{km} - (m-1)s = (n-m)s + \mu_k - T_{km},$$

and for $k = n+1, \ldots, n+m-1$,

$$R_k = (m-2)s + \sum_{i=1}^{n} (T_{i,k-n} - s) = (m-2)s + \nu_{k-n} - ns = (m-n-2)s + v_{k-n}.$$

Then it is easy to see that

$$S_{kk} - R_k = \begin{cases} \mu_k - R_k = T_{km} - (n-m)s, & k = 1, \ldots, n \\ \nu_{k-n} - R_k = (n+2-m)s, & k = n+1, \ldots, n+m-1 \end{cases}.$$

Let

$$\min_{1 \leq i \leq n} T_{im} - (n-m)s = (n+2-m)s,$$

and then $s = \min_{1 \leq i \leq n} T_{im}/(2n - 2m + 2)$, and

$$S_{kk} - R_k \geq L := \frac{n-m+2}{2(n-m+1)} \cdot \min_{1 \leq i \leq n} T_{im} > 0$$

for all $k$. By the Gershgorin circle theorem, every eigenvalue of $S$ must be greater than $L$. Since $H = S + sJ$ and $J$ is nonnegative definite, we also have $\sigma_{\min}(H) \geq L > 0$, implying that $H$ is positive definite.

For the second formula, it is easy to find that the $D$ matrix is the Schur complement of the block $\mathbf{diag}(\mu)$ of the $H$ matrix. So by Theorem 3.1 of Fan (2002), we have $\sigma_{\min}(D) \geq \sigma_{\min}(H)$ and $\sigma_{\max}(D) \leq \sigma_{\max}(\mathbf{diag}(\tilde{\nu})) = \max_{1 \leq j \leq m-1} \nu_j$.

Finally, let $c = \max_{1 \leq j \leq m-1} \nu_j$, and then $D$ can be expressed as $D = cI_{m-1} - B$, where $B = \tilde{T}^{\mathrm{T}} \mathbf{diag}(\mu)^{-1} \tilde{T} + \mathbf{diag}(c\mathbf{1}_{m-1} - \tilde{\nu})$ is a matrix that have nonnegative entries. In addition, we have proved that $D$ is positive definite, so $D$ is a nonsingular $M$-matrix by the definition in Tian & Huang (2010). Then Theorem 3.2 of Tian & Huang (2010) shows that

$$\sigma_{\min}(D) \geq \min_{1 \leq i \leq m-1} \sum_{j=1}^{m-1} D_{ij}.$$

Let $\delta = D\mathbf{1}_{m-1}$, and then clearly $\min_{1 \leq i \leq m-1} \sum_{j=1}^{m-1} D_{ij} = \min_i \delta_i$. Note that

$$\delta = D\mathbf{1}_{m-1} = \tilde{\nu} - \tilde{T}^{\mathrm{T}} \mathbf{diag}(\mu)^{-1} \tilde{T} \mathbf{1}_{m-1} = \tilde{\nu} - \tilde{T}^{\mathrm{T}} \mathbf{diag}(\mu)^{-1} (T\mathbf{1}_m - T_m)$$
$$= \tilde{\nu} - \tilde{T}^{\mathrm{T}} \mathbf{diag}(\mu)^{-1} (\mu - T_m) = \tilde{\nu} - \tilde{T}^{\mathrm{T}} \mathbf{1}_n + \tilde{T}^{\mathrm{T}} \mathbf{diag}(\mu)^{-1} T_m = \tilde{T}^{\mathrm{T}} \mathbf{diag}(\mu)^{-1} T_m,$$

where $T_m$ stands for the $m$-th column of $T$. Therefore,

$$\min_{1 \leq i \leq m-1} \delta_i = \min_{1 \leq j \leq m-1} \sum_{i=1}^{n} \mu_i^{-1} T_{ij} T_{im}.$$

$\square$

## C.2    Proof of Lemma 1

Let $T = \mathrm{e}_\lambda[\alpha \oplus \beta - M]$, and then it is easy to find that $\nabla_\alpha \mathcal{L}(\alpha, \beta) = a - T\mathbf{1}_m$ and $\nabla_{\tilde{\beta}} \mathcal{L}(\alpha, \beta) = \tilde{b} - \tilde{T}^{\mathrm{T}} \mathbf{1}_n$. Since $\alpha^*(\beta) = \arg\max_\alpha \mathcal{L}(\alpha, \beta)$, we find that $\alpha_i \equiv \alpha^*(\beta)_i$ is the solution to the equation $a - T\mathbf{1}_m = 0$. By definition, we have

$$a_i = \sum_{j=1}^{m} e^{\lambda(\alpha_i + \beta_j - M_{ij})} = e^{\lambda \alpha_i} \sum_{j=1}^{m} e^{\lambda(\beta_j - M_{ij})}, \quad i = 1, \dots, n,$$

so the solution is $\alpha_i = \lambda^{-1} \log a_i - \lambda^{-1} \log \left( \sum_{j=1}^{m} e^{\lambda(\beta_j - M_{ij})} \right)$. Since $T\mathbf{1}_m = a$, we immediately get $\mathbf{1}_n^{\mathrm{T}} T\mathbf{1}_m = 1$, so

$$\mathcal{L}(\alpha^*(\beta), \beta) = \alpha^*(\beta)^{\mathrm{T}} a + \beta^{\mathrm{T}} b - \lambda^{-1} \mathbf{1}_n^{\mathrm{T}} T\mathbf{1}_m = \alpha^*(\beta)^{\mathrm{T}} a + \beta^{\mathrm{T}} b - \lambda^{-1},$$

and we get the expression for $f(\beta) = -\mathcal{L}(\alpha^*(\beta), \beta)$. Finally,

$$\nabla_{\tilde{\beta}} \mathcal{L}(\alpha^*(\beta), \beta) = \left[ \frac{\partial \alpha^*(\beta)}{\partial \tilde{\beta}^{\mathrm{T}}} \right]^{\mathrm{T}} \nabla_\alpha \mathcal{L}(\alpha, \beta) \big|_{\alpha = \alpha^*(\beta)} + \nabla_{\tilde{\beta}} \mathcal{L}(\alpha, \beta) \big|_{\alpha = \alpha^*(\beta)}.$$

Since $\alpha^*(\beta) = \arg\max_\alpha \mathcal{L}(\alpha, \beta)$ implies that $\nabla_\alpha \mathcal{L}(\alpha, \beta) \big|_{\alpha = \alpha^*(\beta)} = 0$, we have $\nabla_{\tilde{\beta}} \mathcal{L}(\alpha^*(\beta), \beta) = \nabla_{\tilde{\beta}} \mathcal{L}(\alpha, \beta) \big|_{\alpha = \alpha^*(\beta)}$, and hence

$$\nabla_{\tilde{\beta}} f(\beta) = -\nabla_{\tilde{\beta}} \mathcal{L}(\alpha^*(\beta), \beta) = \tilde{T}(\beta)^{\mathrm{T}} \mathbf{1}_n - \tilde{b}.$$

## C.3    Proof of Theorem 4.1

By definition we have

$$S_\lambda(\mu, \nu) = \langle T^*, M \rangle = \mathbf{vec}(T^*)^{\mathrm{T}} \mathbf{vec}(M),$$

so Lemma 3 gives

$$\frac{\partial S_\lambda(\mu, \nu)}{\partial \mathbf{vec}(M)^{\mathrm{T}}} = \mathbf{vec}(M)^{\mathrm{T}} \frac{\partial \mathbf{vec}(T^*)}{\partial \mathbf{vec}(M)^{\mathrm{T}}} + \mathbf{vec}(T^*)^{\mathrm{T}} \frac{\partial \mathbf{vec}(M)}{\partial \mathbf{vec}(M)^{\mathrm{T}}}. \tag{18}$$

Obviously, $\partial \mathbf{vec}(M) / \partial \mathbf{vec}(M)^{\mathrm{T}}$ is the $(nm) \times (nm)$ identity matrix $I_{(nm)}$, so the second term of (18) is essentially $\mathbf{vec}(T^*)^{\mathrm{T}}$, and the remaining task is to derive $\partial \mathbf{vec}(T^*) / \partial \mathbf{vec}(M)^{\mathrm{T}}$.

Let $R = \alpha^* \oplus \beta^* - M = \alpha^* \mathbf{1}_m^{\mathrm{T}} + \mathbf{1}_n \beta^{*\mathrm{T}} - M$, and then $T^* = \mathrm{e}_\lambda[R]$. Using the chain rule of derivatives, we have

$$\frac{\partial \mathbf{vec}(T^*)}{\partial \mathbf{vec}(M)^{\mathrm{T}}} = \frac{\partial \mathbf{vec}(T^*)}{\partial \mathbf{vec}(R)^{\mathrm{T}}} \cdot \frac{\partial \mathbf{vec}(R)}{\partial \mathbf{vec}(M)^{\mathrm{T}}}. \tag{19}$$

It is easy to find that $\partial \mathbf{vec}(T^*)/\partial \mathbf{vec}(R)^{\mathrm{T}}$ is an $(nm) \times (nm)$ diagonal matrix with diagonal elements $\mathbf{vec}(\lambda T^*)$, so

$$\mathbf{vec}(M)^{\mathrm{T}} \frac{\partial \mathbf{vec}(T^*)}{\partial \mathbf{vec}(R)^{\mathrm{T}}} = \lambda \mathbf{vec}(M \odot T^*)^{\mathrm{T}}. \tag{20}$$

Furthermore,

$$\begin{aligned}
\frac{\partial \mathbf{vec}(R)}{\partial \mathbf{vec}(M)^{\mathrm{T}}} &= \frac{\partial \mathbf{vec}(\alpha^* \mathbf{1}_m^{\mathrm{T}} + \mathbf{1}_n \beta^{*\mathrm{T}} - M)}{\partial \mathbf{vec}(M)^{\mathrm{T}}} \\
&= (\mathbf{1}_m \otimes I_n) \frac{\partial \alpha^*}{\partial \mathbf{vec}(M)^{\mathrm{T}}} + (I_m \otimes \mathbf{1}_n) \frac{\partial \beta^*}{\partial \mathbf{vec}(M)^{\mathrm{T}}} - I_{(nm)},
\end{aligned} \tag{21}$$

where the second identity is an application of Lemma 4. Combine (19), (20) and (21), and then we get

$$\begin{aligned}
\mathbf{vec}(M)^{\mathrm{T}} \frac{\partial \mathbf{vec}(T^*)}{\partial \mathbf{vec}(M)^{\mathrm{T}}} &= \lambda \mathbf{vec}(M \odot T^*)^{\mathrm{T}} \frac{\partial \mathbf{vec}(R)}{\partial \mathbf{vec}(M)^{\mathrm{T}}} \\
&= \lambda \mathbf{vec}(M \odot T^*)^{\mathrm{T}} (\mathbf{1}_m \otimes I_n) \frac{\partial \alpha^*}{\partial \mathbf{vec}(M)^{\mathrm{T}}} \\
&\quad + \lambda \mathbf{vec}(M \odot T^*)^{\mathrm{T}} (I_m \otimes \mathbf{1}_n) \frac{\partial \beta^*}{\partial \mathbf{vec}(M)^{\mathrm{T}}} \\
&\quad - \lambda \mathbf{vec}(M \odot T^*)^{\mathrm{T}},
\end{aligned}$$

which is the first term of (18).

Using the identities in Lemma 2, we have

$$\begin{aligned}
\mathbf{vec}(M \odot T^*)^{\mathrm{T}} (\mathbf{1}_m \otimes I_n) &= \left[ (\mathbf{1}_m \otimes I_n)^{\mathrm{T}} \mathbf{vec}(M \odot T^*) \right]^{\mathrm{T}} = \left[ (\mathbf{1}_m^{\mathrm{T}} \otimes I_n) \mathbf{vec}(M \odot T^*) \right]^{\mathrm{T}} \\
&= \mu_r^{\mathrm{T}} := \left[ \mathbf{vec}((M \odot T^*) \mathbf{1}_m) \right]^{\mathrm{T}}, \\
\mathbf{vec}(M \odot T^*)^{\mathrm{T}} (I_m \otimes \mathbf{1}_n) &= \left[ (I_m \otimes \mathbf{1}_n)^{\mathrm{T}} \mathbf{vec}(M \odot T^*) \right]^{\mathrm{T}} = \left[ (I_m \otimes \mathbf{1}_n^{\mathrm{T}}) \mathbf{vec}(M \odot T^*) \right]^{\mathrm{T}} \\
&= \mu_c^{\mathrm{T}} := \left[ \mathbf{vec}(\mathbf{1}_n^{\mathrm{T}} (M \odot T^*)) \right]^{\mathrm{T}}.
\end{aligned}$$

Since we have set $\beta_m^* = 0$, (18) simplifies to

$$\frac{\partial S_\lambda(\mu, \nu)}{\partial \mathbf{vec}(M)^{\mathrm{T}}} = \lambda \left[ \mu_r^{\mathrm{T}} \frac{\partial \alpha^*}{\partial \mathbf{vec}(M)^{\mathrm{T}}} + \tilde{\mu}_c^{\mathrm{T}} \frac{\partial \tilde{\beta}^*}{\partial \mathbf{vec}(M)^{\mathrm{T}}} - \mathbf{vec}(M \odot T^*)^{\mathrm{T}} \right] + \mathbf{vec}(T^*)^{\mathrm{T}}. \tag{22}$$

Let $w^* = (\alpha^{*\mathrm{T}}, \tilde{\beta}^{*\mathrm{T}})^{\mathrm{T}}$, and then the main challenge is to calculate $\partial w^* / \partial \mathbf{vec}(M)^{\mathrm{T}}$.

First, note that the optimality condition for $(\alpha^*, \beta^*) = \arg\max_{\alpha, \beta} \mathcal{L}(\alpha, \beta)$ is

$$\nabla_\alpha \mathcal{L}(\alpha, \beta)|_{(\alpha, \beta) = (\alpha^*, \beta^*)} = \mathbf{0}, \quad \nabla_\beta \mathcal{L}(\alpha, \beta)|_{(\alpha, \beta) = (\alpha^*, \beta^*)} = \mathbf{0}. \tag{23}$$

Section C.2 has shown that $\nabla_\alpha \mathcal{L}(\alpha, \beta) = a - T\mathbf{1}_m$ and $\nabla_{\tilde{\beta}} \mathcal{L}(\alpha, \beta) = \tilde{b} - \tilde{T}^{\mathrm{T}} \mathbf{1}_n$. Moreover,

$$\begin{aligned}
\nabla_\alpha^2 \mathcal{L}(\alpha, \beta) &= -\lambda \mathbf{diag}(T\mathbf{1}_m) \\
\nabla_{\tilde{\beta}}^2 \mathcal{L}(\alpha, \beta) &= -\lambda \mathbf{diag}(\tilde{T}^{\mathrm{T}} \mathbf{1}_n) \\
\nabla_{\tilde{\beta}} (\nabla_\alpha \mathcal{L}(\alpha, \beta)) &= -\lambda \tilde{T}.
\end{aligned}$$

Define the function

$$F(w, M) = \begin{pmatrix} \nabla_\alpha \mathcal{L}(\alpha, \beta) \\ \nabla_{\tilde{\beta}} \mathcal{L}(\alpha, \beta) \end{pmatrix} = \begin{pmatrix} a - T\mathbf{1}_m \\ \tilde{b} - \tilde{T}^{\mathrm{T}} \mathbf{1}_n \end{pmatrix}, \tag{24}$$

where $w = (\alpha^{\mathrm{T}}, \tilde{\beta}^{\mathrm{T}})^{\mathrm{T}}$, and then $\tilde{w}^*$ satisfies the equation $F(w^*, M) = \mathbf{0}$, indicating that $w^*$ is implicitly a function of $M$, written as $w^* = w(M)$. By the implicit function theorem,

$$\frac{\partial w(M)}{\partial \mathbf{vec}(M)^{\mathrm{T}}} = -F_w^{-1} F_M := -\left[ \frac{\partial F(w, M)}{\partial w^{\mathrm{T}}} \bigg|_{w=w^*} \right]^{-1} \frac{\partial F(w, M)}{\partial \mathbf{vec}(M)^{\mathrm{T}}} \bigg|_{w=w^*}.$$

Note that

$$F_w = F_w^{\mathrm{T}} = -\lambda \begin{pmatrix} A & \tilde{B} \\ \tilde{B}^{\mathrm{T}} & \tilde{D} \end{pmatrix} := \begin{pmatrix} \nabla_\alpha^2 \mathcal{L}(\alpha, \beta) & \nabla_{\tilde{\beta}} \left( \nabla_\alpha \mathcal{L}(\alpha, \beta) \right) \\ \left[ \nabla_{\tilde{\beta}} \left( \nabla_\alpha \mathcal{L}(\alpha, \beta) \right) \right]^{\mathrm{T}} & \nabla_{\tilde{\beta}}^2 \mathcal{L}(\alpha, \beta) \end{pmatrix}$$

Then by the inversion formula for block matrices, we have

$$F_w^{-1} = -\lambda^{-1} \begin{pmatrix} A & \tilde{B} \\ \tilde{B}^{\mathrm{T}} & \tilde{D} \end{pmatrix}^{-1} = -\lambda^{-1} \begin{pmatrix} A^{-1} + A^{-1}\tilde{B}\tilde{\Delta}^{-1}\tilde{B}^{\mathrm{T}}A^{-1} & -A^{-1}\tilde{B}\tilde{\Delta}^{-1} \\ -\tilde{\Delta}^{-1}\tilde{B}^{\mathrm{T}}A^{-1} & \tilde{\Delta}^{-1} \end{pmatrix},$$

where $\tilde{\Delta} = \tilde{D} - \tilde{B}^{\mathrm{T}}A^{-1}\tilde{B}$. For $g = (\mu_r^{\mathrm{T}}, \tilde{\mu}_c^{\mathrm{T}})^{\mathrm{T}}$, the vector $\tilde{s} = (s_u^{\mathrm{T}}, \tilde{s}_v^{\mathrm{T}})^{\mathrm{T}} = -\lambda F_w^{-1}\tilde{g}$ has the following expression:

$$\tilde{s}_v = -\tilde{\Delta}^{-1}\tilde{B}^{\mathrm{T}}A^{-1}\mu_r + \tilde{\Delta}^{-1}\tilde{\mu}_c$$

$$s_u = A^{-1}\mu_r + A^{-1}\tilde{B}\tilde{\Delta}^{-1}\tilde{B}^{\mathrm{T}}A^{-1}\mu_r - A^{-1}\tilde{B}\tilde{\Delta}^{-1}\tilde{\mu}_c,$$

$$= A^{-1}\mu_r - A^{-1}\tilde{B}\tilde{s}_v.$$

After some simplification, we obtain

$$\tilde{\Delta} = \mathbf{diag}(\tilde{T}^{\mathrm{T}}\mathbf{1}_n) - \tilde{T}^{\mathrm{T}}\mathbf{diag}((T\mathbf{1}_m)^-)\tilde{T}$$

$$\tilde{s}_v = \tilde{\Delta}^{-1}\tilde{\mu}_c - \tilde{\Delta}^{-1}\tilde{T}^{\mathrm{T}}((T\mathbf{1}_m)^- \odot \mu_r)$$

$$s_u = (T\mathbf{1}_m)^- \odot \mu_r - (T\mathbf{1}_m)^- \odot (\tilde{T}\tilde{s}_v).$$

Next, partition $F_M$ as $F_M = \begin{pmatrix} G_M \\ \tilde{H}_M \end{pmatrix}$, where $G_M \in \mathbb{R}^{n \times (nm)}$ and $\tilde{H}_M \in \mathbb{R}^{(m-1) \times (nm)}$. By definition,

$$G_M = \begin{pmatrix} \frac{\partial G_1}{\partial \mathbf{vec}(M)^{\mathrm{T}}} \\ \vdots \\ \frac{\partial G_n}{\partial \mathbf{vec}(M)^{\mathrm{T}}} \end{pmatrix}, \quad G_i = -\sum_{j=1}^m T_{ij} = -\sum_{j=1}^m e^{\lambda(\alpha_i + \beta_j - M_{ij})},$$

so

$$\frac{\partial G_i}{\partial M_{kl}} = \begin{cases} 0, & i \neq k \\ \lambda T_{kl}, & i = k \end{cases}.$$

This indicates that $G_M = \lambda \left( \mathbf{diag}(T_1), \ldots, \mathbf{diag}(T_m) \right)$, where $T_1, \ldots, T_m$ are the column vectors of $T$. Similarly, for $H_j = -\sum_{i=1}^n T_{ij}$,

$$\tilde{H}_M = \begin{pmatrix} \frac{\partial H_1}{\partial \mathbf{vec}(M)^{\mathrm{T}}} \\ \vdots \\ \frac{\partial H_{m-1}}{\partial \mathbf{vec}(M)^{\mathrm{T}}} \end{pmatrix}, \quad \frac{\partial H_j}{\partial M_{kl}} = \begin{cases} 0, & j \neq l \\ \lambda T_{kl}, & j = l \end{cases},$$

implying that

$$\tilde{H}_M = \lambda \begin{pmatrix} T_1^{\mathrm{T}} & & \\ & \ddots & \\ & & T_{m-1}^{\mathrm{T}} & \mathbf{0}_n^{\mathrm{T}} \end{pmatrix}.$$

As a result,

$$\mu_r^{\mathrm{T}} \frac{\partial \alpha^*}{\partial \mathbf{vec}(M)^{\mathrm{T}}} + \tilde{\mu}_c^{\mathrm{T}} \frac{\partial \tilde{\beta}^*}{\partial \mathbf{vec}(M)^{\mathrm{T}}}$$

$$= (\mu_r^{\mathrm{T}}, \tilde{\mu}_c^{\mathrm{T}}) \frac{\partial w^*}{\partial \mathbf{vec}(M)^{\mathrm{T}}} = -(\mu_r^{\mathrm{T}}, \tilde{\mu}_c^{\mathrm{T}}) F_w^{-1} F_M$$

$$= \left[ -\lambda F_w^{-1} \begin{pmatrix} \mu_r \\ \tilde{\mu}_c \end{pmatrix} \right]^{\mathrm{T}} \begin{pmatrix} \lambda^{-1} G_M \\ \lambda^{-1} \tilde{H}_M \end{pmatrix} = (s_u^{\mathrm{T}}, \tilde{s}_v^{\mathrm{T}}) \begin{pmatrix} \lambda^{-1} G_M \\ \lambda^{-1} \tilde{H}_M \end{pmatrix}$$

$$= \left( (s_u \odot T_1)^{\mathrm{T}}, \ldots, (s_u \odot T_m)^{\mathrm{T}} \right) + \left( \tilde{s}_{v,1} T_1^{\mathrm{T}}, \ldots, \tilde{s}_{v,m-1} T_{m-1}^{\mathrm{T}}, \mathbf{0}_n^{\mathrm{T}} \right)$$

$$= \left[ \mathbf{vec} \left( \mathbf{diag}(s_u)T + T\mathbf{diag}(s_v) \right) \right]^{\mathrm{T}}. \tag{25}$$

Finally, substitute (25) into (22), and we have

$$\frac{\partial S_\lambda(\mu, \nu)}{\partial \mathbf{vec}(M)^{\mathrm{T}}} = \lambda \left[ \mathbf{vec}\left(\mathbf{diag}(s_u)T^* + T^*\mathbf{diag}(s_v)\right) - \mathbf{vec}(M \odot T)\right]^{\mathrm{T}} + \mathbf{vec}(T)^{\mathrm{T}}$$

Transforming back to the matrix form, and we obtain

$$\frac{\partial S_\lambda(\mu, \nu)}{\partial M} = \lambda(s_u \oplus s_v - M) \odot T + T.$$

Replacing $T$ with $T^*$ and noting that $a = T^*\mathbf{1}_m$, $\tilde{b} = \tilde{T}^{*\mathrm{T}}\mathbf{1}_n$, we get the stated result. The positive definiteness of the $\tilde{\Delta}$ matrix is a direct consequence of Lemma 8.

### C.4 Proof of Theorem 5.1

In the proof of Theorem 4.1 we have already shown that

$$\nabla^2_{\alpha,\tilde{\beta}}\mathcal{L}(\alpha, \beta) = -\lambda H := -\lambda \begin{pmatrix} \mathbf{diag}(T\mathbf{1}_m) & \tilde{T} \\ \tilde{T}^{\mathrm{T}} & \mathbf{diag}(\tilde{T}^{\mathrm{T}}\mathbf{1}_n) \end{pmatrix},$$

where $T = e_\lambda[\alpha \oplus \beta - M]$. Plugging $\alpha^*(\beta)$ into $\mathcal{L}(\alpha, \beta)$, and then $\nabla^2_{\tilde{\beta}}\mathcal{L}(\alpha^*(\beta), \beta)$ is the Schur complement of the top left block of $\nabla^2_{\alpha,\tilde{\beta}}\mathcal{L}(\alpha, \beta)$, given by

$$\nabla^2_{\tilde{\beta}}\mathcal{L}(\alpha^*(\beta), \beta) = -\lambda\left[\mathbf{diag}(\tilde{T}^{\mathrm{T}}\mathbf{1}_n) - \tilde{T}^{\mathrm{T}}\mathbf{diag}(T\mathbf{1}_m)^{-1}\tilde{T}\right].$$

Since $f(\beta) = -\mathcal{L}(\alpha^*(\beta), \beta)$, by Lemma 8 we find that $\nabla^2_{\tilde{\beta}}f(\beta)$ is positive definite, so $f(\beta)$ is strictly convex on $\tilde{\beta}$, and hence $\beta^*$ is unique.

The optimality conditions for $(\alpha^*, \beta^*)$ are $T^*\mathbf{1}_m = a$ and $T^{*\mathrm{T}}\mathbf{1}_n = b$, where $T^* = e_\lambda[\alpha^* \oplus \beta^* - M]$. Since $T^*_{ij} = \exp\{\lambda(\alpha^*_i + \beta^*_j - M_{ij})\} \geq 0$ and $a_i = \sum_{j=1}^m T^*_{ij}$, $b_j = \sum_{i=1}^n T^*_{ij}$, we have $T^*_{ij} \leq \min\{a_i, b_j\}$ for all $i$ and $j$, implying that

$$\alpha^*_i + \beta^*_j \leq U_{ij} := M_{ij} + \lambda^{-1}\min\{\log(a_i), \log(b_j)\}.$$

Since $\beta^*_m = 0$ by design, we have $\alpha^*_i \leq U_{\alpha_i} < +\infty, i = 1, \ldots, n$, where $U_{\alpha_i} = U_{im} = M_{im} + \lambda^{-1}\min\{\log(a_i), \log(b_m)\}$. This indicates that $\alpha^*_i$ is upper bounded.

Next, let $I = \arg\max_i T^*_{im}$. Since $T^{*\mathrm{T}}\mathbf{1}_n = b$ implies that $b_m = \sum_{i=1}^n T^*_{im} \leq nT^*_{Im}$, we have

$$T^*_{Im} = \exp\{\lambda(\alpha^*_I - M_{Im})\} \geq b_m/n,$$

and hence $\alpha^*_I \geq M_{Im} + \lambda^{-1}\log(b_m/n)$. Again, since $\alpha^*_i + \beta^*_j \leq U_{ij}$ for all $i$ and $j$, it holds that

$$\begin{aligned} \beta^*_j &\leq U_{Ij} - \alpha^*_I \leq U_{Ij} - M_{Im} - \lambda^{-1}\log(b_m/n) \\ &= M_{Ij} + \lambda^{-1}\min\{\log(a_I), \log(b_j)\} - M_{Im} - \lambda^{-1}\log(b_m/n) \\ &= M_{Ij} - M_{Im} + \lambda^{-1}\min\{\log(na_I/b_m), \log(nb_j/b_m)\} \\ &:= U_{\beta_j} < +\infty, \quad j = 1, \ldots, m. \end{aligned} \tag{26}$$

On the other hand, $T^{*\mathrm{T}}\mathbf{1}_n = b$ implies that $b_j = \sum_{i=1}^n T^*_{ij} = e^{\lambda\beta^*_j} \cdot \sum_{i=1}^n e^{\lambda(\alpha^*_i - M_{ij})}$ for any $j$, so

$$\log b_j = \lambda\beta^*_j + \log\left[\sum_{i=1}^n e^{\lambda(\alpha^*_i - M_{ij})}\right] \leq \lambda\beta^*_j + \log\left[\sum_{i=1}^n e^{\lambda(U_{\alpha_i} - M_{ij})}\right].$$

It is well-known that

$$\max\{x_1, \ldots, x_n\} \leq \log\left(\sum_{i=1}^n e^{x_i}\right) \leq \max\{x_1, \ldots, x_n\} + \log n$$

for any $x_1, \ldots, x_n \in \mathbb{R}$, so

$$
\begin{aligned}
\beta_j^* &\geq \lambda^{-1} \log b_j - \lambda^{-1} \log \left[ \sum_{i=1}^n e^{\lambda(U_{\alpha_i} - M_{ij})} \right] \\
&\geq \lambda^{-1} \log b_j - \max_i \left( U_{\alpha_i} - M_{ij} \right) - \lambda^{-1} \log n \\
&\geq L_{\beta_j} := \lambda^{-1} \log(b_j/n) - \max_i \left( U_{\alpha_i} - M_{ij} \right) > -\infty, \quad j = 1, \ldots, m.
\end{aligned}
\tag{27}
$$

Then (26) and (27) together show that $|\beta_j^*| < \infty$.

Similarly, $T^* \mathbf{1}_m = a$, so

$$
\log a_i \leq \lambda \alpha_i^* + \log \left[ \sum_{j=1}^m e^{\lambda(U_{\beta_j} - M_{ij})} \right],
$$

$$
\alpha_i^* \geq L_{\alpha_i} := \lambda^{-1} \log(a_i/m) - \max_j \left( U_{\beta_j} - M_{ij} \right) > -\infty, \quad i = 1, \ldots, n.
$$

The trivial bounds $\underline{L}_{\alpha_i}$ and $\overline{U}_{\beta_j}$ are obtained by removing the unknown index $I$.

The results above verify that $|\alpha_i^*| < \infty$ and $|\beta_j^*| < \infty$, and hence $\|\alpha^*\| < \infty$ and $\|\beta^*\| < \infty$. Finally, plugging in $\beta^*$ to the objective function, and we immediately get $f^* > -\infty$.

## C.5 Proof of Theorem 5.2

Claims (a) and (b) are direct consequences of the convergence property of the L-BFGS algorithm (Theorem 7.1, Liu & Nocedal (1989)), and we only need to verify its three assumptions. The new results here are explicit expressions for the constants $C_1$, $C_2$, and $r$.

First, $f$ is twice continuously differentiable, so Assumption 7.1(1) of Liu & Nocedal (1989) is verified. Second, $f$ is a closed convex function, and we define the level set of $f$ as $L_c = \{\tilde{\beta} \in \mathbb{R}^{m-1} : f(\beta) \leq c\}$. Theorem 5.1 has shown that $f^* > -\infty$, and when $c = f^*$, obviously $L_c = \{\tilde{\beta}^*\}$ is non-empty and bounded. Then Corollary 8.7.1 of Rockafellar (1970) shows that $L_c$ is bounded for every $c$. In particular, for a fixed initial value $\tilde{\beta}^{(0)}$, define $L = \{\tilde{\beta} : f(\beta) \leq f(\beta^{(0)})\}$, and then $L$ is a bounded, closed, and convex set, which verifies Assumption 7.1(2) of Liu & Nocedal (1989). Third, let $H(\beta) := \nabla_{\tilde{\beta}}^2 f(\beta)$, and then in the proof of Theorem 5.1 we have already shown that

$$
H(\beta) = \lambda \left[ \mathbf{diag}(\tilde{T}^{\mathrm{T}} \mathbf{1}_n) - \tilde{T}^{\mathrm{T}} \mathbf{diag}(T \mathbf{1}_m)^{-1} \tilde{T} \right],
$$

where $T = \mathrm{e}_\lambda[\alpha^*(\beta) \oplus \beta - M]$. Lemma 8 verifies that

$$
\sigma_{\min}(H(\beta)) \geq \lambda \cdot \frac{n - m + 2}{2(n - m + 1)} \cdot \min_{1 \leq i \leq n} T_{im}, \qquad \sigma_{\max}(H(\beta)) \leq \lambda \cdot \max_{1 \leq j \leq m-1} \nu_j,
$$

where $\nu = T^{\mathrm{T}} \mathbf{1}_n$. On the $L$ set, Lemma 7 shows that $\max_j \beta_j \leq U_c$ and $\min_j \beta_j \geq L_c$, with $c = f(\beta^{(0)})$. Therefore,

$$
\begin{aligned}
\alpha_i &:= \lambda^{-1} \log a_i - \lambda^{-1} \log \left[ \sum_{j=1}^m e^{\lambda(\beta_j - M_{ij})} \right] \geq \lambda^{-1} \log a_i - \lambda^{-1} \log \left[ e^{-\lambda M_{im}} + \sum_{j=1}^{m-1} e^{\lambda(U_c - M_{ij})} \right] \\
&= A_i := \lambda^{-1} \log a_i - U_c - \lambda^{-1} \log \left( e^{-\lambda(M_{im} + U_c)} + \sum_{j=1}^{m-1} e^{-\lambda M_{ij}} \right),
\end{aligned}
$$

$$
T_{im} = e^{\lambda(\alpha_i - M_{im})} \geq e^{\lambda(A_i - M_{im})}.
$$

On the other hand, $T_{ij}$ must satisfy $T_{ij} > 0$ and $\sum_{j=1}^m T_{ij} = a_i$ for any $i$ and $j$, so for $j = 1, \ldots, m - 1$, we have $T_{ij} \leq a_i - T_{im}$. Therefore,

$$
\nu_j = \sum_{i=1}^n T_{ij} \leq \sum_{i=1}^n (a_i - T_{im}) = 1 - \sum_{i=1}^n T_{im} \leq 1 - \sum_{i=1}^n e^{\lambda(A_i - M_{im})}, \quad j = 1, \ldots, m - 1.
$$

This implies that there exist constants $M_1, M_2 > 0$ such that

$$M_1 \|x\|^2 \le x^{\mathrm{T}} H(\beta) x \le M_2 \|x\|^2$$

for all $x \in \mathbb{R}^{m-1}$ and $\tilde{\beta} \in L$, with

$$M_1 = \lambda \cdot \frac{n - m + 2}{2(n - m + 1)} \cdot \min_{1 \le i \le n} e^{\lambda(A_i - M_{im})},$$

$$M_2 = \lambda \left[ 1 - \sum_{i=1}^{n} e^{\lambda(A_i - M_{im})} \right].$$

This verifies Assumption 7.1(3) of Liu & Nocedal (1989).

Next, we derive the explicit constants in the theorem. Following the notations in equation (7.3) of Liu & Nocedal (1989), the BFGS matrix $B_k$ for the L-BFGS algorithm has the following expression

$$B_k = B^{(\tilde{m})}, \quad B^{(l+1)} = B^{(l)} - \frac{B^{(l)} s_l s_l^{\mathrm{T}} B^{(l)}}{s_l^{\mathrm{T}} B^{(l)} s_l} + \frac{y_l y_l^{\mathrm{T}}}{y_l^{\mathrm{T}} s_l},$$

where $\tilde{m} = \min\{k+1, m_0\}$, $m_0$ is a user-defined constant explained in Section A, and $\{y_l\}$ and $\{s_l\}$ are two sequences of vectors. We also choose $B^{(0)} = I$ to be the identity matrix.

Fix $l = \tilde{m}$, and let $\cos\theta_k = s_l^{\mathrm{T}} B^{(l)} s_l / (\|s_l\| \cdot \|B^{(l)} s_l\|)$, $\rho_k = y_l^{\mathrm{T}} s_l / \|s_l\|^2$, $\tau_k = \|y_l\|^2 / y_l^{\mathrm{T}} s_l$, and $q_k = s_l^{\mathrm{T}} B^{(l)} s_l / \|s_l\|^2$. Then it can be verified that

$$\mathrm{tr}(B^{(l+1)}) = \mathrm{tr}(B^{(l)}) - \frac{\|B^{(l)} s_l\|^2}{s_l^{\mathrm{T}} B^{(l)} s_l} + \frac{\|y_l\|^2}{y_l^{\mathrm{T}} s_l} = \mathrm{tr}(B^{(l)}) - \frac{q_k}{\cos^2\theta_k} + \tau_k, \tag{28}$$

$$\det(B^{(l+1)}) = \det(B^{(l)}) \rho_k / q_k. \tag{29}$$

Define $\psi(B) = \mathrm{tr}(B) - \log\det(B)$, and it is known that $\psi(B) > 0$ for any positive definite matrix $B$. Equation (6.50) of Nocedal & Wright (2006) shows that

$$0 < \psi(B^{(l+1)}) = \mathrm{tr}(B^{(l)}) - \frac{q_k}{\cos^2\theta_k} + \tau_k - \log\det(B^{(l)}) - \log\rho_k + \log q_k$$

$$= \psi(B^{(l)}) + (\tau_k - \log\rho_k - 1) + \left(1 - \frac{q_k}{\cos^2\theta_k} + \log\frac{q_k}{\cos^2\theta_k}\right) + \log\cos^2\theta_k.$$

Under the assumptions verified above, equations (7.8) and (7.9) of Liu & Nocedal (1989) show that $M_1 \le y_l^{\mathrm{T}} s_l / \|s_l\|^2 \le M_2$ and $\|y_l\|^2 / y_l^{\mathrm{T}} s_l \le M_2$ for every $l$, so $M_1 \le \rho_k \le M_2$ and $\tau_k \le M_2$. Since $h(x) = 1 - x + \log(x) \le 0$ for all $x > 0$, we have

$$0 < \psi(B_k) + M_3 + \log\cos^2\theta_k, \tag{30}$$

where $M_3 := M_2 - \log M_1 - 1$. Now we show that $\psi(B_k)$ can be upper bounded. First, Lemma 6 implies that for $l = 0, \ldots, \tilde{m} - 1$,

$$\sigma_{\max}(B^{(l+1)}) \le \sigma_{\max}(B^{(l)}) + 0 + \|y_l\|^2 / y_l^{\mathrm{T}} s_l \le \sigma_{\max}(B^{(l)}) + M_2,$$

so $\sigma_{\max}(B_k) \le 1 + m_0 M_2$. This also implies that $q_k \le 1 + m_0 M_2$. Next, (28) shows that $\mathrm{tr}(B_k) \le m - 1 + m_0 M_2$, and (29) gives

$$\log\det(B^{(l+1)}) = \log\det(B^{(l)}) + \log\rho_k - \log q_k \ge \log\det(B^{(l)}) + \log M_1 - \log(1 + m_0 M_2),$$

implying that $\log\det(B_k) \ge m_0 [\log M_1 - \log(1 + m_0 M_2)]$. As a result, we get

$$\psi(B_k) \le M_4 := m - 1 + m_0 M_2 - m_0 [\log M_1 - \log(1 + m_0 M_2)]. \tag{31}$$

Combining (30) and (31), we have $\cos^2\theta_k > e^{-(M_3 + M_4)}$.

Finally, using the argument in Byrd et al. (1987), we have $f^{(k+1)} - f^* \leq r(f^{(k)} - f^*)$, where

$$r = 1 - c_1(1-c_2)M_1/M_2 \cos^2\theta_k < 1 - c_1(1-c_2)M_1/M_2 e^{-(M_3+M_4)},$$

and $c_1, c_2$ are two constants for the Wolfe condition as explained in Section A. The constant $C_1$ is simply $2/M_1$.

For (c), we follow the analysis in Nocedal et al. (2002). Let $g(\beta) = \nabla_{\tilde{\beta}} f(\beta)$, and then $g(\beta^*) = \mathbf{0}$. By Taylor's theorem we have

$$f(\beta) - f^* = \frac{1}{2}(\tilde{\beta} - \tilde{\beta}^*)^{\mathrm{T}} H_1(\tilde{\beta} - \tilde{\beta}^*), \tag{32}$$

where $H_1 = H(\xi)$ for some $\xi$ in the line segment connecting $\tilde{\beta}$ and $\tilde{\beta}^*$. Also,

$$g(\beta) - g(\beta^*) = g(\beta) = H_2(\tilde{\beta} - \tilde{\beta}^*), \quad H_2 = \int_0^1 H(\tilde{\beta} + t(\tilde{\beta}^* - \tilde{\beta}))\mathrm{d}t. \tag{33}$$

Combining (32) and (33), we get

$$\|g(\beta)\|^2 = (\tilde{\beta} - \tilde{\beta}^*)^{\mathrm{T}} H_2^2 (\tilde{\beta} - \tilde{\beta}^*) = \frac{2(\tilde{\beta} - \tilde{\beta}^*)^{\mathrm{T}} H_2^2 (\tilde{\beta} - \tilde{\beta}^*)}{(\tilde{\beta} - \tilde{\beta}^*)^{\mathrm{T}} H_1 (\tilde{\beta} - \tilde{\beta}^*)} \cdot (f(\beta) - f^*)$$

for any $\tilde{\beta} \in L$. Let $x = H_1^{1/2}(\tilde{\beta} - \tilde{\beta}^*)$, and then

$$\|g(\beta)\|^2 = \frac{2x^{\mathrm{T}} H_1^{-1/2} H_2^2 H_1^{-1/2} x}{\|x\|^2} \cdot (f(\beta) - f^*).$$

It is easy to find that

$$\sigma_{\max}(H_1^{-1}) = [\sigma_{\min}(H_1)]^{-1} \leq M_1^{-1}, \quad \sigma_{\max}(H_2^2) = [\sigma_{\max}(H_2)]^2 \leq M_2^2.$$

By Lemma 5, we have $x^{\mathrm{T}} H_1^{-1/2} H_2^2 H_1^{-1/2} x \leq M_1^{-1} M_2^2 \|x\|^2$ for any $x \in \mathbb{R}^{m-1}$, and hence $\|g(\beta)\|^2 \leq C_2(f(\beta) - f^*)$ for all $\tilde{\beta} \in L$, where $C_2 = 2M_1^{-1}M_2^2$. Since $f^{(k)} \leq f^{(0)}$ due to claim (a), we find that $\tilde{\beta}^{(k)} \in L$ for all $k > 0$. Therefore,

$$\|g(\gamma^{(k)})\|^2 \leq C_2(f^{(k)} - f^*) \leq C_2 r^k (f^{(0)} - f^*),$$

and claim (c) is proved.

Claims (d) and (e) can be verified as follows. For any $\tilde{\beta} \in L$, recall that $T = e_\lambda[\alpha^*(\beta) \oplus \beta - M]$, and then $T\mathbf{1}_m - a = 0$ and

$$\|\tilde{T}^{\mathrm{T}} \mathbf{1}_n - \tilde{b}\|_\infty \leq \|\tilde{T}^{\mathrm{T}} \mathbf{1}_n - \tilde{b}\| = \|\nabla_{\tilde{\beta}} f(\beta)\|,$$

indicating that

$$\left| \sum_{i=1}^n T_{ij} - b_j \right| \leq \|\nabla_{\tilde{\beta}} f(\beta)\|, \quad j = 1, \ldots, m-1. \tag{34}$$

Then $0 < T_{ij} \leq \sum_{j=1}^m T_{ij} = a_i$ and $0 < T_{ij} \leq \sum_{i=1}^n T_{ij} \leq b_j + \|\nabla_{\tilde{\beta}} f(\beta)\|$. The gradient $\|\nabla_{\tilde{\beta}} f(\beta)\|$ can be bounded using claim (c), so (d) is also proved. On the other hand, (34) shows that

$$a_i = \sum_{j=1}^m T_{ij} \leq m \cdot \max_j T_{ij},$$

and similarly we have

$$b_j - \|\nabla_{\tilde{\beta}} f(\beta)\| \leq \sum_{i=1}^n T_{ij} \leq n \cdot \max_i T_{ij}.$$

Replacing $\|\nabla_{\tilde{\beta}} f(\beta)\|$ by its upper bound, and claim (e) is verified.

## C.6   Proof of Theorem 5.3

For matrix $A_{n \times m} = (a_{ij})$, define $\|A\|_\infty = \max_{i,j} |a_{ij}|$, and the notation $A \geq 0$ means $a_{ij} \geq 0$ for all $i$ and $j$. First, it is easy to show that $\|A \odot B\|_F \leq \|A\|_\infty \|B\|_F$, since

$$\|A \odot B\|_F = \sqrt{\sum_{i,j} (a_{ij} b_{ij})^2} \leq \sqrt{\sum_{i,j} \|A\|_\infty^2 b_{ij}^2} = \|A\|_\infty \|B\|_F.$$

Next, we show that if $B_{n \times m} \geq 0$, $C_{p \times n} \geq 0$, and $v \geq 0$, where $v$ is a vector, then $\|C(A \odot B)v\| \leq \|A\|_\infty \|CBv\|$.

*Proof.* Let $u_i$ be the $i$-th element of $(A \odot B)v$, and then

$$u_i = \sum_{j=1}^m a_{ij} b_{ij} v_j, \quad |u_i| \leq \|A\|_\infty \sum_{j=1}^m b_{ij} v_j.$$

Consequently,

$$\|C(A \odot B)v\| = \sqrt{\sum_{k=1}^p \left| \sum_{i=1}^n c_{ki} u_i \right|^2} \leq \sqrt{\sum_{k=1}^p \left( \sum_{i=1}^n c_{ki} |u_i| \right)^2} \leq \|A\|_\infty \sqrt{\sum_{k=1}^p \left[ \sum_{i=1}^n c_{ki} \left( \sum_{j=1}^m b_{ij} v_j \right) \right]^2}$$

$$= \|A\|_\infty \|CBv\|.$$

$\square$

Similarly, if $A \geq 0$ and $v \geq 0$, then

$$\|A(u \odot v)\| = \sqrt{\sum_{i=1}^n \left| \sum_{j=1}^m a_{ij} u_j v_j \right|^2} \leq \|u\|_\infty \sqrt{\sum_{i=1}^n \left( \sum_{j=1}^m a_{ij} v_j \right)^2} = \|u\|_\infty \|Av\|.$$

Moreover, for matrices $B_{n \times m} \geq 0$, and $C_{m \times p} \geq 0$, let $C_j$ be the $j$-th column of $C$, and then

$$\|(A \odot B)C\|_F = \sqrt{\sum_{j=1}^p \|(A \odot B)C_i\|^2} \leq \|A\|_\infty \sqrt{\sum_{j=1}^p \|BC_i\|^2} = \|A\|_\infty \|BC\|_F.$$

Let $\alpha = \alpha^* + f$ and $\beta = \beta^* + g$ for some perturbation vectors $f$ and $g$, and define $T = e_\lambda[\alpha \oplus \beta - M]$. Then it is easy to find that

$$T - T^* = e_\lambda[\alpha^* \oplus \beta^* - M + (f \oplus g)] - T^* = e_\lambda[f \oplus g] \odot T^* - T^*.$$

Let $E_T = e_\lambda[f \oplus g] - \mathbf{1}_n \mathbf{1}'_m$, so $T - T^* = E_T \odot T^*$. Since $|e^x - 1| < 2|x|$ for $|x| < 1$, we have

$$|(E_T)_{ij}| = |e^{\lambda(f_i + g_j)} - 1| < 2\lambda |f_i + g_j|$$

as long as $\lambda |f_i + g_j| < 1$. This can be achieved by assuming $\varepsilon := 2\lambda(\|f\|_\infty + \|g\|_\infty) < 1$, since in this case $\lambda |f_i + g_j| \leq \lambda(\|f\|_\infty + \|g\|_\infty) < 1/2$. Then clearly $\|E_T\|_\infty < \varepsilon$.

Consider $\hat{s}_u = s_u + \delta_u$ and $\hat{s}_v = s_v + \delta_v$ for some perturbation vectors $\delta_u$ and $\delta_v$, and let

$$\widehat{\nabla_M S} = T + \lambda(\hat{s}_u \oplus \hat{s}_v - M) \odot T$$
$$= T + \lambda(s_u \oplus s_v - M) \odot T + \lambda(\delta_u \oplus \delta_v) \odot T$$
$$= \nabla_M S + E_T \odot \nabla_M S + \lambda(\delta_u \oplus \delta_v) \odot (T^* + E_T \odot T^*).$$

Then we have

$$
\begin{aligned}
\|\widehat{\nabla_M S} - \nabla_M S\|_F &\leq \|E_T \odot \nabla_M S\|_F + \|\lambda(\delta_u \oplus \delta_v) \odot (T^* + E_T \odot T^*)\|_F \\
&\leq \|E_T\|_\infty \|\nabla_M S\|_F + \lambda\|\delta_u \oplus \delta_v\|_\infty \|T^* + E_T \odot T^*\|_F \\
&< \varepsilon\|\nabla_M S\|_F + \lambda\|\delta_u \oplus \delta_v\|_\infty \|\mathbf{1}_n \mathbf{1}'_m + E_T\|_\infty \|T^*\|_F \\
&\leq \varepsilon\|\nabla_M S\|_F + \lambda\|\delta_u \oplus \delta_v\|_\infty (1 + \|E_T\|_\infty)\|T^*\|_F \\
&< \varepsilon\|\nabla_M S\|_F + \lambda(1 + \varepsilon)\|\delta_u \oplus \delta_v\|_\infty \|T^*\|_F \\
&< \varepsilon\|\nabla_M S\|_F + 2\lambda\|T^*\|_F(\|\delta_u\|_\infty + \|\delta_v\|_\infty).
\end{aligned}
$$

Therefore, we just need to show proper bounds for $\|\delta_u\|_\infty$ and $\|\delta_v\|_\infty$.

Consider $\hat{\mu}_r = (M \odot T)\mathbf{1}_m$ and $\hat{\mu}_c = (M \odot T)^{\mathrm{T}}\mathbf{1}_n$, and then

$$
\begin{aligned}
\hat{\mu}_r &= (M \odot (T^* + E_T \odot T^*))\mathbf{1}_m = \mu_r + \delta_r := \mu_r + (E_T \odot M \odot T^*)\mathbf{1}_m, \\
\hat{\mu}_c &= (M \odot (T^* + E_T \odot T^*))^{\mathrm{T}}\mathbf{1}_n = \mu_c + \delta_c := \mu_c + (E_T \odot M \odot T^*)^{\mathrm{T}}\mathbf{1}_m.
\end{aligned}
$$

It can be easily verified that

$$
\begin{aligned}
\|a^- \odot \delta_r\| &= \|\mathbf{diag}(a^-)(E_T \odot M \odot T^*)\mathbf{1}_m\| \leq \|E_T\|_\infty \|\mathbf{diag}(a^-)(M \odot T^*)\mathbf{1}_m\| < \varepsilon\|\mu_r\|, \\
\|T^{*\mathrm{T}}(a^- \odot \delta_r)\| &= \|T^{*\mathrm{T}}\mathbf{diag}(a^-)(E_T \odot M \odot T^*)\mathbf{1}_m\| \leq \|E_T\|_\infty \|T^{*\mathrm{T}}\mathbf{diag}(a^-)(M \odot T^*)\mathbf{1}_m\| \\
&< \varepsilon\|T^{*\mathrm{T}}(a^- \odot \mu_r)\|, \\
\|\delta_c\| &= \|(E_T \odot M \odot T^*)^{\mathrm{T}}\mathbf{1}_m\| \leq \|E_T\|_\infty \|(M \odot T^*)^{\mathrm{T}}\mathbf{1}_m\| < \varepsilon\|\mu_c\|.
\end{aligned}
$$

Define $b_v = \mu_c - T^{*\mathrm{T}}(a^- \odot \mu_r)$ and $\hat{b}_v = \hat{\mu}_c - T^{\mathrm{T}}(a^- \odot \hat{\mu}_r)$, and we have

$$
\begin{aligned}
\hat{b}_v &= \mu_c + \delta_c - (T^* + E_T \odot T^*)^{\mathrm{T}}(a^- \odot (\mu_r + \delta_r)) \\
&= \mu_c + \delta_c - T^{*\mathrm{T}}(a^- \odot (\mu_r + \delta_r)) - (E_T \odot T^*)^{\mathrm{T}}(a^- \odot (\mu_r + \delta_r)) \\
&= \mu_c - T^{*\mathrm{T}}(a^- \odot \mu_r) + \delta_c - T^{*\mathrm{T}}(a^- \odot \delta_r) - (E_T \odot T^*)^{\mathrm{T}}(a^- \odot (\mu_r + \delta_r))
\end{aligned}
$$

As a result,

$$
\begin{aligned}
\|\hat{b}_v - b_v\| &= \|\delta_c - T^{*\mathrm{T}}(a^- \odot \delta_r) - (E_T \odot T^*)^{\mathrm{T}}(a^- \odot (\mu_r + \delta_r))\| \\
&\leq \|\delta_c\| + \|T^{*\mathrm{T}}(a^- \odot \delta_r)\| + \|E_T\|_\infty \|T^{*\mathrm{T}}(a^- \odot (\mu_r + \delta_r))\| \\
&< \varepsilon\|\mu_c\| + \varepsilon\|T^{*\mathrm{T}}(a^- \odot \mu_r)\| + \varepsilon\|T^{*\mathrm{T}}(a^- \odot \mu_r)\| + \varepsilon\|T^{*\mathrm{T}}(a^- \odot \delta_r)\| \\
&< \varepsilon\|\mu_c\| + \varepsilon\|T^{*\mathrm{T}}(a^- \odot \mu_r)\| + \varepsilon\|T^{*\mathrm{T}}(a^- \odot \mu_r)\| + \varepsilon^2\|T^{*\mathrm{T}}(a^- \odot \mu_r)\| \\
&< \varepsilon\|\mu_c\| + 3\varepsilon\|T^{*\mathrm{T}}(a^- \odot \mu_r)\|.
\end{aligned}
$$

On the other hand, let $\hat{t}_{ij}$ be the $(i,j)$ element of the matrix $\tilde{T}^{\mathrm{T}}\mathbf{diag}(a^-)\tilde{T}$, and $t_{ij}$ be the $(i,j)$ element of $\tilde{T}^{*\mathrm{T}}\mathbf{diag}(a^-)\tilde{T}^*$. Then

$$
\hat{t}_{ij} = \sum_{k=1}^n T_{ki}T_{kj}/a_k = \sum_{k=1}^n [1 + (E_T)_{ki}][1 + (E_T)_{kj}]T^*_{ki}T^*_{kj}/a_k,
$$

$$
|\hat{t}_{ij} - t_{ij}| = \left| \sum_{k=1}^n [(E_T)_{ki} + (E_T)_{kj} + (E_T)_{ki}(E_T)_{kj}]T^*_{ki}T^*_{kj}/a_k \right|
$$

$$
\leq (2\varepsilon + \varepsilon^2) \sum_{k=1}^n |T^*_{ki}T^*_{kj}/a_k| < 3\varepsilon t_{ij}.
$$

This implies that

$$
\|\hat{D} - D\|_F = \|\tilde{T}^{*\mathrm{T}}\mathbf{diag}(a^-)\tilde{T}^* - \tilde{T}^{\mathrm{T}}\mathbf{diag}(a^-)\tilde{T}\|_F = \sqrt{\sum_{i,j} |\hat{t}_{ij} - t_{ij}|^2} \leq 3\varepsilon\|D\|_F.
$$

Then by Theorem 7.2 of Higham (2002), if $3\varepsilon\sigma\|D\|_F < 1$, we have

$$\frac{\|\delta_v\|}{\|s_v\|} \le \frac{\varepsilon\sigma}{1 - 3\varepsilon\sigma\|D\|_F}\left(\frac{\|\mu_c\| + 3\|T^{*\mathrm{T}}(a^- \odot \mu_r)\|}{\|s_v\|} + 3\|D\|_F\right).$$

where $\sigma = \|D^{-1}\|_{\mathrm{op}} = 1/\sigma_{\min}(D)$. Assume that $\varepsilon < \min\{1, 1/(6\sigma\|D\|_F)\}$, then with slight simplification, we have $\|\delta_v\|_\infty \le \|\delta_v\| \le C_v\varepsilon$, where

$$C_v = 2\sigma(\|\mu_c\| + 3\|T^{*\mathrm{T}}(a^- \odot \mu_r)\| + 3\|D\|_F\|s_v\|).$$

On the other hand,

$$
\begin{aligned}
\|\delta_u\|_\infty \le \|\delta_u\| &\le \|a^- \odot \delta_r\| + \|a^- \odot (T\hat{s}_v - T^*s_v)\| \\
&\le \varepsilon\|\mu_r\| + \|a^- \odot (T^*\hat{s}_v + (E_T \odot T^*)\hat{s}_v - T^*s_v)\| \\
&\le \varepsilon\|\mu_r\| + \|a^- \odot (T^*\delta_v)\| + \|a^- \odot ((E_T \odot T^*)\hat{s}_v)\| \\
&\le \varepsilon\|\mu_r\| + \|a^- \odot (T^*\delta_v)\| + \|\mathbf{diag}(a^-)(E_T \odot T^*)\hat{s}_v\| \\
&\le \varepsilon\|\mu_r\| + \|a^- \odot (T^*\delta_v)\| + \varepsilon\|\mathbf{diag}(a^-)T^*\hat{s}_v\| \\
&= \varepsilon\|\mu_r\| + \|a^- \odot (T^*\delta_v)\| + \varepsilon\|\mathbf{diag}(a^-)T^*(s_v + \delta_v)\| \\
&\le \varepsilon\|\mu_r\| + (1+\varepsilon)\|a^- \odot (T^*\delta_v)\| + \varepsilon\|\mathbf{diag}(a^-)T^*s_v\| \\
&\le \varepsilon\|\mu_r\| + (1+\varepsilon)\|\mathbf{diag}(a^-)T^*\|_F\|\delta_v\| + \varepsilon\|a^- \odot (T^*s_v)\| \\
&\le \varepsilon(\|\mu_r\| + 2C_v\|\mathbf{diag}(a^-)T^*\|_F + \|a^- \odot (T^*s_v)\|).
\end{aligned}
$$

Combining the results together, we get

$$\|\widehat{\nabla_M S} - \nabla_M S\|_F \le \varepsilon\left[\|\nabla_M S\|_F + 2\lambda\|T^*\|_F(C_v + C_u)\right],$$

where $C_u = \|\mu_r\| + 2C_v\|\mathbf{diag}(a^-)T^*\|_F + \|a^- \odot (T^*s_v)\|$.

Finally, Theorem 5.2(b) shows that $\|g\|_\infty \le \|g\| = \|\beta^{(k)} - \beta^*\| \le \sqrt{C_1\varepsilon^{(k)}}$. In addition, by (11) we have

$$\alpha_i^{(k)} := \alpha^*(\beta^{(k)})_i = \lambda^{-1}\log a_i - \lambda^{-1}\log\left[\sum_{j=1}^m e^{\lambda(\beta_j^{(k)} - M_{ij})}\right].$$

Then using Lemma (**??**), we obtain

$$|\alpha_i^{(k)} - \alpha^*| = \lambda^{-1}\left|\log\left[\sum_{j=1}^m e^{\lambda(\beta_j^{(k)} - M_{ij})}\right] - \log\left[\sum_{j=1}^m e^{\lambda(\beta_j^* - M_{ij})}\right]\right| \le \lambda^{-1}\|\lambda(\beta_j^{(k)} - \beta_j^*)\|_\infty = \|g\|_\infty,$$

implying that $\|f\|_\infty = \|\alpha^*(\beta^{(k)}) - \alpha^*\|_\infty \le \|g\|_\infty \le \sqrt{C_1\varepsilon^{(k)}}$. As a result, $\varepsilon = 2\lambda(\|f\|_\infty + \|g\|_\infty) \le 4\lambda\sqrt{C_1\varepsilon^{(k)}}$.

## C.7 Proof of Theorem 5.4

**Lemma 9.** *Let $(\mathcal{X}, d)$ be a Polish space, and $p \in [1, \infty)$. Denote the space of Borel probability measures on $\mathcal{X}$ as $P(\mathcal{X})$. The Wasserstein space of order $p$ is defined as $P_p(\mathcal{X}) = \{\mu \in P(\mathcal{X}); \int_{\mathcal{X}} d(x_0, x)^p\,\mu(dx) < \infty\}$, where $x_0 \in \mathcal{X}$ is arbitrary. Let $(\mu_k \in P_p(\mathcal{X}))_{k \in \mathbb{N}}$ be a sequence of probability measure, and $\mu \in P_p(\mathcal{X})$. If $W_p(\mu, \mu_k) \to 0$, then $\mu_k \xrightarrow{d} \mu$.*

This lemma is a direct result of Theorem 6.9 in Villani (2009).

**Lemma 10.** *Let $\widetilde{\Delta}_N := \{\nu_N | \nu_N = \sum_{i=1}^N \omega_i\delta_{X_i}, \forall X_i \in \mathcal{X}, \omega_i \ge 0, \sum_{i=1}^N \omega_i = 1\}$, meaning the set of all probability measures on $\mathcal{X}$ supported on at most $N$ points. Then*

$$\min_{\widetilde{\mu}_N \in \widetilde{\Delta}_N} W_p(\mu, \widetilde{\mu}_N) \to 0$$

This lemma is proved in Theorem 1.1 of Kloeckner (2012).

Now we use the above two lemmas to prove Theorem 5.4.

Let $\widetilde{\mu}_N^* = \underset{\widetilde{\mu}_N \in \widetilde{\Delta}_N}{\operatorname{argmin}} W_p(\mu, \widetilde{\mu}_N)$, $\mu_B^* = \underset{\mu_B \in \Delta_B}{\operatorname{argmin}} W_p(\mu, \mu_B)$.

By Lemma 10, we have $W_p(\mu, \widetilde{\mu}_N^*) \to 0$. Therefore, $\forall \epsilon > 0, \exists N \in \mathbb{N}$, s.t. $W_p(\mu, \widetilde{\mu}_N^*) < \epsilon/2$. Note that with $B$ large enough, we can always construct $\mu_B \in \Delta_B$ to approximate $\widetilde{\mu}_N^*$ well. For example, we can construct the $\widehat{\mu}_B \in \Delta_B$ as empirical distribution of samples generated from $\mathbf{X} \sim \widetilde{\mu}_N$, i.e. for each $N$, $\exists B_N \in \mathbb{N}$, s.t. $\forall B \in \mathbb{N}, B > B_N, W_p(\widetilde{B}_N^*, \widehat{\mu}_B) < \epsilon/2$.

Since $\mu_B^*$ is the minimizer of $W_p(\mu, \mu_B)$ over $\mu_B \in \Delta_B$, $\forall \epsilon$, $\exists N, B_N \in \mathbb{N}$, $\forall B \in \mathbb{N}, B > B_N$, s.t.:

$$\begin{aligned} W_p(\mu, \mu_B^*) &\leq W_p(\mu, \widehat{\mu}_B) \\ &\leq W_p(\mu, \widetilde{\mu}_N^*) + W_p(\widetilde{\mu}_N^*, \widehat{\mu}_B) \\ &< \epsilon. \end{aligned} \tag{35}$$

Therefore, $W_p(\mu, \mu_B^*) \to 0$.

By lemma 9, we have $\mu_B^* \xrightarrow{d} \mu$.

## C.8 Proof of Theorem 5.6

The proof is a direct result of the Theorem 4 of Défossez et al. (2020). In order to apply the theorem, There are three conditions in Section 2.3 of Défossez et al. (2020). Slightly revised to be more suitable in our situation, the three conditions are:

(a) The function $S_\lambda(\mathcal{Y}, \mathcal{D})$ is bounded below, i.e., $\exists C_* \in \mathbb{R}$, s.t.

$$\forall \mathcal{Y} \in \mathcal{X}^b, \ \forall \mathcal{D} \in \mathcal{X}^B, \ S_\lambda(\mathcal{Y}, \mathcal{D}) \geq C_*.$$

(b) The gradient of $S_\lambda(\mathcal{Y}, \mathcal{D})$ is uniformly almost surely bounded, i.e., $\exists R \geq \sqrt{\epsilon}$ ($\sqrt{\epsilon}$ is used here to simplify the final bounds), s.t.

$$\|\nabla_{\mathcal{D}} S_\lambda(\mathcal{Y}, \mathcal{D})\|_\infty \leq R - \sqrt{\epsilon}.$$

(c) The gradient of $S_\lambda(\mathcal{Y}, \mathcal{D})$ is $L$-Lipschitz continuous with respect to the $l_2$-norm, i.e., $\exists L > 0$, s.t.

$$\|\nabla_{\mathcal{D}_1} S_\lambda(\mathcal{Y}, \mathcal{D}_1) - \nabla_{\mathcal{D}_2} S_\lambda(\mathcal{Y}, \mathcal{D}_2)\|_2 \leq L\|\mathcal{D}_1 - \mathcal{D}_2\|_2.$$

Since $\nabla_{\mathcal{D}} S_\lambda(\mathcal{Y}, \mathcal{D}) = \nabla_M S_\lambda(\mathcal{Y}, \mathcal{D}) \nabla_{\mathcal{D}} M$ and notice that $\nabla_{\mathcal{D}} M$ is just derivative of point-wise $l_2$-norms, it is sufficient to verify that $\nabla_M S_\lambda(\mathcal{Y}, \mathcal{D})$ satisfies these properties.

According to the analytic form for $\nabla_M S_\lambda$ provided in Theorem 4.1, reviewing the required properties by definitions, it is trivial to verify the three conditions.

Hence, we have, $\exists C$, s.t.:

$$\mathbb{E}\left[\|\nabla_{\mathcal{D}^{(T)}} S_\lambda(\xi, \mathcal{D}^{(T)})\|^2\right] \leq CR^2(R^2 + RL + L^2)\frac{\log N}{\sqrt{N}}.$$

Let $C_* = CR^2(R^2 + RL + L^2)$, and we proved the theorem.

## C.9 Proof of Proposition 5.7

The proof is a direct consequence of the triangle inequality of the Sinkhorn loss (Luise et al., 2018). If $S_\lambda(\mathcal{D}^{(l)}, \mathcal{D}^{(l-1)}) < \delta$, then

$$|S_\lambda(\mathcal{Y}, \mathcal{D}^{(l)}) - S_\lambda(\mathcal{Y}, \mathcal{D}^{(l-1)})| \leq S_\lambda(\mathcal{D}^{(l)}, \mathcal{D}^{(l-1)}) < \delta.$$

