# OpenReview forum: "Wasserstein Coreset via Sinkhorn Loss"
_TMLR — Accepted by TMLR_

### Review · Reviewer_y7hu · 2024-11-08

**Summary Of Contributions:**

The paper studies generating Wasserstein coresets using the Sinkhorn loss, which
is a technique inspired by optimal transport theory. The authors try to address
the numerical instability by applying stochastic gradient decent. The paper
rigorously analyzes the stability of the forward and backward computations from
a theoretical perspective, and provide extensive experiments, including
real-world applications with image data, to demonstrate the performance of the
proposed methods.

**Audience:**

Yes

**Claims And Evidence:**

Yes

**Requested Changes:**

1. Improve the presentations, especially on the consistence and definitions of
   notations. See the detailed comments in the "Weaker Elements".
2. Discuss and/or compare with more benchmarks.
3. Provide some time complexity analysis. This would help readers to understand
   the possibility of scaling the method to larger datasets.
4. If possible, consider a case of larger data, either in simulation or real
   data.

**Strengths And Weaknesses:**

### Strong Aspects:

1. The problem of coreset selection is important, and Sinkhorn's algorithm,
   while powerful, has the issue of numerical instability. The paper tries to
   address this significant instability issue, which is crucial for making the
   method reliable in practical applications.
2. The derivation of an analytic expression for the Sinkhorn loss derivative
   enables its smooth application in practical machine learning frameworks.
3. The proposed WCSL is very general, making it applicable across different
   machine learning tasks without the need for task-specific modifications.
4. The numerical experiments involving both simulation studies and real-world
   applications are comprehensive.

### Weaker Elements:

1. The presentation of the paper is dense in mathematical notations, which is
   acceptable for a theoretical investigation. However, for some key concepts,
   such as Wasserstein distance and Sinkhorn Loss, the notations are not always
   clear. The paper could benefit from more consistent and clear presentation of
   notations.
   - For example, in the introduction, what exactly are $\mathcal{X}$ and $X$?
     While it may be easy to infer their meanings, an explicit statement would
     be helpful.
   - The term "p-Wasserstein distance" is introduced on page 2, but the meaning
     of $p$ is not clearly explained. Similarly, the notation $W_p(\cdot,
     \cdot)$ is not well-defined at this point.
   - What is the variable η referring to in "we globally define
     $\eta=\lambda^{-1}$ for simplicity"?
   - As noted below Equation (1), the Wasserstein distance is presented using
     varied notations, which may make it easier for the authors, but it is not
     reader-friendly. This could confuse readers who are less familiar with the
     topic. More intuitive explanations would make the paper more accessible.

2. There is limited discussion on alternative methods. The proposed method,
   though general, is closely related to existing approaches in the literature,
   such as leverage sampling (1) and optimal subsampling (2). A comparison with,
   or mention of these methods, would broaden the scope and add value to the
   paper.

   (1) Drineas, P., et al. *Fast approximation of matrix coherence and
   statistical leverage.* JMLR, 2012.

   (2) Wang, H., Zhu, R., & Ma, P. *Optimal subsampling for large sample
   logistic regression.* JASA, 2018.

3. There is no time complexity analysis, which hinders understanding of the
   computational efficiency in more detail.

4. While the paper focuses on improving computational efficiency, it lacks
   exploration of scalability to extremely large datasets. Expanding on the
   scalability in larger datasets would improve the paper.

---

> ### Author Response · Authors · 2024-12-10
> **Updates on Notations and Added Baseline Comparisons**
>
> Dear Reviewer y7hu,
>
> **Responses to Reviewer y7hu**
>
> We are grateful for your thorough review and constructive feedback. Below are our responses to your comments.
>
> 1. **Clarity and Consistency of Notation:**
>    > The presentation of the paper is dense in mathematical notations, which is acceptable for a theoretical investigation. However, for some key concepts, such as Wasserstein distance and Sinkhorn Loss, the notations are not always clear. The paper could benefit from more consistent and clear presentation of notations.
>    > For example, in the introduction, what exactly are $\mathcal{X}$ and $X$? While it may be easy to infer their meanings, an explicit statement would be helpful. The term "p-Wasserstein distance" is introduced on page 2, but the meaning of p is not clearly explained. Similarly, the notation $W_p(\cdot,\cdot)$ is not well-defined at this point. What is the variable \(\eta\) referring to in "we globally define $\eta=\lambda^{-1}$ for simplicity"?
>
> We have revised the manuscript to present notations more consistently and intuitively for all the examples mentioned. We explicitly defined $\mathcal{X}$ as a metric space and $X$ as a point in $\mathcal{X}$. Additionally, we clarified the term "p-Wasserstein distance,". The notation $W_p(\cdot,\cdot)$ has also been clearly defined at its first appearance. We removed the global definition $\eta = \lambda^{-1}$, now consistently using $\lambda$ and $\lambda^{-1}$, and clarified the coreset definition to ensure readers are guided more smoothly through the key concepts.
>
> 2. **Varied Notations for the Wasserstein Distance:**
>    > As noted below Equation (1), the Wasserstein distance is presented using varied notations, which may make it easier for the authors, but it is not reader-friendly. This could confuse readers who are less familiar with the topic. More intuitive explanations would make the paper more accessible.
>
> We acknowledge that our initial notation could be confusing. We now removed the notation $S_\lambda(M,a,b)$ and consistently use $S_\lambda(\mu,\nu)$ to represent the Sinkhorn loss, which helps maintain a clear and reader-friendly presentation throughout the paper.
>
> 3. **Comparisons with Alternative Methods:**
>    > There is limited discussion on alternative methods. The proposed method, though general, is closely related to existing approaches in the literature, such as leverage sampling (1) and optimal subsampling (2). A comparison with, or mention of these methods, would broaden the scope and add value to the paper.
>
> We have broadened the scope of our comparisons by incorporating additional baselines such as k-means centroids and kernel thinning. We also mention leverage sampling and optimal subsampling methods in the introduction. Although direct comparisons with them are not included due to limited code availability, we note their relevance and theoretical connections. Experimental results with new baselines are presented in Figures 3, 4, and 9 in the revised manuscript. Figure 3 visually compares the sample selection quality between subsampling methods on 2-dimensional datasets, while Figures 4 and 9 show the MMD and Wasserstein distance for simulated datasets and image datasets.

---

> ### Author Response · Authors · 2024-12-10
> **(Continued) Updates on Time Complexity Analysis, Scalability to Larger Datasets**
>
> 4. **Time Complexity Analysis:**
>    > There is no time complexity analysis, which hinders understanding of the computational efficiency in more detail.
>
> Before we address the time complexity analysis, we would like to briefly explain the two layer optimization framework of our method, as presented in the revised manuscript. The two layers of optimization are as follows:
> - The **outer** optimization, Wasserstein coreset construction, uses gradient descent methods like Adam to generate the coreset, with the gradient being the gradient of Sinkhorn loss.
> - The **inner** optimization, Sinkhorn loss computation and differentiation(SLCD), solves the Sinkhorn problem, where we use BFGS and consider the dual problem to efficiently compute and differentiate the Sinkhorn loss.
>
> To clarify this structure, we have added Algorithm 2 (SLCD) in the revised manuscript. The inner problem’s BFGS optimization addresses the dual formulation of Sinkhorn loss, while the outer loop employs gradient updates (e.g., Adam) for the coreset construction.
>
> We have included a detailed time complexity analysis in Section 6.1 of the revised manuscript. This provides insights into the computational efficiency and scalability of our method. As we mentioned above, the WCSL algorithm involves two layers of optimization: the inner SLCD problem and the outer coreset construction. Each SLCD step requires $O(m^3 p)$ time, where $m$ is the batch size and $p$ is the data dimension. Since $m \ll n$, where $n$ is the full dataset size, we treat $m$ as negligible. Consequently, because each iteration processes the entire dataset in mini-batches, each WCSL iteration has a complexity of $O(n p)$. Multiple iterations are typically needed for WCSL to converge, and the number of these iterations depends on the chosen stopping criterion. Based on our empirical observations, for a fixed stopping criterion, we assume that $O(n)$ iterations are required. Under this assumption, the overall time complexity is $O(n^2 p)$, which remains tractable for large-scale data scenarios.
>
> 5. **Scalability to Larger Datasets:**
>    > While the paper focuses on improving computational efficiency, it lacks exploration of scalability to extremely large datasets. Expanding on the scalability in larger datasets would improve the paper.
>
> To illustrate scalability, we have expanded our runtime analysis to larger datasets, demonstrating how our approach scales as the dataset size increases. To demonstrate the method's computational efficiency, we measured execution times for a range of dataset sizes ($n$) while fixing the data dimension ($p=10$) and coreset size ($B=250$). We varied $n$ from 100,000 to 1,000,000 and recorded the per-iteration runtime. The runtime per iteration increases proportionally with $n$, reflecting the anticipated linear scaling in computational demand. Moreover, the number of required iterations also grows roughly linearly with $n$, underscoring the suitability of the WCSL method for large-scale data applications. Together, these results confirm that WCSL provides an efficient and practical solution for large-scale problems.
>
> Warm regards,
>
> *The Authors*

---

### Review · Reviewer_Gxub · 2024-11-23

**Summary Of Contributions:**

This paper is on selecting a representative subsample, also called a coreset, that summarizes a given dataset. The authors use Wasserstien metric to formalize the notion of a representative subset. Since computing Wasserstein distance between two measures is expensive, this paper works with an approximation of Wasserstein distance, called the Sinkhorn loss. In Algorithm 1, the authors give a gradient based algorithm to construct what they call the Wasserstein coreset via Sinkhorn loss (WCSL). The authors then give analytic expressions for the gradient of the Sinkhorn loss function, state theorems on stability and convergence properties of L-BFGS/ADAM  algorithm for minimizing this loss function. The paper contains various experiments on synthetic and real data to prove the validity of the algorithm's output.

**Audience:**

Yes

**Broader Impact Concerns:**

I don't have any concerns.

**Claims And Evidence:**

No

**Requested Changes:**

I have several concerns and I cannot provide a reocmmendation for acceptance without them being addressed.

(1) How does the coreset algorithm compare with other coreset algorithms? The authors early on make a distinction of task specific vs. task agnostic methods, please define clearly what is a task agnostic coreset. What about baselines like centroids of k-means, Kernel Thinning (https://arxiv.org/abs/2105.05842),...

(2) The authors claim that their coresets are task agnostic. Why ?

(3) It is not clear to me how line 7 in Algorithm 1 type checks. It essentially reads:
 set_new <- set_old + gradient_term .
Please explain what this means. What is '+' for a set, if the gradient_term is a set.

(4) The authors talk of forward and backward passes of Sinkhorn loss function. Please define what this means to someone who has seen forward and backward passes only in the context of neural networks.

(5) How does gradient descent compare with L-BFGS for the regularized OT problem. What is known in terms of stability and convergence of gradient descent?

(6) In Theorem 5.2, what is epsilon? It is not defined before it appears in the guarantees.

(7) In theorem 5.4, what is B fixed or increasing? In the explanation that follows "*when the number of points n grows large", what is 'n'? *

(8) Algorithm 1 has only gradients of Sinkhorn loss. What is the implication of the results on L-BFGS/Adam and duality on Algorithm 1?

The current exposition has a lot of gaps and several things are not clear. I recommend a thorough revision.

**Strengths And Weaknesses:**

Strength:
Novel analytical derivation of Sinkhorn loss derivatives

Weaknesses:
- Limited comparison with existing coreset algorithms beyond SCCP and random sampling
- Insufficient justification of the task-agnostic nature of the coresets.
- Lack of clarity of presentation.

---

> ### Author Response · Authors · 2024-12-10
> **Updates on Added Baseline Comparison, Definition of Task-Agnostic Coreset, and so on.**
>
> Dear Reviewer Gxub,
>
> Thank you for your insightful comments. Your questions and suggestions have helped us clarify several aspects of our methodology. Below are our responses.
>
> **1. Comparison with Other Coreset Methods:**
> > How does the coreset algorithm compare with other coreset algorithms? What about baselines like centroids of k-means, Kernel Thinning (https://arxiv.org/abs/2105.05842),...
>
> We have added comparisons with additional baselines, including k-means centroids and kernel thinning, in the revised manuscript. For large image datasets MNIST and FashionMNIST, the kernel thinning method proved very slow; thus, we did not include the results for kernel thinning on image datasets. Experimental results with new baselines are presented in Figures 3, 4, and 9 in the revised manuscript. Figure 3 visually compares the sample selection quality between subsampling methods on 2-dimensional datasets, while Figures 4 and 9 show the MMD and Wasserstein distance for simulated datasets and image datasets.
>
> **2. Definition of Task-Agnostic Coreset:**
> > The authors early on make a distinction of task specific vs. task agnostic methods, please define clearly what is a task agnostic coreset. The authors claim that their coresets are task agnostic. Why?
>
> We have clarified the notion of a task-agnostic coreset in the introduction. Let $B$ denote the number of samples in a coreset, and define the set $\Delta_B \coloneqq \\{\nu_B \mid \nu_B = \tfrac{1}{B}\sum\_{i=1}\^B \delta\_{X_i}, X\_i \in \mathcal{X}\\}$, which consists of all probability measures supported on $B$ points, each having mass $\tfrac{1}{B}$. Given a measure $\xi$ on the metric space $\mathcal{X}$ and a particular objective function $f: \mathcal{X} \to \mathbb{R}$, we say that $\nu_B \in \Delta_B$ is a $\delta$-coreset of $\xi$ for the specific task defined by $f$ if
> $$
> \left| \mathbb{E}\_\xi[f(X)] - \mathbb{E}\_{\nu_B}[f(X)] \right| < \delta, \quad \text{with } B \ll N.
> $$
>
> To extend this notion beyond a single task, we can consider a family $\mathcal{F}$ of objective functions $f: \mathcal{X} \to \mathbb{R}$. We define the integral probability metric (IPM) associated with $\mathcal{F}$ as
> $$
> d\_\mathcal{F}(\xi,\nu) = \sup\_{f \in \mathcal{F}} \left| \mathbb{E}\_\xi[f(X)] - \mathbb{E}\_\nu[f(X)] \right|.
> $$
> A measure $\nu_B \in \Delta_B$ is then called a task-agnostic $\delta$-coreset of $\xi$ if $d_\mathcal{F}(\xi,\nu_B) < \delta$, thereby controlling the approximation error uniformly over all functions in $\mathcal{F}$.
>
> This task-agnostic formulation emphasizes the coreset's ability to perform well for a wide class of objective functions, rather than being tailored to a single specific task.
>
> In particular, consider the following two classes of functions:
> $$\mathcal{F}\_1 = \\{f \mid \\|\nabla f\\|\leq 1\\}$$
> and
> $$\mathcal{F}\_2 = \\{f \mid \\|f\\|\_{H\^1(\mu)} \leq 1\\}$$,
> where $H\^1$ is the Sobolev space $\\{f\in L^2\mid \partial {x_i}f\in L^2\\}$.
> For these function classes, with $W_1$ and $W_2$ being the 1-Wasserstein distance and 2-Wasserstein distance, one can show that:
> $$d\_{\mathcal{F}\_1}(\mu,\mu_B) = W\_1(\mu,\mu_B)$$ and
> $$d\_{\mathcal{F}\_2}(\mu,\mu_B) \leq \sqrt{C} W\_2(\mu,\mu_B)$$, where $C$ is a constant.
>
> These results indicate that the Wasserstein distance provides a suitable upper bound for the relevant integral probability metrics. Consequently, using a Wasserstein-based objective in coreset construction ensures that the resulting coreset is task-agnostic. By minimizing the Wasserstein distance, we inherently minimize an upper bound on the IPM for a broad range of objective functions $f \in \mathcal{F}_1$ or $f \in \mathcal{F}_2$, and approximates the target measure with respect to a large family of tasks.
>
> **3. Clarification of Algorithm 1 (Line 7):**
> > It is not clear to me how line 7 in Algorithm 1 type checks. It essentially reads: $set\_new \leftarrow set\_old + gradient\_term$. Please explain what this means. What is '+' for a set, if the $gradient\_term$ is a set.
>
> We apologize for the confusion. The set of points is stored as a matrix. The addition operation is element-wise and applies to each point in the matrix. Conceptually, we treat each point’s coordinates as parameters and update them by adding a gradient-based adjustment term. To be more clear, the set of points is represented as a matrix ($B \times p$), and the gradient update is solved as another matrix of the same size, which is then added to the original matrix.

---

> ### Author Response · Authors · 2024-12-10
> **(Continued) Updates on Algorithm Description, Notations, and so on.**
>
> **4. Forward/Backward Passes of Sinkhorn Loss:**
> > The authors talk of forward and backward passes of Sinkhorn loss function. Please define what this means to someone who has seen forward and backward passes only in the context of neural networks.
>
> We have replaced the terms “forward” and “backward” passes with “calculation” and “differentiation” of the Sinkhorn loss to avoid confusion with neural network contexts. The “forward” pass computes the Sinkhorn loss, while the “backward” pass computes its gradient. Moreover, we added Algorithm 2 (SLCD) in the revised manuscript to provide a detailed description of the Sinkhorn loss computation and differentiation process, which should help clarify the terminology and the mathematical operations involved. The Algorithm is given below.
>
> Algorithm 2: Sinkhorn Loss Computation and Differentiation (SLCD)
>
> Solution Stage:
> 1. Use L-BFGS to solve $\beta^* = \argmin_{\beta} f(\beta)$, where
>    $$f(\beta) = -\alpha^*(\beta)^{\mathrm{T}}a - \beta^{\mathrm{T}}b + \lambda^{-1}$$
>    and
>    $$\nabla_{\tilde{\beta}}f = \tilde{T}(\beta)^{\mathrm{T}}\mathbf{1}_n - \tilde{b}.$$
>
> 2. Compute $\alpha^*$ using:
>    $$\alpha\^*(\beta)_i = \lambda^{-1}\log a_i - \lambda^{-1}\log\left[\sum\_{j=1}\^m e\^{\lambda(\beta_j - M\_{ij})}\right].$$
>
> 3. Compute the transport plan $T(\beta)$ as:
>    $$T(\beta) = e_\lambda[\alpha^*(\beta) \oplus \beta - M].$$
>
> 4. Compute the Sinkhorn loss using:
>    $$S_\lambda(\mu,\nu) = \langle T^*, M \rangle.$$
>
> Differentiation Stage:
> 1. Compute the analytical derivative:
>    $$\nabla_M S_\lambda(\mu,\nu) = T^* + \lambda(s_u \oplus s_v - M) \odot T^*.$$
>
> **5. Gradient Descent vs. L-BFGS:**
> > How does gradient descent compare with L-BFGS for the regularized OT problem. What is known in terms of stability and convergence of gradient descent?
>
> Gradient descent uses only first-order information, potentially resulting in slower convergence. L-BFGS uses approximate second-order information to improve convergence speed. Although we do not directly compare L-BFGS with plain gradient descent, we compare with the Sinkhorn-log method, which should have similar performance as gradient descent methods. Our results in Figure 2 show that L-BFGS converges faster, especially for small $\lambda^{-1}$.
>
> **6. Clarification in Theorem 5.2:**
> > In Theorem 5.2, what is epsilon? It is not defined before it appears in the guarantees.
>
> We have revised the presentation of Theorem 5.2 to define $\epsilon$ before it is used, ensuring that the statement and proof are fully self-contained and clear.
>
> **7. Clarification in Theorem 5.4:**
> > In theorem 5.4, what is B fixed or increasing? In the explanation that follows "*when the number of points n grows large", what is 'n'? *
>
> We corrected the typo in the description after Theorem 5.4 and clarified the roles of $B$ and $n$. In the revised version, the statement and the growth conditions for these parameters are clearly stated and consistent.
>
> **8. Implications of L-BFGS/Adam and Duality in Algorithm 1:**
> > Algorithm 1 has only gradients of Sinkhorn loss. What is the implication of the results on L-BFGS/Adam and duality on Algorithm 1?
>
> Our approach involves a two-layer optimization structure:
> - The **outer** optimization uses gradient descent methods like Adam to generate the coreset, with the gradient being the gradient of Sinkhorn loss.
> - The **inner** optimization solves the Sinkhorn problem, where we use BFGS and consider the dual problem to efficiently compute and differentiate the Sinkhorn loss.
>
>     To clarify this structure, we have added Algorithm 2 (SLCD) in the revised manuscript. The inner problem’s BFGS optimization addresses the dual formulation of Sinkhorn loss, while the outer loop employs gradient updates (e.g., Adam) for the coreset construction.
>
> Warm regards,
> *The Authors*

---

### Review · Reviewer_2HMD · 2024-11-25

**Summary Of Contributions:**

The paper proposes an important problem towards coreset selection particularly Wasserstein coresets using Sinkhorn loss. To adress stability issues in Sinkhorn algorithms, authors propose stable algorithms and analyse via experimentation. The paper extensively studies the stabiliy analysis of L-BFGS from 1989 paper which is utilised in several recent works to solve for alternative to forward pass. Further through their experiments, the authors showcase how their approach performs better than other existing methods in terms of sample selection quality and efficiency.

**Audience:**

Yes

**Broader Impact Concerns:**

There are no broader impact concerns in this work.

**Claims And Evidence:**

Yes

**Requested Changes:**

- Fig 5. Runtime analysis if it is possible to overlay Coreset Runtime analysis w.r.t Full data training for a better comparison.
- Improve Figure descriptions: For example Fig 2: pls include the definition of $\lambda$ and high level observations from the plot in the figure description, show that at a glance it becomes clear what the figure tries to convey.
- In the same page , description of Fig 2. its written $\eta$ instead of $\lambda$.

**Strengths And Weaknesses:**

**Strengths**
The paper is well-written overall with clear explainations in issues faced under Sinkhorn stability. Further the convergence analysis and stability proofs strengthen the arguments for the paper's main proposal against solving instability issues of traditional Sinkhorn algorithm settings. As part of experiments, the authors showcase better coreset sample quality selection than other sampling approaches (e.g. Fig 4) as well as achieving better efficiency.

Further the authors showcases new theoretical results showcasing stability analysis for Sinkhorn loss along with numerical analysis to support the same.

**Weakness**

- Not clear about the essence of Proposition 2.1. Why is this proposition important?
- Eq 4 arguments are not clear. How is Eq 3 can be directly refined to Eq 4? Since Sinkhorn loss has 3 arguments in previous definition.
- I might have missed the definition of BFGS from the paper. Could authors elaborate on it more in their writeup? In terms about the description of the algorithm.
- Can you elaborate a bit more on the Sinkhorn log algorithm. Is it the entire u is transformed to log scale or just $M_e$.
- I am not sure in terms of baselines, if you could also have tried out submodular based sample selection, as they also provide better guarantees

---

> ### Author Response · Authors · 2024-12-10
> **Updates on Notations, Algorithm Description, Added Baseline Comparison and so on.**
>
> Dear Reviewer 2HMD,
>
> We sincerely appreciate your time and effort in reviewing our manuscript and offering valuable feedback. Below are our point-by-point responses to your comments:
>
> **1. Importance of Proposition 2.1:**
> > Not clear about the essence of Proposition 2.1. Why is this proposition important?
>
> Thank you for highlighting this. We have revised the text preceding Proposition 2.1 to clarify its purpose. The proposition distinguishes between two variants of the Sinkhorn loss found in the literature: the *sharp* version and the *regularized* version. We choose the sharp version because it achieves a faster convergence rate for approximating the Wasserstein distance, as shown by Proposition 2.1. This theoretical result underpins our choice and guides our subsequent analysis.
>
> **2. From Equation 3 to Equation 4:**
> > Eq 4 arguments are not clear. How is Eq 3 can be directly refined to Eq 4? Since Sinkhorn loss has 3 arguments in previous definition.
>
> We apologize for the previous notational confusion. Initially, we introduced two abbreviations for the Sinkhorn loss, $S_\lambda(\mu,\nu)$ and $S_\lambda(M,a,b)$, which proved unnecessary and caused confusion like "Sinkhorn loss has 3 arguments in previous definition". In the revised manuscript, we consistently use $S_\lambda(\mu,\nu)$ throughout. With this clarification, Eq. 5 (previously Eq. 3) now directly refines into the subsequent Eq. 6 (previously Eq. 4) because the Sinkhorn loss $S_\lambda(\mu,\nu)$ serves as an approximation to the Wasserstein distance.
>
> **3. BFGS Definition and Algorithm Description:**
> > I might have missed the definition of BFGS from the paper. Could authors elaborate on it more in their writeup? In terms about the description of the algorithm.
>
> We agree that additional clarification was needed for how BFGS is employed in our methods. We have introduced Algorithm 2 (SLCD) and Lemma 1 in the revised manuscript, which details the computation, differentiation, and optimization steps involved in the Sinkhorn loss calculation. This includes how the BFGS algorithm, a quasi-Newton method that leverages approximate Hessian information, is employed. For definition of L-BFGS, we have also cited Liu and Nocedal (1989) for readers interested in further details on BFGS. The detail of Algorithm 2 is listed below:
>
> Algorithm 2: Sinkhorn Loss Computation and Differentiation (SLCD)
>
> Solution Stage:
> 1. Use L-BFGS to solve $\beta^* = \argmin_{\beta} f(\beta)$, where
>    $$f(\beta) = -\alpha^*(\beta)^{\mathrm{T}}a - \beta^{\mathrm{T}}b + \lambda^{-1}$$
>    and
>    $$\nabla_{\tilde{\beta}}f = \tilde{T}(\beta)^{\mathrm{T}}\mathbf{1}_n - \tilde{b}.$$
>
> 2. Compute $\alpha^*$ using:
>    $$\alpha^*(\beta)_i = \lambda^{-1}\log a_i - \lambda^{-1}\log\left[\sum\_{j=1}\^m e\^{\lambda(\beta_j - M\_{ij})}\right].$$
>
> 3. Compute the transport plan $T(\beta)$ as:
>    $$T(\beta) = e_\lambda[\alpha^*(\beta) \oplus \beta - M].$$
>
> 4. Compute the Sinkhorn loss using:
>    $$S_\lambda(\mu,\nu) = \langle T^*, M \rangle.$$
>
> Differentiation Stage:
> 1. Compute the analytical derivative:
>    $$\nabla_M S_\lambda(\mu,\nu) = T^* + \lambda(s_u \oplus s_v - M) \odot T^*.$$
>
>
> **4. Sinkhorn-log Algorithm:**
> > Can you elaborate a bit more on the Sinkhorn log algorithm? Is it the entire $u$ transformed to log scale or just $M_e$?
>
> In the Sinkhorn-log algorithm, we apply the log transformation to the entire vector $u$ to improve numerical stability. However, this comes at the cost of slower convergence compared to our proposed SLCD algorithm. We have clarified this point in the revised manuscript.
>
> **5. Additional Baselines (Submodular-Based Selection):**
> > I am not sure in terms of baselines, if you could also have tried out submodular based sample selection, as they also provide better guarantees.
>
> We have included new comparisons, such as k-means centroids and the kernel thinning method, in the revised manuscript. If you have specific recommendations for submodular-based approaches, we would be happy to incorporate them. Experimental results with new baselines are presented in Figures 3, 4, and 9 in the revised manuscript. Figure 3 visually compares the sample selection quality between subsampling methods on 2-dimensional datasets, while Figures 4 and 9 show the MMD and Wasserstein distance for simulated datasets and image datasets.
>
> **6. Runtime Analysis (Fig. 5):**
> > Fig 5. Runtime analysis if it is possible to overlay Coreset Runtime analysis w.r.t Full data training for a better comparison.
>
> We have expanded the runtime analysis to larger datasets, with sample size from $100{,}000$ to $1{,}000{,}000$ in the Fig. 5 of revised manuscript. We didn't clearly understand the meaning of your questions. We welcome further clarification regarding your suggestion of overlaying the Coreset Runtime analysis against full-data training. If you could provide more details, we will consider implementing this comparison in future revisions.

---

> ### Author Response · Authors · 2024-12-10
> **(Continued) Updates on Figure Description.**
>
> **7. Improving Figure Descriptions:**
> > Improve Figure descriptions: For example Fig 2: pls include the definition of $\lambda$ and high-level observations from the plot in the figure description, so that at a glance it becomes clear what the figure tries to convey.
>
> We have improved the figure captions throughout the manuscript. We have also removed the global definition $\eta = \lambda^{-1}$ to maintain consistency and avoid confusion. Now $\lambda^{-1}$, instead of $\eta$, is used for the Sinkhorn loss regularization parameter.
>
> Warm regards,
> *The Authors*

---

### Public Comment · ~Prabhant_Singh2 · 2025-06-18
**Code request**

Hi, can you please provide the updated repository for the paper? The current GitHub repo seems to be empty.

---

> ### Author Response · Authors · 2025-06-19
> **Code Availability Update**
>
> Thank you for your interest in our work. The code for some of the experiments has been made public. Please feel free to check.

---

### Decision · Action_Editor_Yv61 · 2025-01-10

**Recommendation:** Accept with minor revision

**Comment:**

This paper is a solid contribution, and definitely TMLR-worthy after the revisions made during the review process.

There are three minor fixes that need to be made to the manuscript prior to publication.
1. Figure 9 does not have kernel thinning results, but kernel thinning is in the figure legends. In your response to Reviewer Gxub you state that "For large image datasets MNIST and FashionMNIST, the kernel thinning method proved very slow; thus, we did not include the results for kernel thinning on image datasets." Please fix the legends in Figure 9 and add this explanation to the paper.
2. Léon Bottou's name did not typeset correctly in the reference for the Wasserstein GAN paper.
3. Benoît Kloeckner's name did not typeset correctly in the reference for "Approximation by finitely supported measures"

**Audience:**

I expect the analytic expression for the derivative of the Sinkhorn loss, proposed algorithms for computing the Sinkhorn loss and its derivative in an efficient and stable manner, and the analysis in this paper to be picked up by other researchers for their own work, as these have many potential applications. The proposed coreset algorithm is also of interest to the TMLR community.

**Claims And Evidence:**

This paper has three main contributions:
1. Derivation of an analytic expression for the derivative of the Sinkhorn loss.
2. Algorithms for computing the Sinkhorn loss and its derivative in an efficient and stable manner, along with analysis and empirical tests to verify the efficiency and stability claims.
3. An algorithn for finding a coreset of a dataset by optimizing the Sinkhorn loss, along with empirical tests to verify its better performance than a number of baseline coreset algorithms.

The first two contributions are extremely well supported. The third contribution is supported by limited, but convincing, evidence. To be completely convincing, more empirical validation of the task-agnostic claim on a broader set of tasks would have been needed.